# Direct interaction between TDP-43 and Tau promotes their co-condensation, while suppressing Tau fibril formation and seeding

Francesca Simonetti [ID] [1,2,3], Weijia Zhong [ID] [4], Saskia Hutten [3], Federico Uliana [ID] [3], Martina Schifferer [1,5], Ali Rezaei [ID] [1,2], Lisa Marie Ramirez [6,7], Janine Hochmair [ID] [8], Rithika Sankar [ID] [8], Anusha Gopalan [9,10], Fridolin Kielisch [ID] [10], Henrick Riemenschneider [ID] [1], Viktoria Ruf [5], Carla Schmidt [ID] [11], Mikael Simons [1,2,12], Markus Zweckstetter [6,7], Susanne Wegmann [ID] [8], Tammaryn Lashley [ID] [13], Magdalini Polymenidou [ID] [4], Dieter Edbauer [ID] [1,2,12 ✉] & Dorothee Dormann [ID] [3,10,12 ✉]

## Abstract

**Neuronal aggregates of Tau are a hallmark of Alzheimer's disease (AD), but more than half of the patients exhibit additional TDP-43 inclusions, while some have co-aggregates of the two proteins. The presence of such co-aggregates is associated with increased disease severity, although whether there is a causal relationship remains unclear. Here, we demonstrate that Tau and TDP-43 mutually promote each other's condensation through direct interaction in vitro, forming irregularly-shaped or multiphasic co-condensates with lower TDP-43 mobility, but higher Tau mobility. While Tau promotes TDP-43 aggregation in vitro, TDP-43 suppresses formation of Tau fibrils and instead causes formation of oligomeric Tau and Tau/TDP-43 species. These co-assemblies hinder Tau seeding in a biosensor assay specific for proteopathic Tau seeds. Consistent with these data, insoluble material extracted from AD patient brains with Tau/TDP-43 co-aggregates exhibits reduced Tau seeding compared to AD patient brains with Tau aggregates only. In contrast, patient-derived extracts from AD patient brains with Tau/TDP-43 co-aggregates are highly potent in seeding new TDP-43 aggregates in a TDP-43 reporter cell line. Our results suggest that direct interaction between TDP-43 and Tau may suppress Tau pathology, while promoting TDP-43 pathology in Alzheimer's disease patients.**

**Keywords** Alzheimer's Disease; Tau; TDP-43; Phase Separation; Seeding
**Subject Categories** Molecular Biology of Disease; Neuroscience

## Introduction

Intracellular protein aggregates are a common feature of all neurodegenerative diseases. Protein inclusions occur in neurons and/or glial cells and are intimately linked to the process of neurodegeneration (Taylor et al, 2002). In Alzheimer's disease (AD), the microtubule-associated protein Tau, which regulates axonal transport and synapse function, mislocalizes from axons into insoluble cytosolic inclusions called neurofibrillary tangles (NFTs) (Guo et al, 2017). Since Tau deposits in AD correlate with neurodegeneration and cognitive decline, the stereotypic spreading of Tau pathology from the transentorhinal regions to the neocortex is used for disease staging (Braak and Braak, 1995). Data suggest that Tau oligomers may be more neurotoxic than fibrous, amyloid-like Tau aggregates (Berger et al, 2007; Lasagna-Reeves et al, 2011; Takeda et al, 2015; Usenovic et al, 2015). Over 60% of AD patients additionally feature cytosolic deposits of the nuclear DNA/RNA-binding protein TDP-43 (TAR DNA binding protein of 43 kDa) (Amador-Ortiz et al, 2007; Josephs et al, 2014b, 2014a; Kadokura et al, 2009; Tomé et al, 2020, 2021; Tremblay et al, 2011), first discovered in the affected brain regions of amyotrophic lateral sclerosis (ALS) and frontotemporal dementia (FTD) patients (Neumann et al, 2006). TDP-43 is involved in many steps of RNA processing (Ratti and Buratti, 2016), and its nuclear loss and cytoplasmic aggregation causes subtle yet detrimental dysregulation of mRNA processing, for example, incorporation of cryptic exons into mRNA (Ling et al, 2013; Melamed et al, 2019; Polymenidou et al, 2011; Tziortzouda et al, 2021).

In AD, TDP-43 deposits are mainly found in the limbic regions and frontal cortex (Amador-Ortiz et al, 2007; Arai et al, 2009; Josephs et al, 2015; Latimer and Liachko, 2021; Meneses et al, 2021;

[1] Deutsches Zentrum für Neurodegenerative Erkrankungen (DZNE), Munich 81377, Germany. [2] Graduate School of Systemic Neurosciences (GSN), Planegg 82152, Germany. [3] Johannes Gutenberg University (JGU), Institute of Molecular Physiology, Mainz 55122, Germany. [4] University of Zurich, Zurich 8006, Switzerland. [5] Center for Neuropathology and Prion Disease, Faculty of Medicine LMU, Munich 81377, Germany. [6] Max Planck Institute for Multidisciplinary Sciences, Göttingen 37075, Germany. [7] Deutsches Zentrum für Neurodegenerative Erkrankungen (DZNE), Göttingen 37075, Germany. [8] Deutsches Zentrum für Neurodegenerative Erkrankungen (DZNE), Berlin 10117, Germany. [9] Johannes Gutenberg University (JGU), Biocenter, Light Microscopy Core Facility, Mainz 55122, Germany. [10] Institute of Molecular Biology (IMB), Mainz 55122, Germany. [11] Johannes Gutenberg University (JGU), Department of Chemistry-Biochemistry, Mainz 55122, Germany. [12] Munich Cluster of Systems Neurology (SyNergy), Munich 81377, Germany. [13] UCL Queen Square Institute of Neurology, London WC1N 3BG, UK. ✉ E-mail: dieter.edbauer@dzne.de; ddormann@uni-mainz.de

Tomé et al, 2020). TDP-43 deposition in AD is associated with greater hippocampal volume loss and more severe cognitive impairment compared to cases without TDP-43 pathology (Josephs et al, 2017, 2014b; Thomas et al, 2020), suggesting an important contribution of TDP-43 to neurodegeneration. TDP-43 aggregates were shown to colocalize with Tau NFTs in the amygdala and hippocampus of AD patients (Davis et al, 2017; Smith et al, 2017; Tomé et al, 2021), and proximity ligation and co-immunoprecipitation experiments have suggested that TDP-43 interacts with Tau in AD brains (Tomé et al, 2021). Studies in animal models support the idea that the presence of TDP-43 exacerbates Tau pathology. In *C. elegans*, TDP-43 enhances Tau neurotoxicity, resulting in neuronal dysfunction and pathological Tau accumulation (Latimer et al, 2022). TDP-43 overexpression in APP/PSEN1 mice decreased Aβ plaque deposition but increased Tau aggregation (Davis et al, 2017), and cytoplasmic accumulation of endogenous phosphorylated TDP-43 was reported in two different Tau transgenic mouse models, but not in mouse models of Aβ deposition, α-synucleinopathy or Huntington's disease (Clippinger et al, 2013). However, whether Tau and TDP-43 directly interact and influence each other's condensation, aggregation or seeding behavior remains unknown.

Both Tau and TDP-43 contain extended intrinsically disordered regions (IDRs) that drive various self-assembly forms, including oligomers and liquid-like condensates that form through phase separation (PS) (Ambadipudi et al, 2017; Wang et al, 2018; Wegmann et al, 2018). While PS is essential for the formation of membrane-less organelles with various cellular functions (Alberti et al, 2019), there is evidence that this process can give rise to amyloid-like aggregates through aberrant liquid-to-solid phase transitions (Alberti and Dormann, 2019; Alberti and Hyman, 2016; Nedelsky and Taylor, 2019; Patel et al, 2015). However, a protective function of condensates as protein sinks that suppress fibril formation has also recently been reported (Das et al, 2025; Lipiński et al, 2022). Both TDP-43 and Tau have been shown to form liquid- or gel-like condensates in vitro, and it has been postulated that these condensates may facilitate pathological aggregation (Ambadipudi et al, 2017; Kanaan et al, 2020; Wang et al, 2018; Wegmann et al, 2018; Yan et al, 2025; Zhang et al, 2017). Whether a direct interaction of TDP-43 and Tau occurs and how this might affect their phase transition and aggregation behavior is still unknown.

Remarkably, oligomeric or aggregated TDP-43 and Tau can be released and taken up by neighboring cells and thereby spread from cell to cell (Brettschneider et al, 2015; De Rossi et al, 2021; Feiler et al, 2015; Rummens et al, 2025; Scialò et al, 2025; Takeda et al, 2015), providing additional possibilities for their intra- or extracellular encounters. These mechanisms give rise to the stereotypical spreading of protein aggregates throughout the brain, which has been observed for the prion protein (PrP), Tau, α-synuclein and TDP-43 (Dujardin and Hyman, 2019; Jucker and Walker, 2011; Polymenidou and Cleveland, 2011; Uemura et al, 2020). To what extent TDP-43 and Tau influence each other's seeding behavior and when Tau seeds encounter TDP-43 and vice versa is still poorly understood.

Here, we show that TDP-43 and Tau significantly influence each other's condensation and aggregation behavior in vitro and directly interact in their condensed state. While Tau promotes TDP-43 aggregation, TDP-43 prevents the generation of Tau fibrils and causes the formation of small, oligomeric Tau and Tau/TDP-43 species. These species suppress Tau seeding in Tau biosensor cells specific for proteopathic Tau seeds. In line with these findings, seeding experiments with SarkoSpin brain extracts derived from AD patients with Tau/TDP-43 co-pathology reveal a reduced Tau seeding capacity compared to extracts from AD patients with Tau-only pathology, supporting the notion that presence of TDP-43 reduces Tau seeding. However, seeding experiments with a TDP-43 reporter cell line show that SarkoSpin extracts from AD brains with Tau/TDP-43 co-pathology have a high TDP-43 seeding capacity, suggesting the presence of highly seeding-competent TDP-43 species in these patients. Together, our data suggest that TDP-43 co-pathology, present in up to 60% of AD cases, may suppress Tau fibrillation and seeding, while promoting TDP-43 aggregation and seeding.

# Results

## Full length Tau induces the formation of large, irregular TDP-43 condensates in vitro

To gain a molecular understanding of how Tau and TDP-43 influence each other's assembly and aggregation behavior, we performed several in vitro assays with fluorescently labeled recombinant proteins. Full-length wild-type Tau (2N4R isoform) was labeled with DyLight 650, and TDP-43, fused with a maltose-binding protein (MBP) tag for solubilization (Wang et al, 2018), was labeled with Alexa Fluor 488.

We induced TDP-43 condensation by cleaving the MBP-His$_6$ tag from TDP-43-MBP-His$_6$ using TEV protease in a physiological buffer, and visualized the resulting condensates by spinning disk confocal microscopy (Fig. 1A). In the absence of Tau, TDP-43 formed distinct round condensates, but the addition of Tau at equimolar concentration induced the formation of larger, irregularly shaped TDP-43 structures that also contained Tau (Fig. 1B; Movies EV1 and EV2). Moreover, unlabeled Tau induced similar large, irregularly shaped TDP-43 condensates as DyLight650-labeled Tau, excluding effects of dye conjugation (Fig. 1C). Titration of unlabeled Tau (0.2–10 μM) showed that the formation of these irregular TDP-43 condensates already occurred at a Tau:TDP-43 ratio of 1:10 (Appendix Fig. S1A). In addition, neither recombinant MBP, nor α-synuclein, another neurodegeneration-linked protein prone to aggregation and phase-separation (Ray et al, 2020), affected the number or morphology of TDP-43 condensates when added at different concentrations (Fig. 1C). We also observed that Lumidyne650-labeled α-synuclein was excluded from Alexa488-labeled TDP-43 condensates, contrary to DyLight650-labeled Tau (Appendix Fig. S1B,C). Finally, quantitative analysis of condensate images confirmed that only the addition of Tau, but not MBP or α-synuclein, reduced the number and roundness but increased the size of TDP-43 condensates (Fig. 1D).

To further investigate the dynamic properties of fluorescently labeled TDP-43 condensates in presence or absence of Tau or control proteins, we performed Fluorescence Recovery After Photobleaching (FRAP), an established technique used to assess the mobility of molecules within condensates. Fluorescence recovery was significantly slower for the TDP-43 condensates in presence of Tau, compared to condensates formed by TDP-43 alone or with MBP or α-synuclein (Fig. 1E,F).

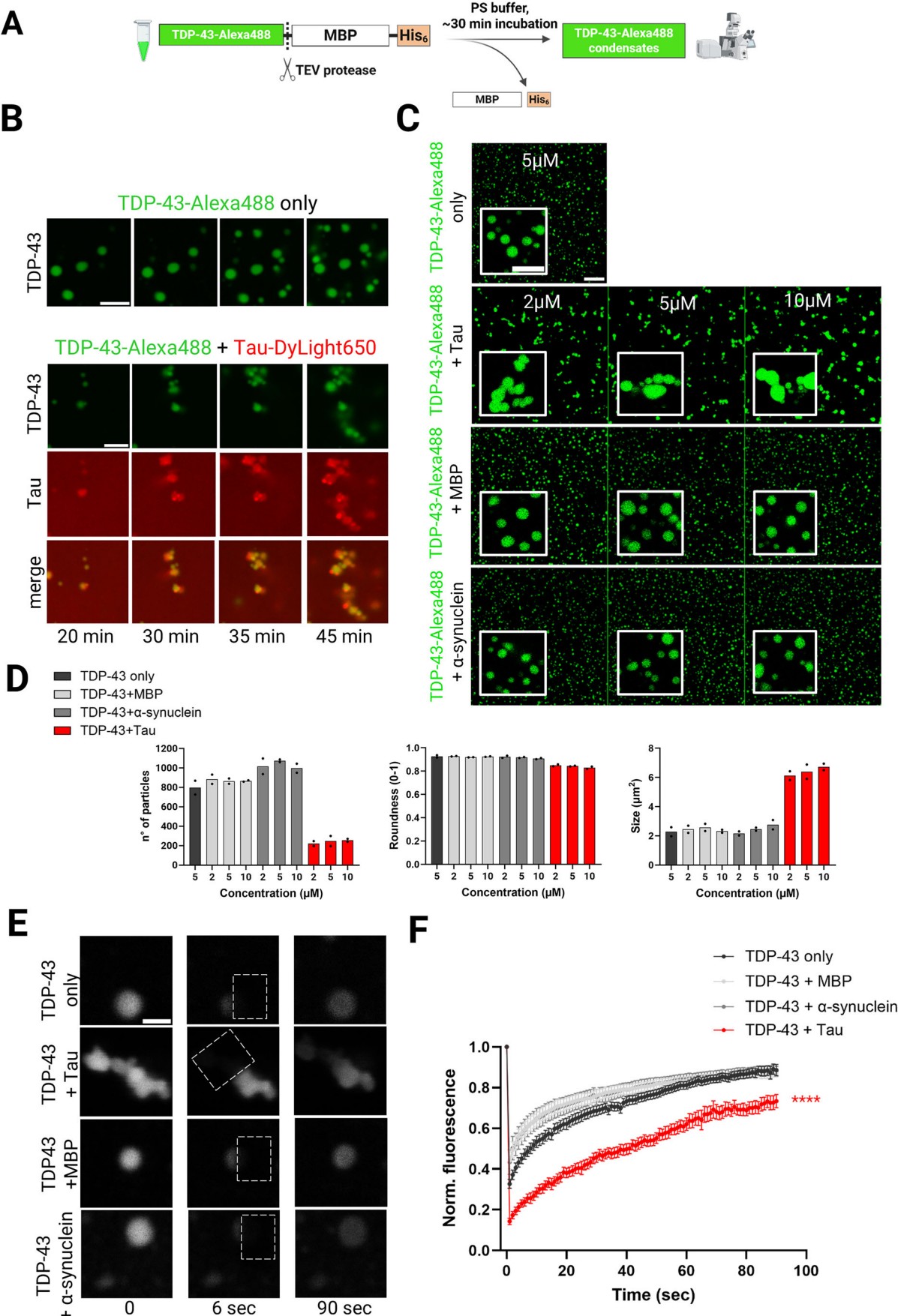

**Figure 1.   Tau, but not MBP or α-synuclein, causes formation of large irregular TDP-43 condensates.**

(A) Scheme of TDP-43 phase separation assay upon TEV-cleavage of the MBP-solubility tag; created with BioRender.com. (B) Time series of Alexa488-labeled TDP-43 condensate formation (8 µM) in presence or absence of DyLight650-labeled Tau (8 µM), from 20 to 45 min after cleavage with TEV protease; images represent frames of Movies EV1 and EV2; scale bar: 10 µm. (C) Confocal images of Alexa488-labeled TDP-43 condensates (5 µM) in presence of unlabeled Tau, MBP, or α-synuclein (2, 5 or 10 µM), 30 min after TEV-cleavage. Scale bar: 15 µm in overview and 8 µm in inset. (D) Graphs showing number of particles, circularity, and size of the TDP-43 condensates. Individual data points from (n = 2) biological replicates are shown. (E) Representative FRAP images of Alexa488-labeled TDP-43 condensates (15 µM) at the indicated time points, in the absence or presence of equimolar unlabeled Tau, MBP, or α-synuclein, within 15 min of TEV protease addition. Dotted box indicates the bleached area (half of the condensate). Scale bar: 1 µm. (F) FRAP recovery curves following half-bleaching of Alexa488-labeled TDP-43 condensates in absence or presence of Tau, MBP, and α-synuclein. Data represent ± SEM from 11 to 12 individual condensates analyzed per condition in four biological replicates. ***$P < 0.0001$ by one-way ANOVA with Dunnett´s multiple comparison test to TDP-43 only condition; significant difference was determined by calculating the area under the curve (AUC) for each condition.

Next, we explored whether specific domains of Tau or its slightly positive net charge ($+1.44$ at pH 7.4) are responsible for altering TDP-43 phase separation. Unlike Tau, TDP-43 and the control proteins MBP and α-synuclein all have a negative net charge at pH 7.4 (TDP-43: $-5.27$; MBP-Tev-His: $-13.29$; α-synuclein: $-9.73$). Therefore, it seems possible that the positive net charge of Tau is responsible for the observed effect. To investigate this further, we compared the effect of full-length non-phosphorylated Tau (2N4R) with in vitro phosphorylated Tau (pTau: net charge ca. $-5.56$ (Chakraborty et al, 2024)), which is more negatively charged, the N-terminal domain of Tau (Tau-NTD, amino acids (aa) 1–256), comprising the proline-rich region that drives Tau phase separation (Zhang et al, 2020), which has a negative net charge ($-7.6$ at pH 7.4), and the Tau repeat domain (RD) fragment (aa 244–372 containing repeat domains R1-R4), which includes the microtubule binding region and core part of Tau fibrillar aggregates (Wille et al, 1992) and has a positive net charge of $+9.1$ at pH 7.4 (Fig. EV1A,B). pTau exhibited a marked accumulation within TDP-43 condensates and showed a significant increase in colocalization compared to the unphosphorylated Tau control (Fig. EV1C,D). This suggests that phosphorylation of Tau, which is typically observed in AD, enhances its interaction with TDP-43. In contrast, neither the negatively charged Tau-NTD nor the positively charged Tau-RD fragment partitioned into TDP-43 condensates or altered the formation or morphology of TDP-43 droplets (Fig. EV1C,D). Together, these results suggest that clustering of TDP-43 condensates requires the combination of N- and C-terminal domains of Tau and does not depend on the positive net charge of Tau.

Collectively, our data show that even at low concentrations, full-length Tau alters the phase separation behavior of TDP-43 by co-partitioning into its condensates, causing the formation of large, irregular TDP-43 condensates with reduced dynamics.

## TDP-43 and Tau directly interact in vitro in the condensate state

To verify whether TDP-43 and Tau can interact directly during condensate formation, we employed chemical crosslinking with di-succimidylsuberate (DSS) and MS identification (XL-MS) (Boczek et al, 2021; Czub et al, 2025; Sahin et al, 2023). In XL-MS, the bifunctional crosslinker DSS covalently reacts with amine groups on two residues situated within 30 Å of each other under native conditions. This method not only provides evidence of direct protein–protein interaction, but can also lead to the identification of the interacting regions. We used this approach to identify inter-protein cross-linked peptides between TDP-43 and Tau, or between

TDP-43 and MBP as control. As in Fig. 1, we induced phase separation of TDP-43 by removing the MBP tag with TEV protease and performed the crosslinking reaction in the bulk condition or in the pellet obtained from centrifugation of condensates (see scheme in Fig. 2A).

Several spectra were assigned to inter-protein crosslinks between TDP-43/Tau and TDP-43/MBP (for a list of all identified cross-linked peptides, see Appendix Fig. S2). The identified crosslinks between TDP-43 and MBP may result from incomplete proteolytic cleavage of the TDP-43-MBP fusion protein or from the physical interaction between the two separate proteins. Notably, these interactions were detected only in the bulk condition (Fig. 2B). This is consistent with the threefold lower number of MBP spectra in the pellet compared to the bulk condition (Appendix Fig. S2B), reflecting the high solubility of MBP. In bulk conditions, we identified inter-protein crosslinks between TDP-43 and Tau (residues 43–132), as well as between MBP and TDP-43; however, these crosslinks were much less abundant compared to the condition without Tau, suggesting a competition between MBP and Tau.

In the pellet condition, inter-protein crosslinks were observed exclusively between residues of TDP-43 and Tau (residues 74 and 225, Fig. 2B, identified spectra for the two peptides are reported in Appendix Fig. S2C,D). These crosslinked peptides involve the N-terminal domain (NTD) of TDP-43 and the repetitive proline-rich domain (PRD) of Tau. Since the NTD of TDP-43 is known to mediate its oligomerization (Afroz et al, 2017; Chang et al, 2012), its interaction with Tau may influence TDP-43 self-association.

To corroborate the regions involved in the interaction of TDP-43 and Tau, we employed an in silico approach to predict the binding regions of TDP-43 and Tau. Using a fragmentation approach (Lee et al, 2024), we modeled with Alphafold-Multimer (version 2.3) (Jumper et al, 2021) the interactions between 6 regions of TDP-43 and 10 regions of Tau, generating 60 models based on pairwise permutations of all fragments. Among the 60 generated models, the top-ranked and third-ranked models based on pDockQ scores include the regions identified by XL-MS, supporting our experimental findings (Appendix Fig. S2E).

Overall, these results demonstrate that TDP-43 and Tau can directly interact in the condensates, primarily involving the NTD of TDP-43 and the PRD of Tau. However, since the C-terminal domain IDR (CTD-IDR) of TDP-43 is a low-complexity sequence lacking nucleophile functional groups for crosslinking with DSS and lysine and arginine for tryptic digest, we cannot exclude the involvement of additional interactions, e.g. involving the TDP-43 CTD-IDR region.

## A

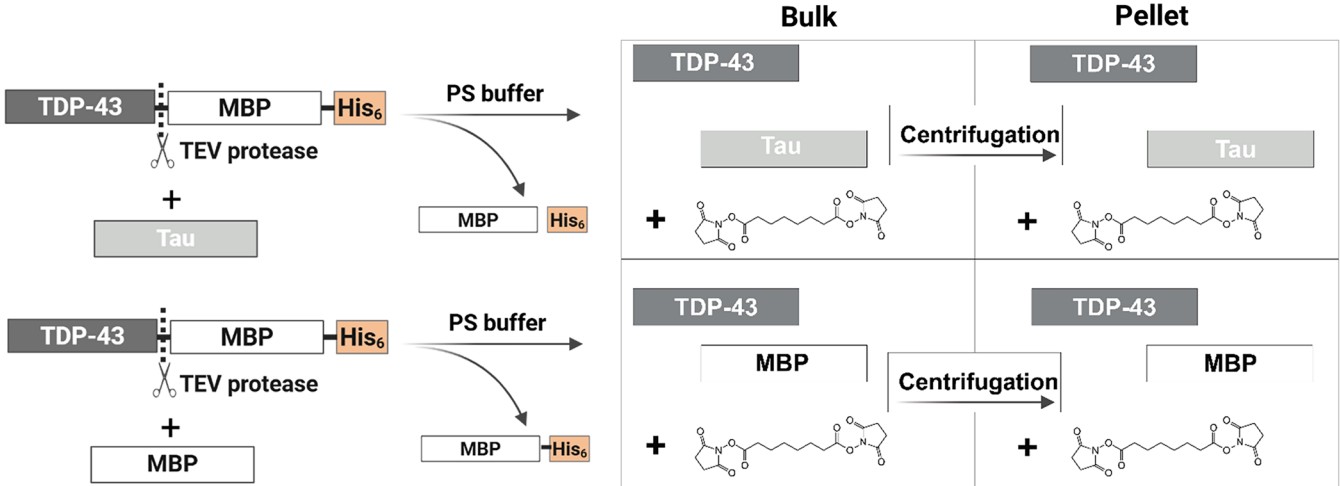

## B

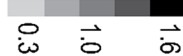

**Average number of identified inter-XL spectra**

**Figure 2.   TDP-43 and Tau directly interact in condensates formed in vitro.**

(A) Scheme of XL-MS experiment; created with BioRender.com. (B) Inter-protein crosslinked peptides identified for TDP-43 + Tau and TDP-43 + MBP under bulk and pellet conditions following condensate induction ($n = 3$) independent replicates. Only crosslinks detected in at least two out of three replicates, with a peptide-spectrum match (PSM) false discovery rate (FDR) < 1% from light and heavy isotope-labeled DSS crosslinking experiments, were included. Inter-protein crosslinks are shown in grey, with line thickness proportional to the spectral count (i.e., relative abundance).

## Tau promotes TDP-43 aggregation in vitro

To test whether Tau affects TDP-43 aggregation in vitro, we first performed semi-denaturing detergent agarose gel electrophoresis (SDD-AGE). In this previously established TDP-43 aggregation assay (French et al, 2019; Gruijs da Silva et al, 2022), TDP-43-MBP was incubated in the presence or absence of Tau, MBP, or α-synuclein in an aggregation-promoting buffer with agitation for 30 min and then incubated for several days (Fig. 3A). TEV cleavage was omitted to slow aggregation and enable a better visualization of the high molecular weight (HMW) species of TDP-43-MBP that form over time (Gruijs da Silva et al, 2022). The addition of Tau, but not MBP or α-synuclein, significantly accelerated the formation of HMW species of TDP-43-MBP (Fig. 3B,C). Similar results were obtained when the SDD-AGE assay was performed with slightly faster aggregation kinetics by removing the MBP solubility tag with TEV protease (Appendix Fig. S3). Thus, both in presence and absence of the MBP-tag, Tau promotes the formation of SDS-stable TDP-43 assemblies.

To visualize these aggregates by confocal microscopy, we performed the same aggregation assay with Alexa488-labeled TDP-43, triggered by TEV protease cleavage (see scheme in Fig. 3D). The addition of recombinant unlabeled Tau, but not MBP, promoted TDP-43 aggregation in a dose-dependent manner, as quantified by the aggregate area. Instead, α-synuclein increased TDP-43 aggregation only at the highest concentration, but to a lesser extent than Tau (Fig. 3E,F). However, when using fluorescently labeled Tau and α-synuclein, we detected a greater enrichment of Tau than α-synuclein within TDP-43 aggregates (Appendix Fig. S1D,E). This experiment also ruled out any influence of the fluorescent dye, as the promotion of TDP-43 aggregation was comparable to that observed with unlabeled proteins.

Additionally, we assessed TDP-43 aggregation in presence of the Tau variants (pTau, Tau-NTD, Tau-RD) described above (Fig. EV1A,B). Consistent with the results obtained in condensation assays, only pTau, but not Tau-NTD or Tau-RD, showed pronounced accumulation within TDP-43 aggregates and a significant increase in colocalization compared to the unphosphorylatedTau control (Fig. EV1E,F). This confirms that both N- and C-terminal domains of Tau are necessary for partitioning into TDP-43 aggregates and for promoting TDP-43 aggregation. Our data furthermore suggest that phosphorylation of Tau, as seen in AD, may promote the TDP-43/Tau interaction.

## TDP-43 promotes Tau phase separation and TDP-43 and Tau co-condense into multiphasic structures

After having shown an influence of Tau on TDP-43 phase separation and aggregation, we addressed effects in the reverse direction and determine the influence of TDP-43 on Tau condensation. Tau is known to phase separate instantaneously at low salt conditions through complex coacervation in presence of polyanions, such as heparin or RNA, or more slowly upon addition of molecular crowding agents, such as polyethylene glycol (PEG) (Ambadipudi et al, 2017; Hochmair et al, 2022; Ukmar-Godec et al, 2019; Wegmann et al, 2018; Zhang et al, 2017). To examine how TDP-43 affects Tau condensation, we induced Tau phase separation at near physiological salt concentrations (150 mM NaCl) with 10% (w/v) PEG (Fig. 4A), and analyzed phase separation of DyLight488-labeled Tau with Alexa633-labeled TDP-43-MBP (molar ratio of proteins 1:1) by confocal microscopy. Consistent with previous reports (Ambadipudi et al, 2017; Kanaan et al, 2020), Tau alone gradually formed larger condensates over time, but TDP-43 addition strongly promoted Tau phase separation already after 1 h of incubation and the two proteins co-phase separated into very large condensates that exhibited pronounced wetting of the dish surface (Fig. 4B), reminiscent of previously reported Tau:RNA co-condensate wetting of charged surfaces (Hochmair et al, 2022). Contrarily, MBP alone and α-synuclein did not alter the Tau condensation behavior. In addition, we also observed numerous small, internal Tau condensates within the Tau/TDP-43-MBP co-phases (Fig. 4B).

To capture the early stages of this multiphasic Tau/TDP-43 co-condensate formation and monitor their progression over time, we performed time-lapse high-resolution imaging starting at the earliest possible observation time point (10 min) up to 24 h. We observed that TDP-43 and Tau occupied distinct localizations within the condensate already at the 10 min time point (Fig. EV2A; see line profile). Importantly, intra-condensate Tau-assemblies were enclosed within the TDP-43 condensate at 2 h and persisted until 24 h (Fig. EV2B,C). This suggests that the small internal Tau condensates likely form by early de-mixing of Tau and TDP-43 within co-condensates. To investigate the effect of TDP-43 on Tau condensation in the absence of crowding agent, we induced Tau condensate formation by coacervation with RNA-U20 in low salt (Appendix Fig. S4A). Under these conditions, we detected amorphous, aggregate-like Tau structures that were strongly promoted by the presence of TDP-43-MBP, but not by MBP or α-synuclein (Appendix Fig. S4B). These results further support the idea that the TDP-43 specifically promotes Tau condensation and also occurs in the absence of a crowding agent.

To assess the dynamic properties of fluorescently labeled Tau condensates in absence or presence of TDP-43-MBP or control proteins, we performed FRAP experiments (Fig. 4C). In absence of an additional protein or in presence of MBP or α-synuclein, Tau condensates exhibited a lower mobile fraction, indicating limited molecular mobility 1 h after PEG addition. In contrast, Tau condensates formed in the presence of TDP-43-MBP showed significantly increased mobile fraction, with a higher half-time (Fig. 4D,E). This suggests that within the TDP-43/Tau co-phases,

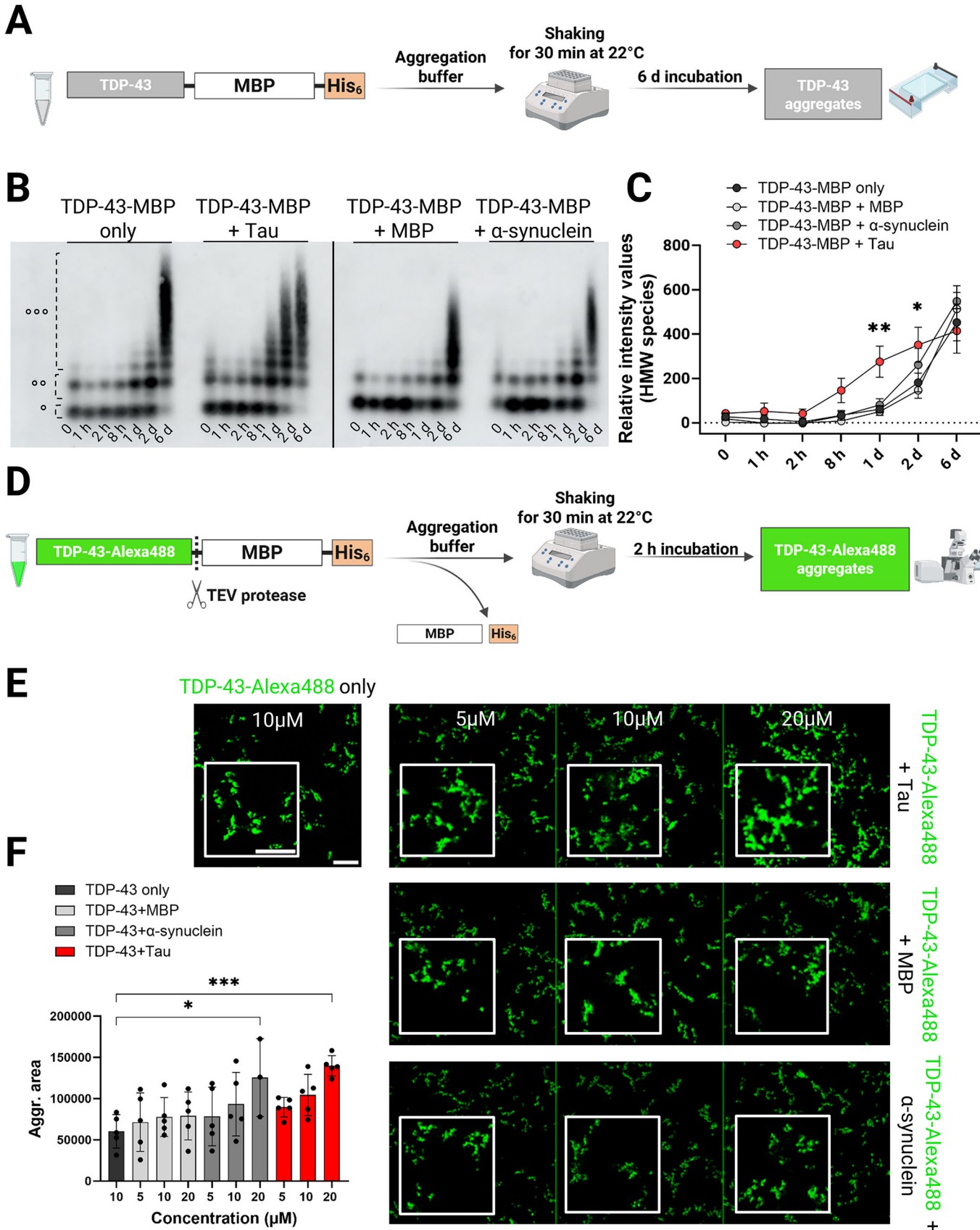

**Figure 3. Tau, but not MBP or α-synuclein, promotes TDP-43 aggregation.**

(A) Scheme of TDP-43 aggregation assay for SDD-AGE experiment; created with BioRender.com. (B) SDD-AGE of TDP-43-MBP (2 μM) in the presence of the indicated protein (2 μM) after agitation and incubation for the presented time period (h = hours, d = days); TDP-43 was visualized by immunoblotting and the middle vertical black line divides two blots which derive from the same experiment and were processed in parallel. °°° = high molecular weight (HMW) species, °° = oligomers and ° = monomers. (C) Quantification of band intensities of the high molecular weight species from ($n = 3$) biological replicates, normalized to the timepoint 0; values show the mean ± SEM; **$P = 0.0039$ and *$P = 0.0273$ by two-way ANOVA with Dunnett's multiple comparison test to TDP-43-MBP only condition at the respective time points (0, 1 h, 2 h, 8 h, 1 d, 2 d, 6 d). (D) Scheme of Alexa488-labeled TDP-43 aggregation for confocal imaging; created with BioRender.com. (E) Confocal images of Alexa488-labeled TDP-43 aggregates (10 μM) in presence of unlabeled Tau, MBP or α-synuclein (5, 10 or 20 μM) 2 h after the agitation step. Scale bar: 20 μm in overview and 15 μm in inset. (F) Quantification of the total area covered by the TDP-43 aggregates (Aggr. area). Bar graph shows the mean of ($n = 4$) biological replicates ± SD. ***$P = 0.0008$ and *$P = 0.0255$ by one-way ANOVA with Dunnett's multiple comparison test to TDP-43 only condition.

Tau molecules are embedded in a more crowded yet dynamic environment.

Taken together, TDP-43 promotes Tau condensation under both crowding and coacervation conditions, driving the formation of multiphasic co-condensates with internal Tau droplets with higher dynamics.

## Multiple regions in TDP-43 are involved in promoting Tau phase separation and multiphasic co-condensate formation

Next, we investigated which regions of TDP-43 are responsible for promoting Tau phase separation and formation of multiphasic Tau/TDP-43 co-condensates. To this end, we purified a series of recombinant TDP-43 deletion mutants: an N-terminal fragment (TDP-43-NTF-MBP, aa 1–266), a C-terminal fragment (TDP-43-CTF-MBP, aa 267–414), a fragment containing the tandem RNA recognition motifs (TDP-43-RRM, aa 102–270), a deletion mutant lacking both the folded N-terminal domain and the NLS region (TDP-43-Δ1-101-MBP), and a mutant lacking the α-helical conserved region (TDP-43-ΔCR-MBP) (Fig. EV3A).

We observed that TDP-43-CTF-MBP and TDP-43-RRM failed to promote Tau droplet formation, while all other TDP-43 variants promoted Tau condensation to varying degrees (Fig. EV3B). Notably, only full-length TDP-43-WT-MBP strongly induced the multiphasic phenotype with internal droplets, and TDP-43-NTF-MBP partially recapitulated this phenotype at the 24 h timepoint. Thus, several different regions of TDP-43, including both the N- and C-terminal fragments, are required for the formation of the multiphasic phenotype with internal Tau droplets.

To rule out potential artifacts from the fluorescent dyes or the crowding agent, we repeated the assays using unlabeled TDP-43 and examined each TDP-43 variant alone in the presence of 10% PEG. Unlabeled TDP-43 recapitulated the formation of multiphasic condensates with internal Tau droplets (Appendix Fig. S5), confirming that the observed effects are not dye-dependent. While some of the TDP-43 variants (Δ1-101, NTF, ΔCR) formed condensates in presence of 10% PEG, no internal TDP-43 substructures were observed for TDP-43 WT in absence of Tau (Appendix Fig. S5), indicating that not the crowding agent but presence of Tau drives the multiphasic co-condensate phenotype shown in Fig. 4B.

Together, our data indicate that multiple domains of TDP-43 cooperate to promote Tau phase separation, but only full-length TDP-43 can drive the formation of multiphasic Tau/TDP-43 co-condensates with internal Tau droplets, suggesting that multivalent

interactions across distinct TDP-43 regions are necessary for the formation of this condensation pattern.

## TDP-43 suppresses Tau fibril formation and generates small, oligomeric Tau and Tau/TDP-43 assemblies

Next, we investigated whether TDP-43 also affects Tau fibrillization in vitro. To address this, we employed the classical heparin-induced polymerization assay and monitored the formation of Tau paired helical filaments (PHFs) (Barghorn et al, 2005). We assessed aggregation either by the addition of Thioflavin T (ThT), a fluorescent dye that binds cross-β sheet structures and thereby detects amyloid fibril formation, or by negative staining transmission electron microscopy (TEM). For the ThT assay, we used a Tau FTD-mutant lacking the lysine at position 280 (Tau ΔK280), which is known to aggregate more rapidly (Barghorn et al, 2000). All aggregation experiments were performed in the presence or absence of equimolar concentrations of TDP-43-MBP, MBP, or α-synuclein (Fig. 5A).

The ThT incorporation assay clearly showed that TDP-43-MBP significantly slowed Tau ΔK280 fibrillization, compared to Tau ΔK280 alone or Tau ΔK280 in presence of MBP or α-synuclein (Fig. 5B, see Appendix Fig. S6A for raw data without background subtraction). This suggests that the Tau/TDP-43 assemblies have partial β-sheet character, consistent with early-stage or aberrant fibrillar structures.

To directly visualize Tau fibrilization, we examined Tau wild-type in presence or absence of TDP-43-MBP or control proteins after 5 days of incubation using TEM. Consistent with previous reports, Tau formed long fibrillar aggregates (Fig. 5C,D). Similar fibrils were observed in the presence of MBP or α-synuclein, but the addition of TDP-43-MBP resulted in the formation of much smaller structures with an average length <100 nm.

To determine whether TDP-43 suppresses Tau fibril formation or leads to the disassembly of already formed Tau PHFs, we examined earlier time points. In the absence of TDP-43-MBP, Tau fibrils were readily detectable after 1 day of incubation (Appendix Fig. S6B) and increased in number and length at day 2 and 3. In contrast, in the presence of TDP-43-MBP, no fibers were observed at any time point, instead small (<100 nm) structures emerged on day 2, which increased in number over time. These data suggest that full-length TDP-43 prevents the formation of Tau fibrils, rather than causing their disassembly.

To further elucidate whether the smaller structures were formed by Tau alone, TDP-43 alone, or by a combination of the two proteins, we performed immunogold labeling on samples

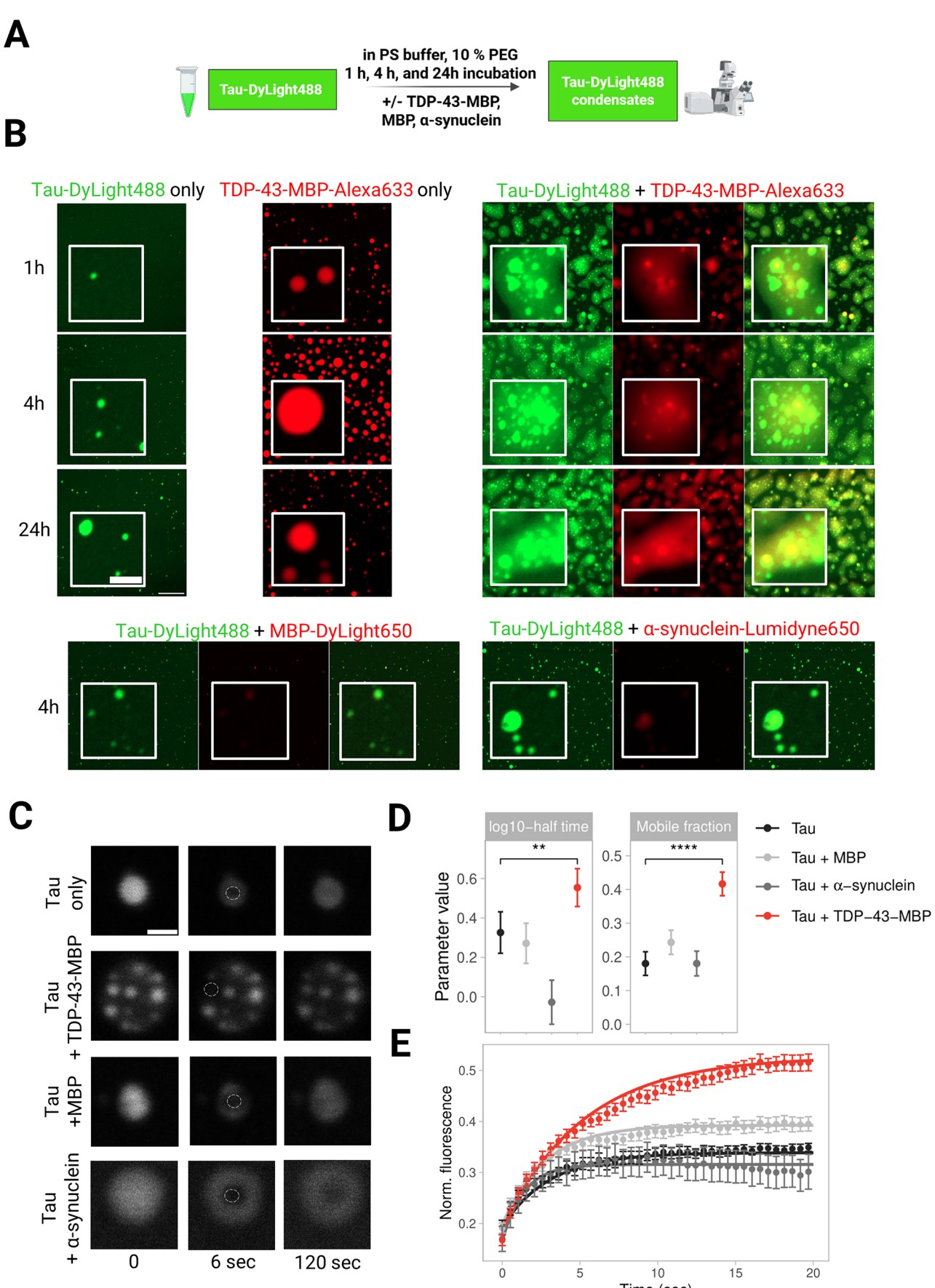

◀

**Figure 4.  TDP-43-MBP, but not MBP or α-synuclein, causes the formation of large, amorphous Tau condensates in vitro.**

(A) Schematic representation of Tau phase separation assay; created with BioRender.com. (B) Confocal microscopy images of 4 µM DyLight488-labeled Tau only, Alexa633-labeled TDP-43-MBP only, and DyLight488-labeled Tau in equimolar presence of Alexa633-labeled TDP-43-MBP, DyLight650-labeled MBP or Lumidyne650-labeled α-synuclein at the indicated timepoints (1, 4, 24 h). Scale bar: 20 µm in overview and 5 µm in inset. Images shown are representative images from one experiment out of 6 experimental replicates, all of which yielded similar results. (C) Representative FRAP images of DyLight488-labeled Tau condensates (15 µM) at the indicated time points, in the absence or presence of equimolar unlabeled TDP-43-MBP, MBP, or α-synuclein, within 1.5 h of 10% PEG addition. Dotted circle indicates the bleached area (point bleach). Scale bar: 1 µm. (D) Parameters of exponential recovery models fitted to FRAP curves of DyLight488-labeled Tau condensates in absence or presence of TDP-43-MBP, MBP, or α-synuclein. Error bars represent 95% confidence intervals based on 14–16 curves per condition, collected across four biological replicates. The center of error bars represents the estimated values. $**P = 0.001744$ and $****P = 2e\text{-}20$ by $t$ tests of regression parameters (Wald-type tests). (E) FRAP curves following point-bleaching of DyLight488-labeled Tau condensates in absence or presence of TDP-43-MBP, MBP, and α-synuclein. Points and error bars represent mean ± SEM from 14 to 16 individual condensates per condition, collected across four biological replicates. Solid lines show exponential recovery model fits used for statistical analysis in (D).

containing Tau only, TDP-43-MBP only, and Tau/TDP-43-MBP co-assemblies, using Tau- and TDP-43-specific antibodies. As expected, fibrils formed by Tau alone were positive for the Tau antibody (Tau mAB), whereas TDP-43-MBP structures were exclusively stained with the TDP-43 antibody (TDP-43 mAB) (Fig. 5E,F). Interestingly, the small Tau/TDP-43-MBP co-assemblies were recognized either by Tau mAB alone or by both Tau mAB and TDP-43 mAB, indicating formation of both Tau structures and Tau/TDP-43 co-assemblies (Fig. 5E,F, see Appendix Fig. S6C for negative control stainings). Together, our data show that TDP-43 suppresses Tau fibrillization in vitro and promotes the formation of small, non-fibrillar oligomeric assemblies composed of Tau with or without TDP-43, indicating a shift from Tau fibril formation to alternative assembly pathways.

## TDP-43 suppresses Tau fibril formation via its N-terminal half and suppresses Tau seeding

To investigate which region of TDP-43 is responsible for suppressing Tau fibril formation, we performed the in vitro Tau fibrillization experiment using the TDP-43-NTF-MBP (aa 1–266) and TDP-43-CTF-MBP (aa 267–414) fragments. As shown above, Tau incubated with full-length TDP-43-MBP did not form any fibrillar structures. Interestingly, addition of the TDP-43-NTF-MBP fragment also prevented Tau fibrillization and resulted in small assemblies with an average size below 100 nm, similar to those observed for Tau in the presence of full-length TDP-43 (Fig. 6A,B). In contrast, addition of TDP-43-CTF-MBP did not impair Tau fibrillization. Thus, full-length TDP-43 and the NTF, but not the CTF, inhibit Tau fibrillization.

One way to assess the potential pathogenicity of the formed assemblies is to test their seeding potency in the so-called Tau biosensor cells, a reporter cell line stably co-expressing two variants of the Tau repeat domain (RD) with the pathogenic P301S mutation fused to either CFP or YFP (HEK293 CFP/YFP-TauRD$^{P301S}$ (Holmes et al, 2014). In this assay, the lipotransfection of material containing aggregated or oligomeric Tau ( = Tau seeds) causes CFP/YFP-TauRD$^{P301S}$ aggregation in the cytoplasm of the reporter cells, detectable as a CFP-YFP FRET signal by fluorescence microscopy in the green channel ($\lambda = 488$ nm excitation; Fig. 6C). Notably, seeding of Tau aggregation in Tau biosensor cells is known to be highly specific for Tau and is not triggered by other protein seeds in the inoculum (Holmes et al, 2014). We transfected the different Tau assemblies shown in Figs. 5C and 6A into Tau biosensor cells and analyzed their effect on TauRD$^{P301S}$ aggregation. Consistent with published results, we observed several Tau-RD

aggregates in the cytoplasm upon transfection of Tau-only fibrils, and similar aggregate levels were observed upon transfection of Tau+MBP, Tau+α-synuclein, and Tau+TDP-43-CTF-MBP samples (Fig. 6D,E). Interestingly, Tau+TDP-43-MBP and Tau+TDP-43-NTF-MBP samples exhibited a strongly reduced seeding potency (Fig. 6D,E).

Taken together, TDP-43 suppresses Tau fibril formation and seeding via its N-terminal region, promoting the formation of small Tau assemblies and Tau/TDP-43 co-assemblies of low Tau seeding potency. Thus, TDP-43 co-pathology in AD patients might protect against Tau aggregation and Tau seeding, rather than promoting Tau pathology.

## Brains from AD patient with Tau/TDP-43 co-pathology exhibit lower Tau seeding capacity than AD patients with Tau-only pathology

To investigate the disease relevance of Tau/TDP-43 interaction, we extracted sarkosyl-insoluble material from the frontal cortex of patients diagnosed as AD− (containing Tau pathology, but lacking TDP-43 pathology), AD+ (Tau+TDP-43 pathology, consistent with limbic predominant age-related TDP-43 encephalopathy neuropathological changes (LATE-NC) (Nelson et al, 2019)), FTLD-Tau, FTLD-TDP Type A or non-neurodegeneration controls (non-ND), and from the cingulate cortex of PD patients (Table EV1), following the established SarkoSpin procedure (Laferrière et al, 2019). Sarkospin extracts from FTLD-TDP brains were shown to contain TDP-43 fibrils, however it is not known if they contain oligomers or other types of TDP-43 assemblies (Laferrière et al, 2019). Western blot analysis detected mainly TDP-43 but not Tau in the FTLD-TDP extracts, in contrast, mainly Tau but little TDP-43 was detected in the FTLD-Tau, AD− and AD+ extracts (Fig. EV4A–C). Interestingly, immunohistochemical (IHC) staining revealed the presence of both phosphorylated Tau (detected with the AT8 antibody), and cytosolically mislocalized TDP-43 in the frontal cortex and hippocampus of AD+ patients, but not in control brains (Fig. EV5A,B, for IHC images in the other patient groups see Appendix Fig. S7). Given that the limbic regions are primary sites of TDP-43 pathology in AD patients (Amador-Ortiz et al, 2007; Arai et al, 2009; Josephs et al, 2015; Latimer and Liachko, 2021; Meneses et al, 2021), we further performed double fluorescence immunostaining in the hippocampus tissue. This revealed numerous aggregates in the hippocampus of AD+ patients, some of which showed colocalization of these two pathological proteins within the same inclusions (Fig. EV5C), in line with a previous report (Tomé et al, 2021).

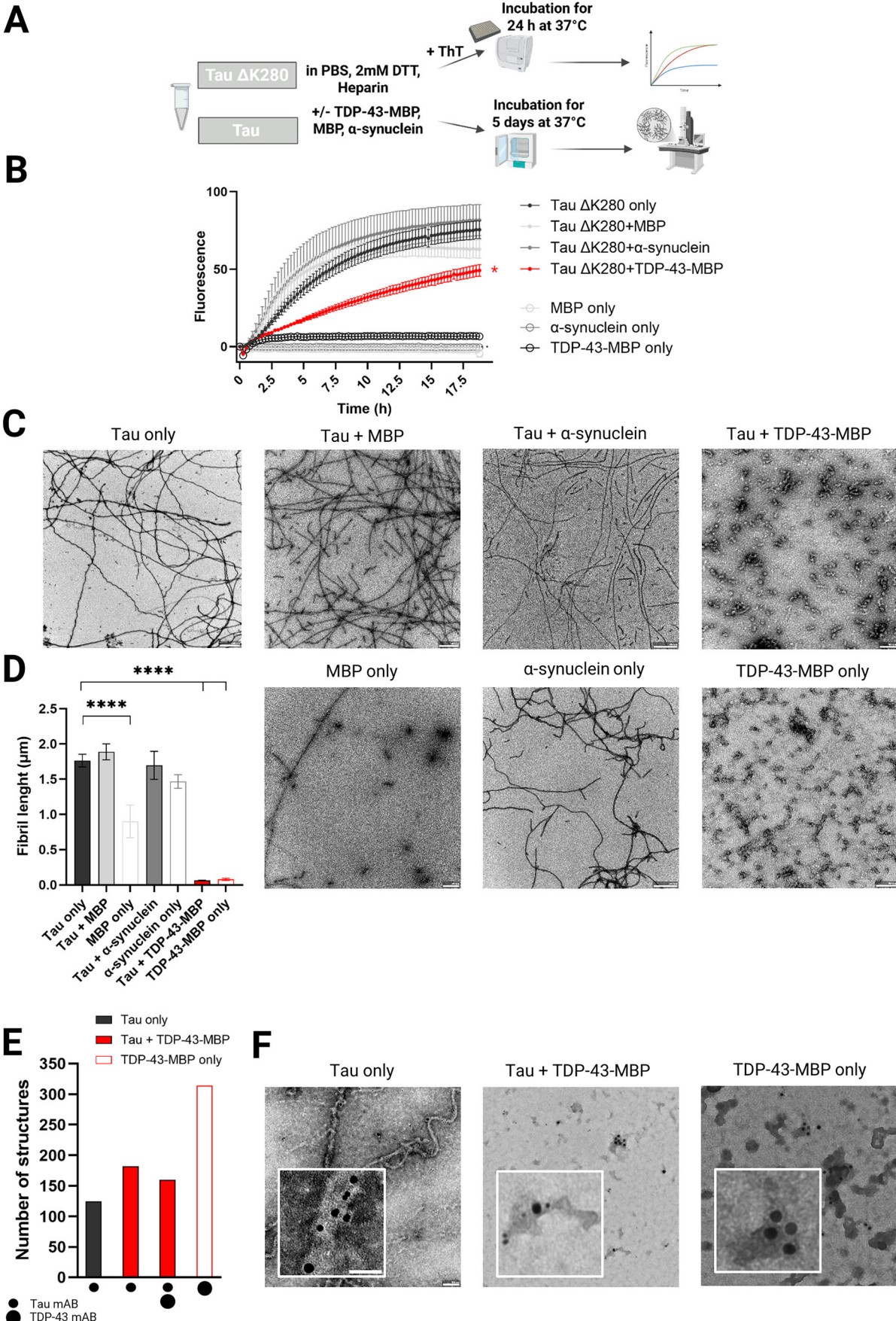

**Figure 5. TDP-43 inhibits Tau fibrillization and promotes formation of small, oligomeric Tau and Tau/TDP-43 assemblies.**

(A) Schemes of Thioflavin T (ThT) and Tau fibrillization assays; created with BioRender.com. (B) Graph showing normalized ThT fluorescence measurements over 18 h ± SEM in ($n > 3$) biological replicates. *$P = 0.0184$ was determined by one-way ANOVA with Dunnett's multiple comparison test, comparing each condition to Tau ΔK280 inly condition; significant difference was determined by calculating the area under the curve (AUC) for each condition. (C) TEM images of Tau, TDP-43-MBP, MBP or α-synuclein only (50 μM), and Tau in presence of TDP-43-MBP, MBP or α-synuclein, respectively, at equimolar ratio after 5 days incubation. Scale bars: 0.1 μm for Tau+TDP-43-MBP and TDP-43-MBP only, and 0.5 μm for Tau only, Tau+MBP, MBP only, Tau+α-synuclein and α-synuclein only. (D) Quantification of fibril length (μm) from ($n ≥ 2$) biological replicates and ($n ≥ 2$) technical replicates; values show the length of protein structures in μm and the bar graphs represent the mean ± SEM; ****$P < 0.0001$ by one-way ANOVA with Dunnett's multiple comparison test to Tau only. (E) Quantification of the number of structures positive for either Tau antibody (Tau mAB, small black circle), TDP-43 antibody (TDP-43 mAB, large black circle), or both, quantified for Tau only, Tau + TDP-43-MBP, and TDP-43-MBP only conditions, following double gold immunostaining. Bar graph depict final values obtained by subtracting the number of positive structures observed in control samples stained with secondary antibodies only (see Appendix Fig. S6). Values were analyzed in $n > 21$ technical experiments. Scale bar: 0.1 μm and 0.03 μm in inset. (F) Representative images of Tau only, Tau + TDP-43-MBP, and TDP-43-MBP only stained with both Tau mAB and TDP-43 mAB.

Upon transfection of these extracts into Tau biosensor cells (Fig. 7A), AD− brains induced the highest levels of Tau-RD aggregation after 3 days, compared to all other groups, and as expected we observed no/low seeding from non-ND controls, PD and FTLD-TDP patient samples (Fig. 7B,C; Appendix Fig. S8A). Interestingly, although AD+ extracts contained similar Tau levels (Fig. EV4A), they exhibited a lower seeding capacity than AD− extracts. Thus, it seems possible that in AD+ cases, Tau is (partially) incorporated into Tau/TDP-43 co-assemblies instead of forming pure Tau fibrils, which may impair seeding activity. This interpretation is consistent with data from our Tau seeding experiments with in vitro generated Tau/TDP-43 co-assemblies (Fig. 6).

Additionally, we tested the seeding potency of the same patient samples in a cellular TDP-43 seeding assay described in (De Rossi et al, 2021). In this assay, HEK293 cells with doxycycline-inducible TDP-43-HA expression are transfected with SarkoSpin extracts. Subsequent immunostaining for the HA-tag and phospho-TDP-43 is used to detect de novo formed TDP-43 "neoaggregates", which result from cytoplasmic mislocalization of TDP-43-HA and its subsequent aggregation and phosphorylation (De Rossi et al, 2021) (Fig. 7D).

Interestingly, in this TDP-43 seeding assay, we found that AD+ extracts had the highest seeding potency, surpassing the FTLD-TDP positive control (Fig. 7E,F; Appendix Fig. S8B). Although the difference between the aggregate area in AD+ and FTLD-TDP samples did not reach statistical significance ($P = 0.0568$), these data suggest that AD+ brains harbor highly TDP-43 seeding-competent species.

Taken together, brains from AD+ patients with Tau/TDP-43 co-pathology exhibit reduced Tau seeding capacity compared to Tau-only AD− brains. Surprisingly, the AD+ cases show even higher TDP-43 seeding capacity than FLTD-TDP cases. Thus, Tau/TDP-43 co-pathology in AD may suppress Tau propagation while promoting TDP-43-driven pathology.

## Discussion

Tau and TDP-43 co-pathology is a frequent and clinically relevant feature in AD, affecting approximately 60% of patients. Thus, understanding the molecular interplay between Tau and TDP-43 and how it affects condensate formation is crucial. Our study reveals a direct interaction between Tau and TDP-43 and that the two proteins significantly affect each other's condensation and

aggregation behavior in vitro, as well as the formation of proteopathic seeds in human brain, supporting the view that the two pathologies are interlinked and do not develop completely independently.

In detail, we observed that in the presence of Tau, TDP-43 forms irregularly shaped condensates with lower TDP-43 dynamics (Fig. 1), and that Tau promotes TDP-43 insolubility and aggregation (Fig. 3). Moreover, we found that the presence of full-length TDP-43 strongly promotes Tau phase separation into large, multiphasic co-condensates with higher Tau dynamics (Fig. 4). Specifically, we found that within 10 min, TDP-43 and Tau are unmixed within the condensates (Fig. EV2A); however, after 2 h, small Tau droplets become encapsulated within TDP-43 condensates, and persisted over 24 h (Fig. EV2B,C). This suggests that the small internal Tau condensates likely form by early de-mixing of Tau and TDP-43 within co-condensates, rather than Tau droplets being nucleated on the surface of TDP-43 condensates. Similar multiphasic co-condensates have been described for Tau and the prion protein (PrP) or Tau and TIA-1 in presence of RNA (Ash et al, 2021; Rai et al, 2023), as well as for α-synuclein and the TDP-43-CTD, where α-synuclein forms clusters on the surface of TDP-43-CTD-RNA condensates and nucleates the formation of heterotypic amyloid fibrils (Dhakal et al, 2023). Moreover, it has been proposed that the co-condensation of Tau and α-synuclein into highly crowded yet dynamic condensates creates an ideal reservoir for amyloid nucleation (Gracia et al, 2022). Based on these studies, it can be speculated that Tau/TDP-43 co-condensates may further develop into heterotypic Tau/TDP-43 co-aggregates, similar to those observed in AD+ patients (Fig. EV5C and Tomé et al, 2021). It should be noted that to analyze the impact of soluble TDP-43 on Tau condensate/fibril formation (Figs. 4 and 5), we had to retain the MBP tag, as removal of this solubility tag leads to immediate TDP-43 condensation above ~1 μM concentration. Thus, we were unable to study the impact of soluble, monomeric TDP-43 on Tau without the MBP tag. Given this limitation, validation studies with alternative tags or a solubility tag at the N-terminus of TDP-43 should be conducted in the future.

Experiments in cultured cells support the notion that condensation and demixing processes of Tau also occur in cells, and Tau condensation was proposed to have a physiological role in regulating microtubule dynamics (Hernández-Vega et al, 2017; Zhang et al, 2020). However, Tau droplets can quickly transition into gel-like structures that over time turn into amyloid-like aggregates with seeding capacities (Kanaan et al, 2020; Wegmann et al, 2018). The extent to which condensation processes contribute

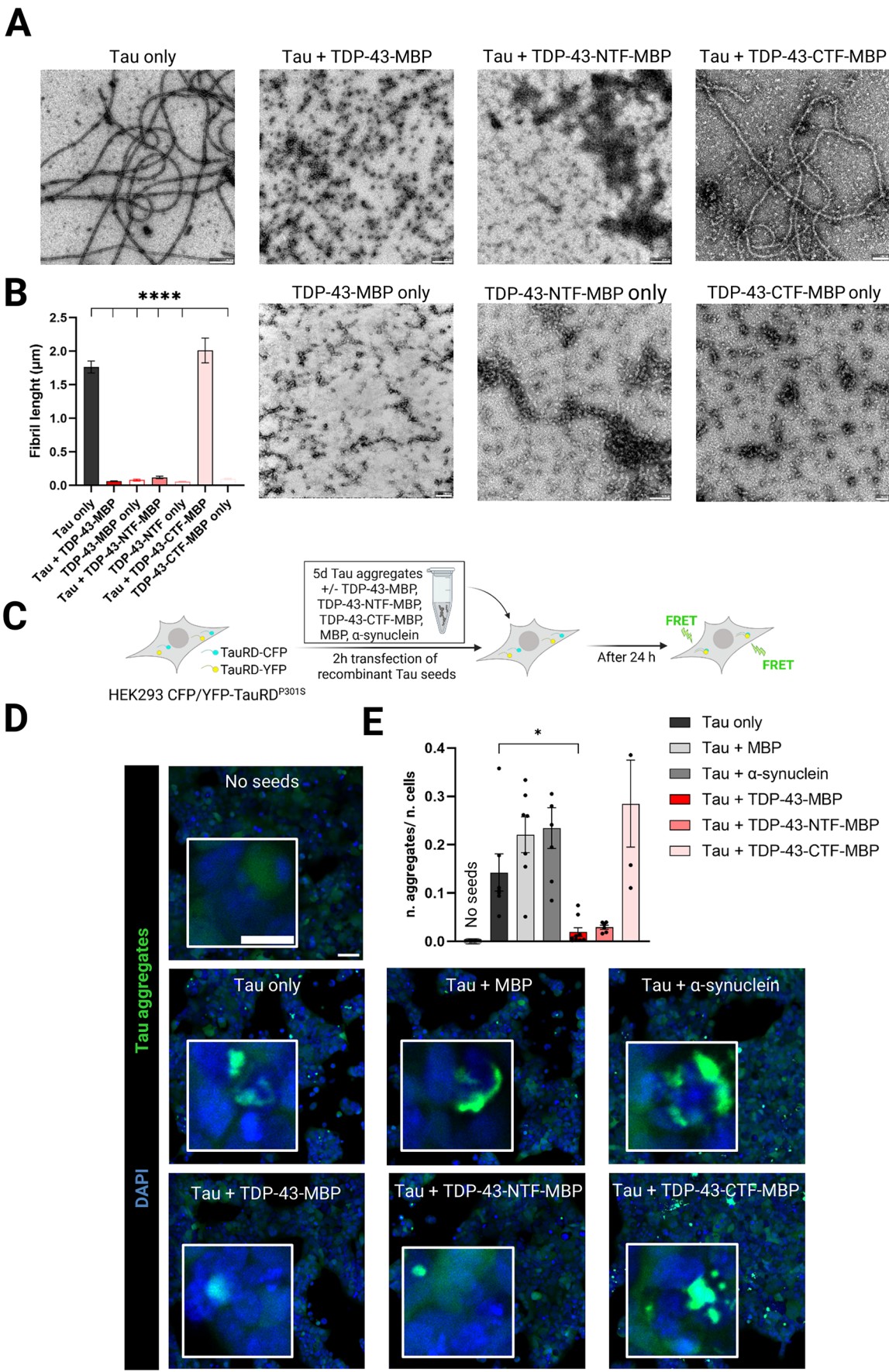

◀ **Figure 6. TDP-43 inhibits Tau fibrillization via its N-Terminal fragment and suppresses Tau seeding in a Tau biosensor cells.**

(A) TEM images of Tau, TDP-43-MBP, TDP-43-NTF-MBP and TDP-43-CTF-MBP only (50 μM), and Tau in presence of TDP-43-MBP, TDP-43-NTF-MBP and TDP-43-CTF-MBP at equimolar ratio after 5 days incubation. Scale bars: 0.1 μm for TDP-43-MBP only, TDP-43-NTF-MBP only, TDP-43-CTF-MBP only, and Tau+TDP-43-CTF-MBP; 0.2 μm for Tau+TDP-43-MBP and Tau+TDP-43-NTF-MBP; 0.5 μm for Tau only. (B) Quantification of fibril length (μm) from ($n \geq 2$) biological replicates and ($n \geq 5$) technical replicates; values show the length of protein structures in μm and the bar graphs represent the mean ± SEM; ****$P < 0.0001$ by one-way ANOVA with Dunnett′s multiple comparison test to Tau only. (C) Scheme of Tau seeding assay in HEK293 GFP/YFP-TauRD$^{P301S}$ using in vitro generated Tau assemblies as seeds; created with BioRender.com. (D) Representative confocal images of cytosolic Tau aggregates (green) formed after seeding Tau biosensor cells with 5 days old recombinant Tau aggregates +/− TDP-43-MBP, TDP-43-NTF-MBP, TDP-43-CTF-MBP, MBP and α-synuclein. The 'No seeds' condition refers to cells treated with lipofectamine only. Scale bar: 70 μm in overview and 20 μm in inset. (E) Quantification of number of aggregates per cell. Bar graphs show the number of biological replicates ± SEM in ($n = 5$) biological replicates (different batches of cells on different days). *$P = 0.0274$ by one-way ANOVA Kruskal–Wallis test with Dunn′s multiple comparison test to Tau only condition.

to aggregate formation in patient tissue remains a crucial question in the neurodegeneration field. Numerous studies have shown that droplet-like condensates can convert into solid and fibrillar structures by liquid-to-solid phase transitions (Hofweber et al, 2018; Molliex et al, 2015; Patel et al, 2015), and that amyloids can emerge at the surface of condensates (Emmanouilidis et al, 2024; Linsenmeier et al, 2023; Shen et al, 2023). Hence, a common view is that condensates could serve as precursors of amyloid-like aggregates in neurodegenerative diseases (Alberti and Dormann, 2019; Alberti and Hyman, 2016; Nedelsky and Taylor, 2019; Zbinden et al, 2020). However, recent studies support a protective role of condensates as protein sinks that suppress fibrilation (Das et al, 2025; Lipiński et al, 2022). Thus, further research is required to clarify the role of condensation processes in protein aggregation.

Notably, TDP-43 profoundly alters the Tau aggregation behavior. In our negative stain EM experiments, TDP-43 suppresses Tau fibrilization and instead promotes the formation of oligomeric species (<100 nm in size) that contain Tau alone or Tau/TDP-43 complexes (Fig. 5). Experiments in Tau biosensor cells, which are highly specific for detecting Tau seeds, but not other protein (Holmes et al, 2014), demonstrated that these assemblies show surprisingly low Tau seeding potency (Fig. 6). This observation challenges the prevailing notion that Tau oligomers are the most seeding-competent form of Tau (Berger et al, 2007; Lasagna-Reeves et al, 2011; Takeda et al, 2015; Usenovic et al, 2015). Structural and biophysical characterization of these assemblies will be essential to understand their seeding competence. It also will be important to further characterize the Tau/TDP-43 assemblies in human AD+ brain, as our seeding experiments with SarkoSpin brain extracts suggest that Tau might be driven into a less seeding-competent state by the presence of TDP-43 (Fig. 7) and thus may not be the main driver of pathology in these patients.

Given our results that TDP-43 suppresses Tau fibril formation in vitro (Fig. 5), we speculate that TDP-43 under these experimental conditions may bind to the protofilament core region of Tau (~306–378 aa) (Fitzpatrick et al, 2017; Oakley et al, 2020). Moreover, we found that the TDP-43 NTD interacts with Tau's PRD region (aa 174 and 225) in the condensed state (Fig. 2), possibly promoted by the high negative charge of the TDP-43 NTF ($z = −8.3$ at pH 7.4). To obtain a complete molecular picture, further interaction studies, e.g., experiments with additional truncation/mutant proteins or high resolution structural methods, such as nuclear magnetic resonance (NMR) spectroscopy, should be performed in the future. Since the PRD has been shown to modulate Tau aggregation kinetics and to drive phase separation of

Tau (Eidenmüller et al, 2001; Zhang et al, 2020), the interaction of TDP-43 with this region may drive Tau into an alternative conformation and may favor different assembly types, such as oligomers and condensates, which may still be toxic or promote pathology through alternative mechanisms.

Our finding that TDP-43 suppresses Tau fibril formation echoes reports that TDP-43 inhibits the early stages of amyloid-beta (Aβ) fibrilization (Shih et al, 2020). This activity also depends on the NTF (amino acids 1–256) and not the CTF of TDP-43, as does the suppression of Tau fibril formation in our experiments (Fig. 6). Interestingly, TDP-43 injection in AD mice not only inhibits Aβ fibrilization and increases Aβ oligomers, but also elicits memory deficits and inflammation (Shih et al, 2020). Our data suggest similar effects of TDP-43 on Tau: TDP-43 inhibits Tau fibrilization, yet at the same time causes formation of Tau and Tau/TDP-43 assemblies$^{P301S}$ with low Tau seeding potency, but high TDP-43 seeding activity (Figs. 6 and 7, graphical abstract). Studies in more sensitive neuronal seeding platforms (Rummens et al, 2025; Scialò et al, 2025) and in vivo models will be required to test the impact of the formed assemblies on the neurodegenerative phenotype.

An interesting question is where and under which conditions TDP-43 and Tau (in different assembly forms) may encounter each other in the neuronal cytoplasm, potentially influencing each other's condensation, aggregation and seeding behavior. Normally, TDP-43 is predominantly located in the nucleus, but small amounts are also found in the cytoplasm, especially under cellular stress conditions (Dewey et al, 2011; Gruijs da Silva et al, 2022), or after nucleocytoplasmic transport defects associated with neurodegenerative diseases and/or aging (Chou et al, 2018; D'Angelo et al, 2009; Hutten et al, 2020). Elevated cytosolic TDP-43 might trigger condensation or somatodendritic mislocalization of Tau. Upon cellular stress, cytosolic TDP-43 is recruited into stress granules (SGs) (Bentmann et al, 2012; Dewey et al, 2011), membrane-less RNP granules that sequester many RNA-binding proteins, including the aggregation-prone, disease-linked proteins FUS, hnRNP-A1/A2, TIA-1, but also Tau (Ash et al, 2021; Vanderweyde et al, 2012; Wolozin and Ivanov, 2019). Hence, under conditions of cellular stress, TDP-43 and Tau might encounter each other in SGs and undergo co-condensation and subsequent aggregation within these compartments.

Using two HEK293 reporter cell lines previously established to investigate TDP-43 and Tau seeding (De Rossi et al, 2021; Holmes et al, 2014), we found that seeding-competent Tau and TDP-43 species are present in brains of AD+ patients (Fig. 7), even without TDP-43/phosphoTDP-43 being detectable by Western blot (Fig. EV4). Nonetheless, co-aggregates were detectable by both

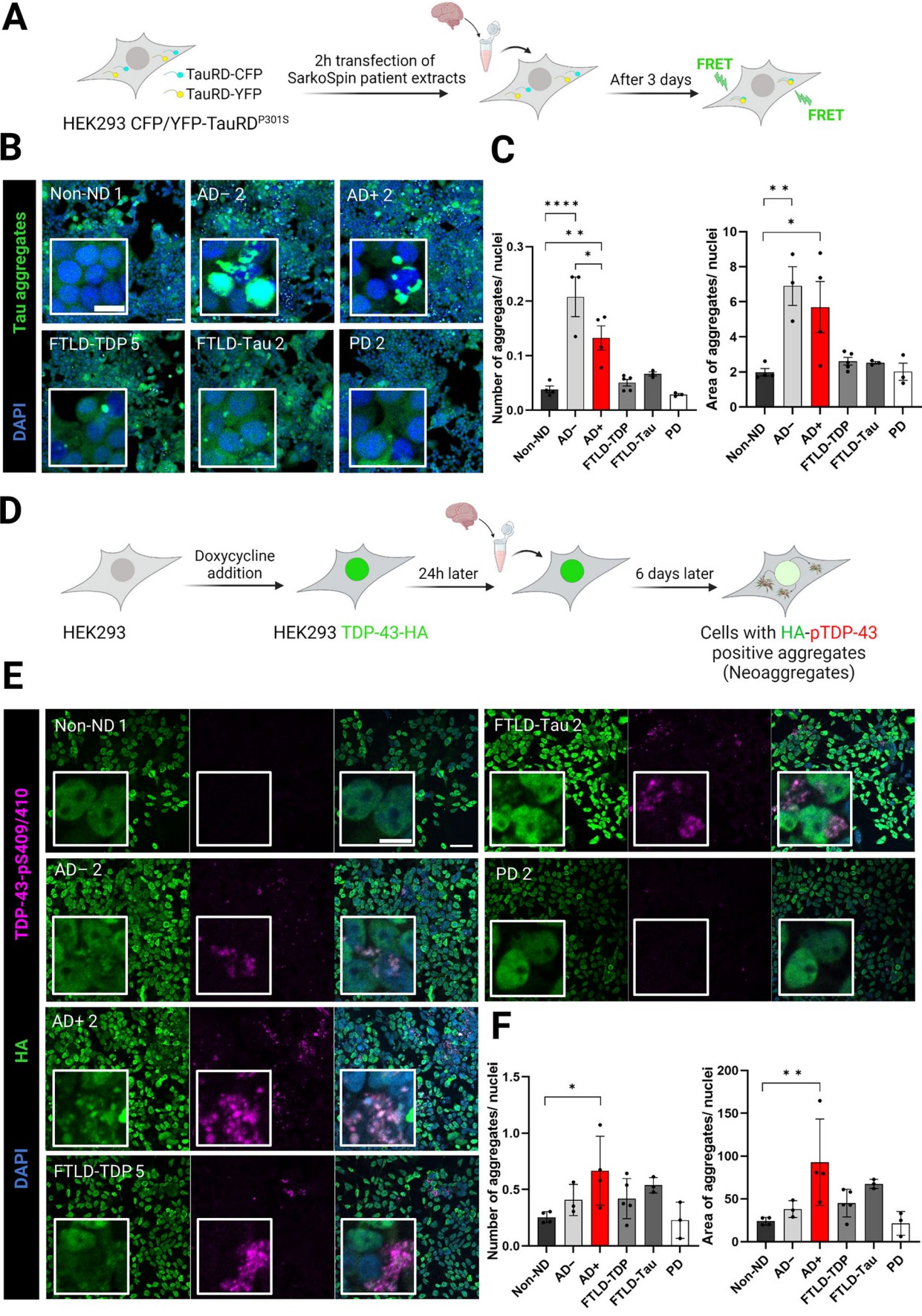

**Figure 7. Sarkospin extracts from AD patients with Tau/TDP-43 co-pathology (AD+) show a lower Tau seeding potency compared to AD patients with Tau-only pathology (AD−) and contain TDP-43 seeding competent species.**

(A) Scheme of Tau seeding assay in HEK293 GFP/YFP-TauRD$^{P301S}$ using SarkoSpin patient extracts as seeds; created with BioRender.com. (B) Representative confocal images of cytosolic Tau aggregates (green) formed after seeding the cells with non-ND 1, FTLD-Tau 2, AD − 2, AD + 2, FTLD-TDP 5, or PD 2 SarkoSpin extracts. Scale bar: 70 μm in overview and 15 μm in inset. (C) Quantification of number and area of aggregates per cell. Bar graphs show values ± SEM in (n = 4) biological replicates (different batches of cells in different days), and individual dots correspond to different patients. ****$P < 0.0001$, **$P = 0.0030$, and *$P = 0.0317$ (number of aggregates), and ****$P < 0.0001$, **$P = 0.0033$, and *$P = 0.0173$ (area of aggregates) by one-way ANOVA with Šídák´s multiple comparison test to non-ND. (D) Scheme of TDP-43 seeding assay using doxycycline-inducible TDP-43-HA HEK293 cells; created with BioRender.com. (E) Representative confocal images of cytosolic pS409/410-positive TDP-43 neoaggregates in HEK293 cells, after transfection with non-ND 1, FTLD-TDP 5, AD + 2, AD − 2, FTLD-Tau 2, or PD 2 SarkoSpin patient extracts. DAPI is depicted in blue, HA staining in green, and pS409/410 in magenta. Scale bar: 50 μm in overview and 10 μm in inset. (F) Quantification of HA and pS409/410-TDP-43 colocalization as number and area of neoaggregates per cell. Bar graphs show values ± SEM in (n = 4) biological replicates (different batches of cells in different days), and individual dots correspond to different patients. **$P = 0.0043$ and *$P = 0.0207$ by one-way ANOVA with Dunnett´s multiple comparison test to non-ND.

immunohistochemistry and dual fluorescent staining of AD+ tissue sections (Fig. EV5). Interestingly, AD+ extracts showed a tendency for lower Tau seeding capacity than AD− (Tau-only) samples, which aligns with our in vitro seeding assays showing that Tau/TDP-43 assemblies are seeding-incompetent (Fig. 6). However, our findings with brain extracts on Tau seeding contrast a recent study by Tomé and colleagues who showed the opposite behavior (Tomé et al, 2023). This discrepancy may be attributed to the inherent complexity and high heterogeneity of patient-derived brain samples, e.g. due to variability between different donors, as noted in previous studies employing the SarkoSpin extraction method (De Rossi et al, 2021; Laferrière et al, 2019; Scialò et al, 2025). Nevertheless, our results that AD+ brains contain highly TDP-43 seeding competent species is in line with an in vivo study in a TDP-43 transgenic mouse model reporting loss of physiological nuclear TDP-43 in mice injected with AD+ brain extracts (Tomé et al, 2023). Even though the molecular nature of the patient brain-derived TDP-43 "seeds" remains to be determined, our data support the idea that TDP-43 seeding and TDP-43 nuclear loss of function may contribute to the disease course in AD+ patients and may be one factor contributing to neurodegeneration in these patients.

In summary, our data provide molecular insights into the interaction of Tau and TDP-43, which is the basis of Tau/TDP-43 co-pathology common in animal models and AD patients (Chornenkyy et al, 2019; Latimer and Liachko, 2021; Montalbano et al, 2020; Spires-Jones et al, 2017; Tomé et al, 2023). Based on our data it seems possible that full-length TDP-43, once in the cytoplasm, interacts with Tau and suppresses Tau fibril formation. At the same time, this interaction promotes Tau/TDP-43 co-condensation, which may initially be protective by sequestering Tau and TDP-43 molecules and thereby reducing their availability for oligomer or fibril formation, and may even dampen Tau seeding. However, over time Tau/TDP-43 co-condensates may undergo aberrant phase transitions into solid co-aggregates (Hofweber et al, 2018; Molliex et al, 2015; Patel et al, 2015). Alternatively, their surface might nucleate amyloid fibers, as shown for other condensate systems (Emmanouilidis et al, 2024; Linsenmeier et al, 2023; Shen et al, 2023), providing a possible mechanistic explanation for how Tau/TDP-43 co-aggregates in brains of AD+ patients may form. Together, our data support a model in which Tau/TDP-43 co-condensation initially mitigates Tau pathology but may ultimately promote TDP-43-driven neurodegeneration.

## Methods

### Reagents and tools table

| Reagent/resource | Reference or source | Identifier or catalog number |
|---|---|---|
| **Experimental models** | | |
| HEK293 CFP/YFP-TauRD$^{P301S}$ cells | Holmes et al, 2014 | N/A |
| HEK293T Flp-In-T-REx cells | Laferrière et al, 2019 | N/A |
| *E. coli* BL21-DE3 Rosetta-LysS | Novagen (Merck) | 70956-3 |
| *E. coli* BL21 (DE3) | Novagen (Merck) | 70235-3 |
| *E. coli* BL21 Star* | D. Niessing (Ulm, Germany) | N/A |
| *E. coli* BL21 Star (DE3) | Thermo Scientific | N/A |
| **Recombinant DNA** | | |
| pJ4M TDP-43-TEV-MBP-His$_6$ | Addgene | 104480 |
| pNG2 vector | Barghorn et al, 2005 | N/A |
| pET5a/aSynuclein (136 TAT) | Philipp Kahle, Matthias Habeck | N/A |
| pT7-7 α-synuclein N122C | Addgene | 36046 |
| **Antibodies** | | |
| Rabbit anti-TDP-43 N-terminal | Proteintech | 10782-2-AP |
| Rabbit anti-TDP-43 C-terminal | Proteintech | 12892-1-AP |
| Mouse anti-Tau, 15-25 | BioLegend | 835204 |
| Mouse anti-HA | Sigma | 9658 |
| Rabbit anti-phospho TDP-43 (Ser409/410) | Proteintech | 80007-1-RR |
| Mouse anti-phospho TDP-43 (Ser409/410) | CosmoBio | TIP-PTD-M01 |
| Mouse anti-phospho Tau (AT8) | Thermo Scientific | MN1020 |
| Rat anti-α-synuclein | BD Bioscience | 610787 |
| Rabbit anti-α-synuclein | Abcam | MJFR1 |
| Biotinylated anti-Rabbit IgG | DAKO | N/A |
| Biotinylated anti-Mouse IgG | DAKO | N/A |
| Peroxidase conjugated goat anti-mouse | Jackson | AB_2338503 |
| Donkey-anti-Mouse IgG (H&L), 6 nm | Aurion | SKU: 806.322 |

| Reagent/resource | Reference or source | Identifier or catalog number |
|---|---|---|
| Goat-anti-Rabbit IgG (H&L), 15 nm | Aurion | SKU: 815.011 |
| **Oligonucleotides and other sequence-based reagents** | | |
| RNA-U20 | Sigma | N/A |
| **Chemicals, enzymes and other reagents** | | |
| DMEM high glucose GlutaMAX Supplement | Thermo Scientific | 61965026 |
| Opti-MEM GlutaMAX Supplement | Thermo Scientific | 51985026 |
| FBS Tet system approved | Thermo Scientific | A47364-01 |
| Lipofectamine 2000 | Thermo Scientific | 52887 |
| Hoechst 33342 | Thermo Scientific | H3570 |
| Doxycycline | Sigma | D9891 |
| Palbociclib | MedChemExpress | HY-50767 |
| Ni-NTA agarose beads | Qiagen | 30210 |
| Ceramic beads | Precellys | P000918LYSK0-A |
| MWCO 30 Amicon ultra centrifugal filters | Merck | UFC5030 |
| MWCO 10 Amicon ultra centrifugal filters | Merck | UFC5010 |
| Zeba Spin Desalting Columns, 7 K MWCO | Thermo Scientific | 89890 |
| Pierce Protein Concentrators PES, 30 K MWCO, 5-20 ml | Thermo Scientific | 88502 |
| Pierce Protein Concentrators PES, 30 K MWCO, 5-20 ml | Thermo Scientific | 88513 |
| Isopropyl β-D-Thiogalactopyranoside (IPTG) | Peqlab, Erlangen, Germany | N/A |
| Alexa Fluor 488 $C_5$ Maleimide | Thermo Scientific | A10254 |
| Alexa Fluor 633 $C_5$ Maleimide | Thermo Scientific | A20342 |
| DyLight 488 NHS Ester | Thermo Scientific | 46402 |
| DyLight 650 NHS Ester | Thermo Scientific | 62266 |
| LD650-MAL | Lumidyne | N/A |
| Recombinant GSK3ß kinase | BPS Bioscience | BPS-40007 |
| Pur-A-Lyzer Mini Dialysis tubes | Sigma | PURN12050-1KT |
| Di-succimidylsuberate (DSS-d0 and DSS-d12) | Creative Molecules Inc. | N/A |
| Pierce Protease Inhibitor Tablets EDTA-free | Thermo Scientific | A32965 |
| Complete EDTA-free protease inhibitors | Roche | 11836170001 |
| PhosSTOP Phosphatase inhibitors | Roche | 4906845001 |
| Benzonase | Millipore | 71205-3 |
| N-lauroyl-sarcosine (sarkosyl) | Sigma | 61739 |
| Phosphate-buffered saline | Gibco | 10010015 |
| Polyethylene glycol 8000 (PEG) | Roche | 0263.1 |
| Heparin Sodium Salt BioChemica | AppliChem | A3004,0005 |

| Reagent/resource | Reference or source | Identifier or catalog number |
|---|---|---|
| Thioflavin T | Millipore | 596200 |
| Uranyl Acetate | Science Services | E22400 |
| Protein LoBind Tubes, PCR clean, VE 100 | neoLab | 0030108 |
| BCA-c (10%) | Aurion | 900.022 |
| ProLong Diamond Antifade reagent | Thermo Scientific | 36961 |
| Pierce BCA Protein Assay Kits | Thermo Scientific | 23225 |
| 10X Bolt Sample Reducing Agent | Thermo Scientific | B0009 |
| EveryBlot Blocking Buffer | Bio-Rad | 12010020 |
| **Software** | | |
| GraphPad Prism 10.5 | https://www.graphpad.com | |
| ImageJ | https://imagej.nih.gov/ij/index.html | |
| MaxQuant 2.4.13 embedded with the Andromeda search engine | Cox and Mann, 2008; Yılmaz et al, 2022 | |
| XiView | Combe et al, 2024 | |
| Python 3.10.14 | Python Software Foundation | |
| BioIO package | Maxfield Brown et al, 2021 | |
| Scikit-image | "scikit-image: image processing in Python [PeerJ]," 2014 | |
| CellPose 3 | Stringer and Pachitariu, 2025 | |
| **Other** | | |
| μ-Slide 18 Well-Flat ibiTreat chambers | Ibidi | 81826 |
| μ-Slide 8 Well ibiTreat chambers | Ibidi | 80826 |
| TEM Grids Formvar-Karbon beschichtet 200 mesh Cu | Science Services | EFCF200-Cu |
| Formvar/carbon-coated nickel grids | Plano | S162 |
| 384-well μClear plates | Greiner | 781906 |
| Size Exclusion Chromatography (SEC) Hiload 16/600 Superdex 200 pg, | GE Healthcare | |
| Superdex 75 10/300 GL | GE Healthcare | |
| Size Exclusion column, Superose 6 10/300 | GE Healthcare | |
| Anion exchange column (His,Trap Q HP) | GE Healthcare | GE17-5156-01 |
| Cation-exchange chromatography column (His,Trap HP) | GE Healthcare | GE17-1152-01 |
| Orbitrap QExactive+ MS coupled to a Dionex Ultimate 3000 RSLCnano System liquid chromatography system | Thermo Scientific | |

| Reagent/resource | Reference or source | Identifier or catalog number |
|---|---|---|
| μ-Precolumn C18 PepMap100, C18 | Thermo Scientific | |
| Q2000 Sonicator | Qsonica | Q2000 |
| Branson Sonifier 250 | Gemini BV | 01886 |
| Homogenizer | Precellys | P000062-PEVO0-A |
| Cytation 3 | Biotek | |
| Basic plasma cleaner | Harrick Plasma | PDC-32G-2 |
| iBlot 2 Gel Transfer Device | Thermo Scientific | IB21001 |
| Fusion FX6 imager | Vilber | |

## Methods and protocols

### cDNA constructs

*Bacterial expressing constructs.* The plasmid pJ4M TDP-43-TEV-MBP-His$_6$ (addgene #104480) was used to generate human TDP-43 tagged with TEV-MBP-His$_6$ (TDP-43-MBP) as described in (Wang et al, 2018); from the same backbone, we cloned TEV-MBP-His$_6$ (MBP), TDP-43-N-terminal fragment (aa 1–266)-TEV-MBP-His$_6$ (TDP-43-NTF-MBP), His$_6$-TDP-43-RRM (aa 102–270) (TDP-43-RRM), TDP-43-Δ1-101 (aa 102–414)-TEV-MBP-His$_6$ (TDP-43-Δ1-101-MBP), and TDP-43-ΔCR (Δaa 321–340)-TEV-MBP-His$_6$ (TDP-43-ΔCR-MBP), as described in (Conicella et al, 2016, 2020).

The pET15-His-FKBP-L20-3C-TDP-43 LCD-TEV-MBP backbone (gift from Philipp Schönberger, IMB) was used to generate TDP-43-C-terminal fragment (aa 267–414)-TEV-MBP-His$_6$ (TDP-43-CTF-MBP).

Human full-length Tau (2N4R isoform), full-length FTD-mutant Tau ΔK280 (Lys280 deletion in 2N4R Tau), the Tau N-terminal domain (aa 1–256 of 2N4R Tau, Tau-NTD), and Tau repeat domain (aa 244-372 of 2N4R Tau: Tau-RD, alias K18) were introduced into a pNG2 vector, as previously described in (Barghorn et al, 2005).

Finally, the plasmid pET5a/aSynuclein (136 TAT), originally developed by Philipp Kahle at LMU Munich and modified with a 136TAC/TAT mutation by Matthias Habeck, was used to purify recombinant α-synuclein (unlabeled protein in the experiments). The pT7-7 α-synuclein N122C plasmid (original wild-type plasmid by Hilal Lashuel, Addgene #36046) was used to purify recombinant α-synuclein (labeled protein in the experiments).

### Recombinant protein expression and purification

*TDP-43-MBP and TDP-43 variants.* TDP-43-MBP, TDP-43-NTF-MBP, TDP-43-Δ1-101-MBP, and TDP-43-ΔCR-MBP were purified as previously described for TDP-43-MBP in (Gruijs da Silva et al, 2022). In brief, the expression of proteins was performed in *E. coli* Rosetta 2 using 0.5 mM isopropyl-beta-thiogalactoside (IPTG) overnight (o/n) at 16 °C. Cells were resuspended in lysis buffer (50 mM Tris pH 8.0, 1 M NaCl, 10 mM imidazole, 10% (v/v) glycerol, 4 mM β-mercaptoethanol and 1 μg/ml of each aprotinin, leupeptin hemisulfate and pepstatin A) supplemented with 100 μg/ml RNase A and 100 μg/ml lysozyme followed by sonication. The protein was then purified by Ni-NTA agarose beads (Qiagen) and eluted with lysis buffer containing 300 mM imidazole. Next, a purification step with size exclusion chromatography (SEC; Hiload 16/600 Superdex 200 pg, GE Healthcare) was carried out in storage

buffer (50 mM Tris pH 8, 300 mM NaCl, 5% (v/v) glycerol supplemented with 2 mM TCEP), in order to separate soluble TDP-43-MBP, TDP-43-NTF-MBP, TDP-43-Δ1-101-MBP, and TDP-43-ΔCR-MBP from protein aggregates and contaminants. The non-oligomeric fractions for TDP-43-MBP, TDP-43-NTF-MBP, TDP-43-Δ1-101-MBP, and TDP-43-ΔCR-MBP were pooled together and protein was concentrated using MWCO 30 Amicon ultra centrifugal filters (Merck Millipore) and then flash frozen and stored at −80 °C. Protein concentration was determined by measuring absorbance at 280 nm, using the molecular weight (kDa) and extinction coefficient (ε) predicted by the ProtParam tool. The A260/280 ratio for recombinant TDP-43-MBP consistently remained ≤ 0.6, indicating minimal nucleic acid contamination.

Similarly, TDP-43-RRM was expressed in *E. coli* BL21-DE3 using 0.5 mM IPTG o/n at 16 °C; the day after the cells were lysed in 20 mM HEPES pH 7.5, 1 M NaCl, 30 mM imidazole, 10% glycerol, freshly supplemented with 5 mM β-mercaptoethanol and 1 μg/mL Aprotinin/Pepstatin/Leupeptin each. The next steps followed the procedure described above.

TDP-43-CTF-MBP was expressed as His-FKBP-3C-TDP-43(267–414)-TEV-MBP in *E. coli* BL21-DE3 codon plus using 0.5 mM IPTG o/n at 16 °C. Cell pellet was lysed in IMAC buffer (30 mM Tris pH 8.0, 300 mM NaCl, 15 mM imidazole), supplemented with 2 mM MgCl$_2$, 0.5 mM TCEP, Sm nuclease and protease inhibitors, using a cell homogenizer. Subsequently, TX-100 (final 0.1%) was added to the lysate and the protein was purified using Ni-NTA agarose and TDP-43-CTF-MBP protein eluted by 1 mg/ml of His-3C protease in IMAC buffer containing 1 mM DTT. Finally, a purification step with Superdex200 16/600 was performed in storage buffer (20 mM HEPES pH 7.4, 150 mM NaCl, 10% Glycerol, 1 mM DTT). Protein containing fractions were pooled, concentrated using MWCO 10 Amicon ultra centrifugal filters (Merck Millipore) and then flash frozen and stored at −80 °C.

*MBP.* The MBP protein expression was performed in *E. coli* BL21 Star* and induced with 0.5 mM IPTG o/n at 20 °C. Next, cells were lysed in 1× PBS with 4 mM β-mercaptoethanol supplemented with 100 μg/ml lysozyme and sonicated. The protein was then purified by Ni-NTA agarose beads and eluted in 1× PBS, 4 mM β-mercaptoethanol, 1 M NaCl containing 300 mM imidazole. Finally, protein concentration was measured after dialysis into TDP-43-MBP storage buffer (20 mM Tris pH 8.0, 300 mM NaCl, 10% (v/v) glycerol supplemented with 2 mM TCEP).

*TEV protease.* TEV expression was performed similarly as the above proteins. The protein expression was induced in *E. coli* BL21-DE3 Rosetta-LysS with 1 mM IPTG o/n at 20 °C. Cells were lysed in 50 mM Tris pH 8, 200 mM NaCl, 20 mM imidazole, 10% (v/v) glycerol, 4 mM β-mercaptoethanol, supplemented with 0.1 mg/ml RNase A and 100 μg/ml lysozyme, followed by sonication. TEV was then purified using Ni-NTA agarose beads, and eluted with lysis buffer at pH 8.5 containing 1 M NaCl and 800 mM imidazole. Finally, TEV concentration was measured and the protein was dialyzed into storage buffer (50 mM Tris, 150 mM NaCl, 20% glycerol, 2 mM Dithiothreitol (DTT)).

*Tau.* Full-length Tau was expressed in *E. coli* BL21 (DE3) with a 2 h induction using 0.5 mM IPTG at 37 °C. Cells were then resuspended in lysis buffer (20 mM Na-MES pH 6.8, 1 mM EDTA,

proteinase inhibitors, 5 mM DTT) followed by lysis using a Constant Flow System. Then, 500 mM NaCl were added and the lysate was boiled at 95 °C for 20 min, followed by o/n dialysis at 4 °C first in 20 mM Na-MES pH 6.8, 50 mM NaCl, 1 mM EDTA (buffer A) and after 2 h in 20 mM Na-MES pH 6.8, 1 M NaCl, 1 mM EDTA (buffer B). Protein was then purified into a cation-exchange chromatography column (HisTrap, HP) and eluted with buffer B.

Next, a purification step with SEC was carried out in gel filtration buffer (1× PBS, 1 mM DTT). The purified monomeric Tau was concentrated and its concentration was measured as described for the other proteins above.

Expression of Tau ΔK280, Tau-NTD and Tau-RD, was carried out similarly, as described previously in (Barghorn et al, 2005; Hochmair et al, 2022). In brief, expression was induced in *E. coli* BL21 Star (DE3) (Thermo Scientific) with 0.5 mM IPTG at an optical density (OD) of 0.6 for ~3 h at 37 °C. For wild-type Tau, cells were harvested, resuspended in lysis buffer (20 mM MES, 1 mM EGTA, 0.2 $MgCl_2$, 1 mM PMSF, 5 mM DTT, protease inhibitors), and lysed using a French press. After initial purification by adding 500 mM NaCl and boiling at 95 °C for 20 min, cell debris was removed by centrifugation and the supernatant was dialyzed against buffer A (20 mM MED, 50 mM NaCl, 1 mM $MgCl_2$, 1 mM EGTA, 2 mM DTT, 0.1 mM PMSF, pH 6.8) o/n at 4 °C, then sterile filtered (0.22-μm membrane filter), run through a cation exchange column HiTrap SP HP, 5 mL (GE Healthcare), and eluted with a high salt buffer (B: 20 mM MES, 1 M NaCl, 1 mM $MgCl_2$ 1 mM EGTA, 2 mM DTT, 0.1 mM PMSF, pH 6.8). Fractions containing Tau were pooled, concentrated using spin column concentrations (Protein concentrators, 10-30 kDa MWCO, Pierce), and run through a size exclusion column Superose 6 10/300 (GE Healthcare). Fractions containing purified monomeric Tau were concentrated as before and buffer exchanged to PBS, 1 mM DTT, pH 7.4.

*α-synuclein.* The procedure for the expression and purification of unlabeled α-synuclein used in the experiments was carried out as previously described (Nuscher et al, 2004; Ruf et al, 2020, 2019). In brief, the plasmid pET-5a containing human wild-type alpha-Synuclein with a codon 136 TAC > TAT nucleotide exchange for improved bacterial expression was introduced into BL21(DE3) *E. coli* (New England Biolabs, Ipswich, MA, USA). Bacterial cultures were grown at 37 °C until an OD of ~0.5. Protein expression was subsequently induced by adding 1 mM IPTG (Peqlab, Erlangen, Germany) and maintained over 4 h at 37 °C. Cells were pelleted by centrifugation for 15 min at 4,000× *g* and pellets were resuspended in a total of 300 ml of 20 mM Tris, pH 8.0 with 25 mM NaCl. After heat inactivation of proteases and cell lysis by boiling the suspension in the microwave and subsequent incubation in a waterbath at 100 °C for 30 min, the lysate was cooled down on ice and centrifuged for 15 min at 17,000× *g*. The supernatant was subsequently filtered through a 0.20 μm syringe filter and the protein was extracted from the filtrated lysate via an anion exchange column (HiTrap Q HP, 5 ml; GE Healthcare, Chicago, IL, USA) and further purified via gel filtration chromatography (Superdex 75 10/300 GL; GE Healthcare, Chicago, IL, USA). The concentration of the protein was adjusted to 1 mg/mL, after which the aliquots were snap-frozen in liquid nitrogen and stored at −80 °C.

The preparation of labeled α-synuclein used in the experiments was conducted as previously described (Hoyer et al, 2002). Briefly, the pT7-7 α-synuclein N122C plasmid (original wild-type plasmid

by Hilal Lashuel, Addgene #36046) was transformed into BL21 Ai cells. The cells were grown in LB medium at 37 °C to an OD of 0.7, and expression was induced with IPTG and arabinose, followed by shaking o/n at 28 °C. After harvesting, the cells were lysed in lysis buffer (50 mM Tris, 10 mM EDTA, 1 mM PMSF) using three cycles through a continuous flow cell disruptor (Constant Systems Ltd.). The cell lysate was centrifuged at 20,000× *g* for 1 h. The supernatant was transferred to a new tube and incubated in boiling water for 25 min. The solution was centrifuged again at 20,000× *g* for 45 min. The resulting supernatant was supplemented with 10 mg/ml streptomycin sulfate and incubated for 15 min at 4 °C, followed by centrifugation at 20,000× *g* for 45 min. The supernatant was then treated with 361 mg/ml ammonium sulfate and incubated for 30 min at 4 °C. After centrifugation at 20,000× *g* for 45 min, the pellet was resuspended in 25 mM Tris, pH 7.7, and dialyzed against the same buffer twice o/n at 4 °C.

Different batches of independently purified proteins were used for all experiments.

### Labeling of recombinant proteins with fluorescent dyes

TDP-43-MBP, TDP-43-NTF-MBP, TDP-43-Δ1-101-MBP, and TDP-43-ΔCR-MBP were labeled with AlexaFluor-488 or AlexaFluor-633 $C_5$ maleimide (Thermo Fisher) following the manufacturer's instructions, at low labeling efficiency (0.05–0.1) to minimize the effect of the dye on the protein condensation and aggregation behavior. In brief, dye was added to the protein in storage buffer (20 mM Tris pH 8.0, 300 mM NaCl, 5% glycerol, 2 mM TCEP) with a protein:dye ratio of ~50:1, for 2 h at Room Temperature (RT) and protected from light.

Similarly, Tau, TDP-43-CTF-MBP, TDP-43-RRM, and MBP were labeled with DyLight-488 or DyLight-650 NHS Ester (Thermo Fisher) at low labeling efficiency (0.05–0.1) with a protein:dye ratio of ~50:1. Labeling was performed according to the manufacturer's instructions by incubating the proteins in the dark with the dye for 1 h at RT in 1× PBS, 2 mM DTT for Tau, and in the same TDP-43-MBP storage buffer for TDP-43 fragments and MBP.

pTau, Tau-NTD, and Tau-RD were fluorescently labeled using DyLight650-NHS ester (Thermo Scientific) following the manufacturer's instructions. The dye was dissolved in DMSO to a final concentration of 10 μg/μl, then added in a fivefold molar excess to the protein in PBS with 1 mM DTT for 2 h at RT at 250 rpm shaking. Excess dye was removed by dialysis with Pur-A-Lyzer Mini Dialysis tubes (Sigma) against PBS, pH 7.4, 1 mM DTT o/n at 4 °C. The labeling degree (amount of dye/molecule protein) was determined by measuring the final protein concentration and correlating it to the maximum absorbance of the conjugated dye.

α-synuclein was labeled as described for the above proteins using AlexaFluor-488 $C_5$ maleimide (Thermo Fisher) or Lumidyne650-MAL (Lumidyne) in PBS storage buffer.

In all cases, free dye was removed by repeated washes in protein-specific storage buffer using an Amicon ultra centrifugal columns (Merck Millipore).

### In vitro phosphorylation of Tau

Tau protein was incubated in phosphorylation buffer (25 mM HEPES, 100 mM NaCl, 5 mM $MgCl_2$, 2 mM EGTA, 1 mM DTT, protease Inhibitors (ROCHE, Complete) with recombinant GSK3ß kinase (BPS Bioscience, 5 mg/ml Tau, 0.5 mg/ml GSK3ß) and 1 mM ATP o/n at 30 °C. To denature and remove the kinase from the

sample, NaCl was added to a final concentration of 500 mM and the protein solution boiled for 10 min at 95 °C, and centrifuged at 100,000× *g* for 30 min. The phosphorylated Tau in the supernatant was dialyzed against phosphate buffered saline (PBS) containing 1 mM DTT.

### In vitro phase separation and aggregation assays

*Formation of Alexa488-labeled TDP-43 condensates.* Prior to each in vitro assay, TDP-43-MBP, TDP-43-NTF-MBP, TDP-43-CTF-MBP, TDP-43-RRM, TDP-43-Δ1-101-MBP, TDP-43-ΔCR-MBP, and MBP were centrifuged at 21,000 *g* for 10 min at 4 °C to discard any preformed protein precipitates.

For fusion event and fluorescence recovery after photobleaching (FRAP) assay, phase separation was performed by cleavage of either 5 µM or 15 µM, respectively, of Alexa488-labeled TDP-43-MBP with 100 µg/mL TEV protease at RT in PS buffer (20 mM HEPES pH 7.5, 150 mM NaCl, 1 mM DTT). Alexa488-labeled TDP-43 condensates were formed in the absence or presence of unlabeled or DyLight650-labeled Tau, MBP, and Lumidyne650-labeled α-synuclein at 0.5 to twofold concentration, and with Dylight650-labeled in vitro phosphorylated Tau (pTau)—previously phosphorylated with Glycogen Synthase Kinase-3 beta (GSK3β)—, Tau-NTD, and Tau-RD at equimolar concentration. Proteins mixtures were added to µ-Slide 18 Well-Flat ibiTreat chambers (Ibidi). Images were taken either ~30 min after TEV addition using a confocal microscope to visualize condensate formation, or 15 min after TEV addition with the spinning disc confocal microscope to capture time-lapse sequences over a 45 min period.

*Cross-Linking mass spectrometry (XL-MS) experiment.* For XL-MS analysis, three independent experiments were conducted to determine whether TDP-43 can physically interact with Tau and to characterize the topology of this interaction. Crosslinking events consistently identified in at least two of the three replicates were considered for downstream analysis. 10 µM TDP-43-MBP was cleaved with 0.08 mg/ml TEV protease in the presence of either 10 µM Tau or MBP in 50 µl of reaction buffer (20 mM HEPES, 150 mM NaCl, 1 mM DTT, pH 7.5). After 30 min from the induction of phase separation, samples were either analyzed (bulk condition) or subjected to 10 min centrifugation at 21,000× *g* to pellet the formed condensates and further resuspend them in 50 µl reaction buffer (pellet condition). Both bulk and pellet samples from TDP-43 incubated with MBP or Tau were subjected to crosslinking reaction with 1 mM isotope labeled di-succimidylsuberate (DSS-d0 and DSS-d12, Creative Molecules Inc.) (Czub et al, 2025; Leitner et al, 2014). After 5 min of reaction, the samples were quenched in liquid nitrogen and subsequently thawed in 50 mM Ammonium bicarbonate. Upon quenching, samples were dried, dissolved in 8 M Urea, reduced (5 mM TCEP), and alkylated (10 mM iodoacetamide). The proteolysis was performed o/n at 37 °C in 1 M urea with 1 µg trypsin (Promega, Sequencing grade). After quenching (5% formic acid), peptides were subjected to cleanup according to manufacturer's procedures (AttractSPE Tips, Affinisep).

Liquid chromatography-tandem mass spectrometry (LC-MS/MS) measurements were performed on Orbitrap QExactive+ MS coupled to a Dionex Ultimate 3000 RSLCnano System liquid chromatography system. Peptides were loaded on a commercial trap column (µ-Precolumn C18 PepMap100, C18, 300 µm I.D., 5 µm particle size) and separated using a reverse phase column Thermo PepMap100 C18 (50 cm length, 75 µm inner diameter, 35 µm particle size). For the identification of crosslinking peptides in DDA mode, an LC method was set up to separate peptides across a gradient from 8%B to 25%B in 60 min and from 25%B to 50%B in 4 min (buffer A: 0.1% (v/v) formic acid; buffer B: 0.1% (v/v) formic acid, 80% (v/v) acetonitrile). The acquisition method was performed with one MS1 scan, followed by a maximum of 20 scans for the top 20 most intense peptides (TOP20) with MS1 scans ($R = 70,000$ at 400 m/z, maxIT = 100 ms AGC=3e6), HCD fragmentation (NCE = 30%), isolation windows (1.6 *m/z*) and MS2 scans ($R = 17,500$ at 400 *m/z*, maxIT = 50 ms, AGC = 1e5). A dynamic exclusion of 30 s was applied, and charge states lower than three and higher than seven were rejected for the isolation.

*Formation of Alexa488-labeled TDP-43 aggregates.* The TDP-43 aggregation assay was performed by incubating 10 µM Alexa488-labeled TDP-43 with 100 µg/mL TEV protease at RT in aggregation buffer (50 mM Tris, pH 8.0; 250 mM NaCl; 5% glycerol; 5% sucrose; 150 mM imidazole, pH 8.0). The reactions were carried out in the absence or presence of Tau, MBP, and α-synuclein— proteins either unlabeled or labeled with DyLight650 or Lumidyne650—at concentrations ranging from 0.5- to twofold molar excess. Additionally, equimolar concentrations of DyLight650-labeled phosphorylated Tau (pTau), Tau-NTD, and Tau-RD were included in separate conditions.

Following setup, the protein mixtures were incubated on a thermomixer at 1000 rpm and 22 °C for 30 min, then transferred to a µ-Slide 18 Well-Flat ibiTreat chamber. Confocal microscopy images of the resulting aggregates were acquired approximately after 2 h.

*Semi-denaturing detergent agarose gel electrophoresis (SDD-AGE).* SDD-AGE experiments were performed according to the protocols described by (French et al, 2019) and (Halfmann and Lindquist, 2008). A total of 2 µM TDP-43-MBP was incubated either alone or in the presence of equimolar amounts of unlabeled Tau, MBP, or α-synuclein in aggregation buffer supplemented with 1× protease inhibitors (Sigma). Protein samples were collected at various time points (0, 1 h, 2 h, 8 h, 1 day (d), 2 d, and 6 d) following a 30 min incubation at 1000 rpm and 22 °C on a thermomixer.

For the experiment shown in Appendix Fig. S3, TEV protease cleavage was performed at RT for 30 min prior to shaking, and samples were collected at 0, 2 h, 5.5 h, 1 d, 2 d, and 3 d.

At each time point, 5 µL of the reaction mixture was diluted in SDD-AGE sample buffer (40 mM Tris-HCl, pH 6.8; 5% glycerol; 0.5% SDS; 0.1% bromophenol blue) and analyzed by horizontal electrophoresis on a 1.5% agarose gel (prepared in 20 mM Tris, 200 mM glycine, and 0.1% SDS). Gels were run in semi-denaturing running buffer (60 mM Tris, 20 mM acetate, 20 mM glycine, 1 mM EDTA, 0.1% SDS) for ~6 h at 60 V.

Following electrophoresis, proteins were transferred to a nitro-cellulose membrane according to the procedure described by (Halfmann and Lindquist, 2008), and subsequently detected via standard Western blotting (WB). Monomeric, oligomeric, and high-molecular-weight TDP-43 species were visualized using a rabbit anti-TDP-43 N-terminal antibody (10782-2-AP, Proteintech).

*Formation of DyLight488-labeled Tau condensates.* For fusion and FRAP assays, Tau phase separation was induced by adding either 4 µM or 15 µM DyLight488-labeled Tau, respectively, to

condensation buffer (20 mM HEPES, pH 7.5; 150 mM NaCl; 0.1 mM EDTA; 2 mM DTT) supplemented with 10% (w/v) polyethylene glycol (PEG). Alternatively, 20 µM DyLight488-labeled Tau was used in a low-salt buffer (25 mM HEPES, pH 7.5; 10 mM NaCl; 1 mM DTT) in the presence of 1.3 µM RNA-U20 (Sigma).

DyLight488-labeled Tau condensates were generated either alone or in the presence of equimolar amounts of unlabeled or Alexa633-labeled TDP-43-MBP, TDP-43-NTF-MBP, TDP-43-Δ1–101-MBP, TDP-43-ΔCR-MBP, DyLight650-labeled TDP-43-CTF-MBP, TDP-43-RRM, MBP, or α-synuclein.

Protein mixtures were then transferred into µ-Slide 18 Well-Flat ibiTreat chambers, and images were acquired at 10 min, 1 h, 2 h, 4 h, and 24 h after PEG addition, or ~1 h after RNA-U20 addition.

*Thioflavin T (ThT) assay.* For ThT assays, 10 µM Tau ΔK280, known to have an accelerated fibrillization rate, was incubated either alone or in the presence of equimolar concentrations of TDP-43-MBP, MBP, or α-synuclein. All reaction mixtures contained 12.5 µM Heparin (AppliChem) and 50 µM ThT (Millipore) in 1× PBS, 2 mM DTT. Subsequently, samples were transferred to 384-well µClear plates (Greiner) for measurement. The assay was conducted in a plate reader (Cytation 3, Biotek) at 37 °C for a total duration of 18 h. Fluorescence readings were taken every 15 min following a 5 s orbital shaking step, with ThT fluorescence measured at an excitation wavelength of 440 nm and an emission wavelength of 485 nm.

*Tau fibrils formation.* To prepare Tau fibrils in vitro, 50 µM unlabeled Tau was incubated alone or in the presence of equimolar concentrations of unlabeled TDP-43-MBP, MBP, α-synuclein, TDP-43-NTF-MBP, or TDP-43-CTF-MBP in buffer containing 1× PBS, 0.3% NaN$_3$, 2 mM DTT, and 12.5 µM heparin. The samples were incubated at 37 °C for 1, 2, or 5 days.

Formed fibrils were subsequently imaged by transmission electron microscopy (TEM) and analyzed using ImageJ/Fiji software, with fibril lengths quantified in micrometers (µm).

*Negative and immunogold staining.* Tau fibrils were diluted in distilled water to a final concentration of 1 µM. For transmission electron microscopic (TEM) observation, carbon coated copper grids (Science Services) were glow discharged for 30 s in a Harrick plasma cleaner (PDC-32G-2) to facilitate adsorption. After fixing the grid by anti-capillary inverse tweezers (Dumont), a sample volume of 1.5 µl was pipetted onto the grid for 2 min and blotted shortly using filter paper (Whatman). Negative staining was performed by addition of 1.5 µL 1% uranyl acetate (Science Services) in water for 30 s.

For immunogold staining, diluted fibrils were adsorbed onto glow-discharged, formvar/carbon-coated nickel grids (Plano) as described before. After washes with water, blocking buffer (0.1% acetylated BSA (BSA-C) (Aurion) in PBS) was applied for 10 min. Samples were incubated in primary antibodies (TDP-43 (Proteintech) 1:2500, Tau (BioLegend) 1:1000) in 0.01% blocking buffer for 60 min and washed in PBS. The first secondary antibody (DAM 6 nm (Aurion)) in 0.01% blocking buffer was applied for 60 min and then washed before applying the other secondary antibody (GAR 15 nm (Aurion)) in 0.01% blocking buffer. After washes in water the samples were stained using 1% uranyl acetate (Science Services) in water for 1 min. After blotting, the grid was air dried for at least 30 min.

## Cell culture assays

*Cell culture.* HEK293 CFP/YFP-TauRD$^{P301S}$ cells co-express two Tau-repeat domains (TauRD) carrying the frontotemporal dementia (FTD)-mutation P301S and fused to either the cyan or the yellow fluorescent proteins (CFP and YFP, respectively) (Holmes et al, 2014) provided by Marc Diamond. These were cultured in Opti-MEM GlutaMAX Supplement (Thermo Fisher), supplemented with 5% fetal bovine serum (FBS), and maintained in a humidified incubator at 37 °C with 5% CO$_2$.

HEK293T Flp-In-T-REx cells (provided by Magdalini Polymenidou, University of Zurich) that express nuclear TDP-43-HA under a doxycycline inducible promoter (Laferrière et al, 2019). These cells were grown in high-glucose DMEM with GlutaMAX Supplement (Thermo Fisher), supplemented with 5% Tet System Approved FBS (Thermo Fisher). Both cell lines were used from passage 2 and maintained to a maximum of passage 5.

*Cellular Tau seeding assay.* HEK293 CFP/YFP-TauRD$^{P301S}$ cells were used for the Tau seeding experiments. µ-Slide 8 Well ibiTreat chambers (Ibidi) were coated with 50 µg/ mL poly-D-lysine for at least 1 h, and subsequently $3 \times 10^4$ cells/well were seeded on these plates and grown in Opti-MEM GlutaMAX Supplement (Thermo Fisher), supplemented with 5% Fetal Bovine Serum (FBS) medium. The next day, 5 µg of either 5 day-old recombinant Tau aggregates formed in presence or absence of TDP-43-MBP, MBP, α-synuclein, TDP-43-NTF-MBP, or TDP-43-CTF-MBP, or SarkoSpin extracts obtained from FTLD-TDP-43 type A (FTLD-TDP), FTLD-Tau, AD (lacking TDP-43 pathology), AD+ (with TDP-43 pathology), PD, or non-ND patients, were added in OPTI-MEM and transfected into the cells with 0.8% lipofectamine 2000 (Invitrogen). After 2 h, cell medium was completely exchanged with fresh grow medium. For live cell confocal microscopy, intracellular aggregates were visualized after 1 or 3 days of cell transfection as bright accumulations by FRET signal (green channel, λ = 488 nm excitation), and nuclei were stained using Hoechst 33342 (1:2000, Invitrogen).

*Cellular TDP-43 seeding assay.* TDP-43 seeding assay was performed using HEK293T Flp-In-T-REx cells. Following coating plates with 50 µg/mL poly-D-lysine for at least 1 h, $5 \times 10^4$ cells/well were seeded and induced after ~5/6 h using 1 µg/mL of doxycycline (Sigma). On the next day, 14.2 µg of FTLD-TDP, FTLD-Tau, AD, AD+, PD, or non-ND patient extracts were incubated with OptiMEM and Lipofectamine2000 for 30 min and subsequently added to the cells. Medium was replaced with fresh medium supplemented with 7 µM Palbociclib (Sigma) and 1 µg/mL of doxycycline 3 h after transfection and once more on the following day, again with 7 µM Palbociclib and 1 µg/mL of doxycycline. Finally, cells were fixed and stained 6 days after transfection.

*Immunofluorescence staining.* After fixation of HEK-TDP-43-HA cells with 4% PFA for 10 min, cells were washed with 1× PBS supplemented with 2 mM MgCl$_2$ and 1 mM CaCl$_2$ (PBS+). Subsequently, cells were permeabilized for 5 min using 0.2% Triton X-100 in PBS. After several washes in PBS+ to remove detergent, cells were blocked with 2% BSA in PBS+ for 30 min at RT. Incubation with primary antibodies mouse anti-HA antibody (Sigma) and rabbit anti-phospho TDP-43 antibody (Proteintech) diluted in blocking buffer was carried out for 2-3 h at RT. After three washes with PBS+, cells were incubated with secondary antibodies

diluted in blocking buffer for 1 h at RT. Finally, nuclei were counterstained using DAPI at 0.5 µg/mL in PBS+ for 5 min, before coverslips were mounted with ProLong™ Diamond Antifade reagent (Invitrogen) on microscopy slides.

To display the images in Fig. 7E and Appendix Fig. S8B, a linear enhancement for brightness and contrast was applied.

### Patient samples

*Postmortem brain tissue.* Patient samples were obtained from Queen Square Brain Bank (QSBB) for Neurological Disorders at University of College of London (UCL)), and the Netherland Brain Bank (NBB) at the Netherlands Institute for Neuroscience, Amsterdam (Table EV1).

Informed consent was obtained from all subjects involved in this study, and all procedures were conducted in accordance with the principles outlined in the WMA Declaration of Helsinki and the Department of Health and Human Services Belmont Report.

*Homogenization of brain tissue.* Homogenization of patient brain tissue was performed as described in (De Rossi et al, 2021; Laferrière et al, 2019). Received brain samples were stored at −80 °C and transported on dry ice to avoid thawing of the tissue. For homogenization, the brain tissue was weighed and cut into very small pieces with a sterile razor and placed into a 50 ml Falcon tube. In all, 1× homogenization-solubilization buffer (10 mM Tris-HCI pH 7.4, 150 mM of NaCl, 0.5 mM of EDTA, 1 mM of DTT, complete EDTA-free protease inhibitors (Roche), PhosSTOP phosphatase inhibitors (Roche)) was added in a ratio of 5:1 to the amount of tissue for a final concentration of 20%. This coarse tissue resuspension was aliquoted into 2-mL tubes containing a mixture of ceramic beads with a diameter of 1.4 and 2.8 mm (Precellys). The samples were then homogenized with a Precellys homogenizer (P000062-PEVO0-A) at 5000 rpm in three 30 s rounds, while cooling the samples on ice between the rounds. After homogenization, aliquots of 150 µl were produced in protein low-binding tubes (Eppendorf). The aliquots were shock-frozen in dry ice and placed back into the −80 °C freezer.

*SarkoSpin preparation on brain homogenates.* SarkoSpin was performed as previously described (Laferrière et al, 2019). 50 µL of Benzonase mix, containing 14 mM of $MgCl_2$ and 250 U benzonase (Merck Millipore, 71205-3) in 1× HS buffer was added to 150 µl of brain homogenate. After 5 min incubation at RT, 200 µl of 4% N-lauroyl-sarcosine (sarkosyl) in 2× HS buffer (20 mM of Tris-HCI pH 7.4, 300 mM of NaCl, 1 mM of EDTA, 2 mM of DTT, complete EDTA-free protease inhibitors (Roche), PhosSTOP phosphatase inhibitors (Roche)) was added to each sample. For solubilization, the samples were put on a heating block (Thermomixer, Eppendorf) for 45 min at 38 °C at 600 rpm. Afterward, ice-cold 400 µl of 40% sucrose in 1× HS buffer was added per sample and was centrifuged at $21,200 \times g$ for 30 min at RT. For PD samples, after adding 40% sucrose, the samples were centrifuged at $250,000 \times g$ at 4 °C for 30 min. The supernatant was discarded, and the pellet was cleaned twice with 100 µl of phosphate-buffered saline (Gibco) to carefully remove lipids and sarkosyl from the pellet. Pellets for immediate use were resuspended in 100 µL of PBS by sonication (Qsonica, Q2000 with an amplitude of 60% power and 3 s on/3 s off for 3 min), or dry pellets were snap-frozen on dry ice and stored at −20 °C prior to use.

*Chemiluminescent western blot of patient samples.* The amount of total protein concentration was measured through Pierce BCA protein assay kit (ThermoFisher) for normalization. For chemiluminescent western blot of patient samples, 10% of the SarkoSpin pellet, normalized to input, was loaded per lane. All reagents used are from ThermoFisher unless differently specified. Protein samples were mixed in final 1× LDS loading buffer with final 1X Bolt sample reducing agent (Life Technologies), denatured at 70 °C for 10 min, and loaded onto 4–12% BisTris Plus Gels. Gels were transferred onto nitrocellulose membranes using iBlot 2 Transfer NC Stacks with iBlot 2 Gel Transfer Device at 7 min at 20 mV. Membranes were washed once in water and then blocked for 20 min at RT in EveryBlot blocking buffer (Bio-Rad). Primary antibodies for anti-TDP-43 pS409/410 (CosmoBio, 1:2500), anti-TDP-43 N-terminal and C-terminal cocktail (Proteintech, 1:2500), anti-synuclein (Abcam, 1:2500), and anti-Tau (BioLegend, 1:1000) were added to membrane in EveryBlot blocking buffer and incubated o/n at 4 °C. The membrane was then washed 3x in PBS-T, and then the secondary incubation with Peroxidase conjugated goat anti-mouse antibody (Jackson) was carried out for 3 h at RT, in EveryBlot blocking buffer. The membrane was then washed 3× in PBS-T before imaging on Fusion FX6 imager (Vilber).

*Immunohistochemistry (IHC) staining.* Eight-micron-thick formalin-fixed paraffin-embedded (FFPE) sections were cut from the frontal cortex and hippocampus. The sections were deparaffinized in xylene and rehydrated using graded alcohols. Endogenous peroxidase activity was blocked using 0.3% $H_2O_2$ in methanol for 10 min followed by pressure cooker pretreatment for 10 min in citrate buffer, pH 6.0. Non-specific binding was blocked using 10% dried milk/Tris buffered saline-Tween (TBS-T) before incubating with the primary antibodies: TDP-43 antibody (Proteintech, 1:1000 o/n at 4 °C); AT8 (Invitrogen, 1:100 1 h at RT) and α-synuclein (BD Bioscience, 1:100 1 h at RT). For α-synuclein immunohistochemistry an additional pretreatment of 10 min incubation in concentrated formic acid was carried out prior to the pressure cooking. A biotinylated anti-rabbit or anti-mouse IgG antibody (DAKO, 1:200) was incubated with the sections at RT for 30 min, followed by other 30 min of avidin-biotin complex (Vector Laboratories). The colour was developed with di-aminobenzidine activated with $H_2O_2$ (Lashley et al, 2011).

*Double immunofluorescence staining.* Sections were processed as detailed in the IHC staining method. Sections were incubated in both TDP-43 antibody (Proteintech, 1:1000) and AT8 (Invitrogen, 1:100) o/n at 4 °C. Sections were then washed and incubated in the Alexa Fluor 488 goat anti-mouse and Alexa Fluor 594 goat anti-rabbit.

### Microscopy
*Confocal laser scanning microscopy.*

a. Images of fluorescently labeled proteins were acquired on an inverted Zeiss LSM800 Axio Observer.Z1/7 and a LSM710 Axio Observer confocal laser scanning microscopes (Carl Zeiss) using Plan-Apochromat 63×/1.20 or 63×/1.40 Oil DIC M27 objectives. Images were captured with one- to four-line averaging and an image pixel size of 90–130 nm.

   To display the dual colors images in Appendix Fig. S1B,D, a linear enhancement for brightness and contrast was applied.

b. For the HEK cells experiments, imaging was performed using the same LSM800 microscope described above. A Plan-Apochromat

10×/0.3 objective with one-line averaging and an image pixel size of 312 nm was used for the Tau seeding assay, while a Plan-Apochromat 20×/0.8 M27 objective and two-line averaging and an image pixel size of 156 nm was used for the TDP-43 seeding assay.

To display the dual colors images in Fig. 7E and Appendix Fig. S8B, a linear enhancement for brightness and contrast was applied.

c. For high resolution imaging of Tau/TDP-43 condensates after 10 min and 2 h, respectively, confocal microscopy was performed with an inverted Zeiss Axio Observer 7 with an LSM 980 laser module equipped with lasers for 493 nm and 632 nm. Images for early timepoints were acquired either in SR-8Y (10 min timepoint) or in SR-4Y (2 h timepoint) mode, respectively, using sequential, bidirectional scanning without averaging.

For high resolution imaging of Tau/TDP-43 condensates after 24 h, confocal microscopy was performed with an inverted Zeiss Axio Observer 7 with an LSM 900 laser module equipped with lasers for 488 and 640 nm. Images were acquired using sequential, bidirectional scanning and twofold frame averaging.

50 (for images after 10 min and 2 h) or 20 (for 24 h timepoint) Z-stacks with 0.17 µm intervals were acquired with a C-Plan Apochromat 63×/1.4× objective at an image pixel size of 40 nm using the Airyscan 2 detector (380–735 nm for early timepoints up to 2 h, 490–700 nm for 24 h timepoint).

Subsequently, images were processed using the built-in Airyscan imaging processing plugin in the Zen software. Line profile analysis was performed using the ZEN software.

d. Brightfield images of immunohistochemically stained sections for TDP-43 and Tau were acquired using a Nikon Eclipse Ni-U upright microscope equipped with a Nikon DS-Ri2 high-resolution colour camera. A 40× Plan Fluor objective (NA 0.50) was used to capture representative fields of view. Images were acquired using NIS-Elements software (Nikon) under consistent exposure settings. For figure presentation, linear adjustments to brightness and contrast were applied uniformly across the entire image, without altering the original signal.

e. Images of double immunohistochemically stained sections (Tau and TDP-43) were acquired using a Leica LMD5500 microscope. Representative regions of interest from the frontal cortex and hippocampus were identified and imaged using a 40×. Leica Application Suite (LAS) software was used for image capture and preliminary processing. All images were processed uniformly with only linear adjustments to brightness and contrast applied for figure preparation.

The following excitation and emission wavelength fluorescence settings were used for detection: DAPI 353–465 nm, AF488 493–517 nm, AF555 553–568 nm, AF633 631–647, AF647 653–668, and Dy650 654–675.

*Spinning disc confocal microscopy.*

a. The TDP-43 PS time lapse experiments were performed using an inverted spinning disc microscope (Visiscope 5 Elements, Visitron Systems GmbH, Germany - IMB Microscopy Core Facility), built on a Nikon Ti2 stand equipped with a confocal spinning disc (CSU-W1; Yokogawa, Japan) and a 60x/1.2 NA water objective was used for acquisition.

b. FRAP experiments were conducted using a Stellaris spinning disk confocal microscope, with a 63×/1.4 oil objective (IMB Microscopy Core Facility).

For the TDP-43 FRAP assay, images were acquired within 15 min of adding TEV protease. Half of each condensate was bleached using a 488 nm laser at 10% intensity, and fluorescence recovery in the bleached region was monitored for 90 s in streaming mode.

For the Tau FRAP assay, imaging was performed within 1.5 h after the addition of 10% PEG. The center of each condensate was bleached using a 488 nm laser at 7% intensity, and fluorescence recovery was monitored for 120 s in streaming mode.

We are grateful for the IMB Microscopy facility staff for technical support.

*Transmission electron microscopy (TEM).* Transmission electron micrographs were acquired on a JEM 1400plus (JEOL) equipped with a XF416 camera (TVIPS) and the EM-Menu software (TVIPS).

### Quantification and analysis

Unless otherwise specified, all statistical analyses were performed using GraphPad Prism 10.5, with specific details provided in each figure legend. Blinding was applied during data acquisition and analysis to prevent observational bias.

*Quantification of in vitro condensation and aggregation.* The formation of Alexa488-labeled TDP-43 condensates and aggregates in the presence or absence of unlabeled Tau, MBP or α-synuclein were quantified using an in-house script made with Python 3.8 through the AICSImageIO library (Maxfield Brown et al, 2021), a tool designed for reading multidimensional bioimaging formats. To mitigate background noise, the intensity of pixels with an intensity below a threshold of 10 or 50 (for the phase separation and aggregation assay, respectively) was set to 0, while preserving the other pixel values. Then, the images were thresholded using Li thresholding algorithm and labeled through scikit-image package. The images were then assessed for object quantity and area through scikit-image to analyse condensates and aggregates, respectively.

The colocalization of DyLight650-labeled Tau and its variants within Alexa488-labeled TDP-43 condensates and aggregates was quantified using Python 3.10.14 through the BioIO library (Maxfield Brown et al, 2021). To mitigate background noise, the background level intensities were set to 0 for each channel, while preserving the other pixel values. Colocalization of Alexa488-labeled TDP-43 (channel 2) and DyLight650-labeled Tau, pTau, Tau-NTD, or Tau-RD (channel 1) were performed using numpy arrays. Objects were labeled with a threshold intensity of 65–15 using the scikit-image library. Objects with a pixel size smaller than 10–20 pixels were identified through the scikit-image library and discarded from the analysis. The area of objects in the colocalized signal was calculated using the scikit-image library.

For the other TDP-43 and Tau in vitro experiments, the quantification was always carried out with the ImageJ/Fiji software, analysing the area or the mean fluorescence intensity of particles bigger than 0.3 µm, with a circularity of 0.00–1.00.

*XL-MS analysis.* For the identification of crosslinking data, spectra were search using the MaxQuant software version 2.4.13 embedded

with the Andromeda search engine (Cox and Mann, 2008; Yılmaz et al, 2022) against a database containing the sequence of TDP-43-MBP, Tau, MBP and TEV proteins and contaminants. The acquired data were searched using as crosslinking reagents DSS light (linked composition $H_{10}C_8O_2$, hydrolyzed composition $H_{12}C_8O_3$, specificity for KSTY) and against DSS heavy isotope label (linked composition $H_2C_8O_2Hx_{10}$, hydrolyzed composition $Hx_{12}C_8O_3$, specificity for KSTY). The search was performed with the predefined options, except for the maximum number of missing cleavages (4) and peak refined enabled. Results were filtered based on FDR model at PSM FDR crosslink (<1%FDR). In the analysis, only inter-XL peptides identified in the same run with light and heavy DSS crosslink at a retention time that differs at maximum 5 min were considered. Furthermore, inter-XL peptides identified in at least two of three replicates, after manual spectra inspection were considered and showed using XiView (Combe et al, 2024). The entire dataset, including raw data, generated tables (Fig. 2, Appendix Fig. S2), and scripts used for the data analysis are available in the PRIDE repository (Deutsch et al, 2023).

*AlphaFold prediction analysis.* AlphaFold prediction of crosslinking: AlphaFold2 multimer (version 2.3) (Evans et al, 2021; Jumper et al, 2021) to model the interaction between TDP-43 and TAU using a fragmented approach (Lee et al, 2024). Briefly, we designed an AlphaFold query by splitting the protein sequence according to known folded domains, for the IDRs, the sequence was divided into 50 amino acids segments using overlapping windows of 5 amino acids offset. In total we tested 60 combinations generated from 6 TDP-43 fragments (1–82; 83–104; 105–263; 264–312; 308–357; 354–414) and 10 TAU441 fragments (1–50; 46–95; 91–140; 136–185; 181–230; 226–275; 271–320; 316–365; 361–410; 406–441). For each prediction, 5 structure models were generated. Predicted structures were assessed with the pDockQ-score and only the highest score model for each pairwise combination was considered. A summary of the structural models that were produced is reported in Appendix Fig. S2.

*SDD-AGE analysis.* The SDD-AGE analysis was performed in ImageJ/Fiji by creating a ROI for each different band size (monomers, oligomers, high molecular species as indicated) for all the time points. ROIs were then measured as area of the bands and values were normalized to the sum of the area of the bands at time 0.

*FRAP analysis.* For the TDP-43 FRAP experiment, analysis was performed using ImageJ/Fiji by measuring fluorescence intensity over time (I(t)) with the Multi Measure function. The analysis was based on the following formula: (ROI1(t) - ROI3(t))/(ROI2(t) − ROI3(t)), where ROI1 represents the average gray value within the bleached region, ROI2 the average gray value of the entire droplet, ROI3 the average background intensity. To account for background noise and photobleaching over time, the fluorescence intensity values were normalized to the initial pre-bleach intensity by dividing I(t) by the mean gray value of the frame immediately before bleaching, denoted as ⟨I(1)⟩. All statistical analyses were performed using GraphPad Prism version 10.5. Significant differences were determined by calculating the area under the curve (AUC) for each condition.

For the Tau FRAP experiment, FRAP recovery curves were obtained using a custom script on Fiji v1.54p (Schindelin et al, 2012). First, the TrackMate plugin (Tinevez et al, 2017) was used to correct for xy-drift of the bleached region of interest over the timelapse using the Linear Assignment Problem tracker (Jaqaman et al, 2008). The mean intensity inside the bleached spot was then measured in each frame of the drift-corrected timelapse ($I_{bleach}$ (t)).

The mean intensity in another unbleached droplet in the field of view was used as a reference to account for variations in signal intensity due to global bleaching ($I_{ref}$ (t)).

Finally, a representative region in the background was chosen to determine the mean background intensity in each frame ($I_{bg}$ (t)) and the mean intensities of the bleached spot and reference droplet were corrected by subtracting the background intensity

$$I'_{bleach}(t) = I_{bleach}(t) - I_{bg}(t),$$

$$I'_{ref}(t) = I_{ref}(t) - I_{bg}(t).$$

The intensity in the bleached spot was then double-normalized relative to the reference intensity and pre-bleach Intensity as follows:

$$I_{norm}(t) = \frac{I'_{bleach}(t)/I'_{bleach,pre}}{I'_{ref}(t)/I'_{ref,pre}},$$

where $I'_{bleach,pre}$ and $I'_{ref,pre}$ refer to the mean values from all pre-bleach timepoints.

For statistical analysis, we fitted a nonlinear mixed effects model with autocorrelation in R version 4.4.3, using the R packages *nlme* (*Mixed-Effects Models in S and S-PLUS*, 2000) and *emmeans* (Lenth, 2025). We used the time series up to 20 s post-bleaching. For later times, artifactual intensity changes due to movement along the *z* axis dominated. We fitted the exponential recovery model

$$I_{norm}(t) = I_0 + q_{mobile}(1 - I_0)\left(1 - 2^{-\frac{t}{t_{0.5}}}\right),$$

for the post-bleach dynamics. This model is equivalent to a classical exponential recovery model, which is implemented in the easyFRAP tool (Koulouras et al, 2018). The difference is that the parameters of interest—initial value $I_0$, half value time $t_{0.5}$, and mobile fraction $q_{mobile}$—appear directly in the model equation, facilitating statistical inference. We included experimental *group*, *replicate*, and individual *curve* as predictive factors in the model to analyze their influence on the model parameters. The primary factor *group* was implemented as a fixed effect. The nested factors *replicate* and *curve* were included as random effects to account for the longitudinal nature of the data and batch effects. We assumed a diagonal covariance structure for the factor *replicate* (i.e., independent effects on model parameters), while an unstructured covariance matrix was used for *curve* (allowing correlated effects). We modelled serial correlation by applying an autoregressive correlation structure of order 1 (corAR1) within curves. Model selection was guided by the Akaike Information Criterion (AIC) and likelihood ratio tests.

*Electron microscopy and immunogold quantification.* Electron microscopy (EM) structures shown in Figs. 5C and 6A were quantified by manually drawing a line along each structure and applying the *Measure* function in ImageJ, with lengths recorded in nanometers (nm).

For the immunogold staining experiment, structures were classified as positive if they displayed a minimum of two gold

particle signals. Background signal was estimated by counting the number of false-positive structures in the control conditions (Appendix Fig. S6C). For each experimental condition depicted in Fig. 5F, structures labeled with Tau antibody (Tau mAB), TDP-43 antibody (TDP-43 mAB), or co-labeled with both antibodies were counted using the same two-particle threshold for positivity. Final values were obtained by subtracting the background count from the total number of positive structures in each condition.

*Analysis of cellular images (Tau and TDP-43 seeding).* Analysis of the Tau cellular seeding assay was performed in ImageJ/Fiji software. Briefly, cytoplasmic Tau aggregates were quantified based on the FRET signal detected in the green channel. Aggregates were identified using the Find Maxima function with a prominence threshold set between 120 and 160, and the results were plotted as the number of aggregates per cell. Alternatively, a threshold range of 80–200 was applied to the green channel to measure the total aggregate area, which was then used for quantitative analysis in the final graph.

For the TDP-43 seeding assay, images were read using Python 3.10.14 through the BioIO library (Maxfield Brown et al, 2021). To mitigate background noise, the background level intensities were set to 0 for each channel, while preserving the other pixel values. Cellpose3 (Stringer and Pachitariu, 2025) was used to create an AI-assisted mask of nuclei from the DAPI channel (with the parameters: flow threshold=1, diameter=40, cell probability threshold=0, model type = "cyto3"), enabling the separation of cytoplasmic and nuclear signal for each channel. Colocalization of cytoplasmic HA (channel 2) and cytoplasmic pS409/410-TDP-43 (channel 1) were performed using numpy arrays. Objects were labeled with a threshold intensity of 20–40 using the scikit-image library. Objects with a pixel size smaller than 10–20 pixels were identified through the scikit-image library and discarded from the analysis. The number and area of objects in the colocalized signal was calculated using the scikit-image library and divided per the number of nuclei as determined from Cellpose output.

## Data availability

The mass spectrometry proteomics data have been deposited to the ProteomeXchange Consortium (http://proteomecentral.proteomexchange.org, https://www.ebi.ac.uk/pride/login) via the PRIDE partner repository (Perez-Riverol et al, 2025) with the dataset identifier PXD063858. The source data (both image and numerical data) for Main Figures and Expanded View and Appendix Figures were submitted to BioImage Archive (https://www.ebi.ac.uk/biostudies/bioimages/studies/S-BIAD2123) with accession number S-BIAD2123.

The source data of this paper are collected in the following database record: biostudies:S-SCDT-10_1038-S44318-025-00590-2.

## Peer review information

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

## Acknowledgements

We thank Cornelia Niemann, Fereniki Moschogiannaki, and Georg Kislinger for technical support and Eszter Katona for valuable discussions and comments on the manuscript. We thank Marc Diamond for the generous gift of the HEK293 CFP/YFP-TauRD$^{P301S}$ reporter cell line, Tom Scheidt for labeled alpha-synuclein, Philipp Schönberger for the gift of the His-FKBP-3C-TDP-43(267–414)-TEV-MBP plasmid, and Emre Pekbilir for purification of TDP-43-RRM. We are particularly thankful to the Imaging Core Facilities at the Institute for Molecular Biology (IMB) Mainz for access to microscopy equipment and technical assistance, especially Márton Gelléri's expert input on the FRAP analysis. Our gratitude also extends to the Protein Production Core Facility at IMB Mainz, with special thanks to Sabine Heinen and Martin Möckel for producing recombinant hTau441-WT and TDP-43-CTF-MBP proteins. This work was supported by the Alzheimer Research Award by the Hans and Ilse Breuer Foundation (to DD), an anonymous foundation, and by the Deutsche Forschungsgemeinschaft (DFG) within the Heisenberg Programme (project ID 442698351), SFB1551 (project ID 464588647) and SPP2191 (project ID 419138680 to SW and 419139133 to DD). FU is supported by the Marie Curie Individual Fellowship (Grant agreement number 101207537). MP gratefully acknowledges the support of the Swiss National Science Foundation (SNF, 310030_192650) and the Association for Frontotemporal Dementia (AFTD Biomarker Initiative grant). CS acknowledges the Federal Ministry for Education and Research (BMBF, ZIK programme, 03Z22HN22). The transmission electron microscopy was supported by the DFG under Germany's Excellence Strategy within the framework of the Munich Cluster for Systems Neurology (EXC 2145 SyNergy, ID 390857198, to DD, DE and MS) and SFB TRR 274/2 2024 – 408885537 project Z01 (to MS). The ZEISS LSM 900 was

funded by the Ministry of Science and Health of Rhineland-Palatinate and the European Regional Development Fund (ERDF/REACT-EU, Grant No. 84012490). The Spinning Disk Confocal System (IMB Microscopy Core Facility) was supported by the DFG (INST 247/912-1FUGG), and the Stellaris spinning disc confocal microscope (IMB Microscopy Core Facility) is funded by the DFG (project number 497669232).

## Author contributions

**Francesca Simonetti**: Conceptualization; Data curation; Formal analysis; Validation; Investigation; Visualization; Methodology; Writing—original draft; Writing—review and editing. **Weijia Zhong**: Data curation; Formal analysis; Investigation; Visualization; Methodology; Writing—review and editing. **Saskia Hutten**: Data curation; Formal analysis; Investigation; Visualization; Methodology; Writing—review and editing. **Federico Uliana**: Data curation; Formal analysis; Investigation; Visualization; Methodology; Writing—review and editing. **Martina Schifferer**: Resources; Methodology; Writing—review and editing. **Ali Rezaei**: Data curation; Software; Formal analysis; Writing—review and editing. **Lisa Marie Ramirez**: Formal analysis; Investigation; Methodology; Writing—review and editing. **Janine Hochmair**: Methodology; Writing—review and editing. **Rithika Sankar**: Methodology; Writing—review and editing. **Anusha Gopalan**: Formal analysis; Writing—review and editing. **Fridolin Kielisch**: Formal analysis; Writing—review and editing. **Henrick Riemenschneider**: Methodology; Writing—review and editing. **Viktoria Ruf**: Resources; Writing—review and editing. **Carla Schmidt**: Resources. **Mikael Simons**: Resources; Writing—review and editing. **Markus Zweckstetter**: Resources; Methodology; Writing—review and editing. **Susanne Wegmann**: Resources; Methodology; Writing—review and editing. **Tammaryn Lashley**: Resources; Methodology; Writing—review and editing. **Magdalini Polymenidou**: Resources; Methodology; Writing—review and editing. **Dieter Edbauer**: Resources; Supervision; Funding acquisition; Validation; Visualization; Methodology; Writing—original draft; Project administration; Writing—review and editing. **Dorothee Dormann**: Conceptualization; Resources; Supervision; Funding acquisition; Validation; Visualization; Methodology; Writing—original draft; Project administration; Writing—review and editing.

Source data underlying figure panels in this paper may have individual authorship assigned. Where available, figure panel/source data authorship is listed in the following database record: biostudies:S-SCDT-10_1038-S44318-025-00590-2.

## Funding

## Disclosure and competing interests statement

The authors declare no competing interests.

# Expanded View Figures

**Figure EV1.   pTau, but not the Tau N-terminal domain or repeat domain, enriches in TDP-43 condensates and aggregates.**

(**A**) Scheme of recombinant Tau variants with their respective net charges; created with BioRender.com. (**B**) Coomassie-stained SDS-PAGE gel showing the molecular sizes of Tau, phosphorylated Tau (pTau), Tau N-terminal domain (Tau-NTD), and the Tau-RD fragment (R1–R4 repeat domains). (**C**) Confocal microscopy images of Alexa488-labeled TDP-43 (5 µM) in absence or presence of DyLight650-labeled Tau, pTau, Tau-NTD, or Tau-RD in a phase separation assay. Scale bar: 15 µm in overview and 3 µm in inset. (**D**) Colocalization of Tau, pTau, Tau-NTD, or Tau-RD within TDP-43 condensates quantified by measuring the area of condensates exhibiting overlapping green and far-red signals. Quantification was performed across ($n = 3$) biological replicates, and bar graphs show values ± SEM. (**E**) Confocal microscopy images of Alexa488-labeled TDP-43 (5 µM) in absence or presence of DyLight650-labeled Tau, pTau, Tau-NTD, or Tau-RD in an aggregation assay. Scale bar: 15 µm in overview and 3 µm in inset. (**F**) Colocalization of Tau, pTau, Tau-NTD, or Tau-RD within TDP-43 aggregates quantified by measuring the area of aggregates exhibiting overlapping green and far-red signals. Quantification was performed across ($n = 3$) biological replicates, and bar graphs show values ± SEM.

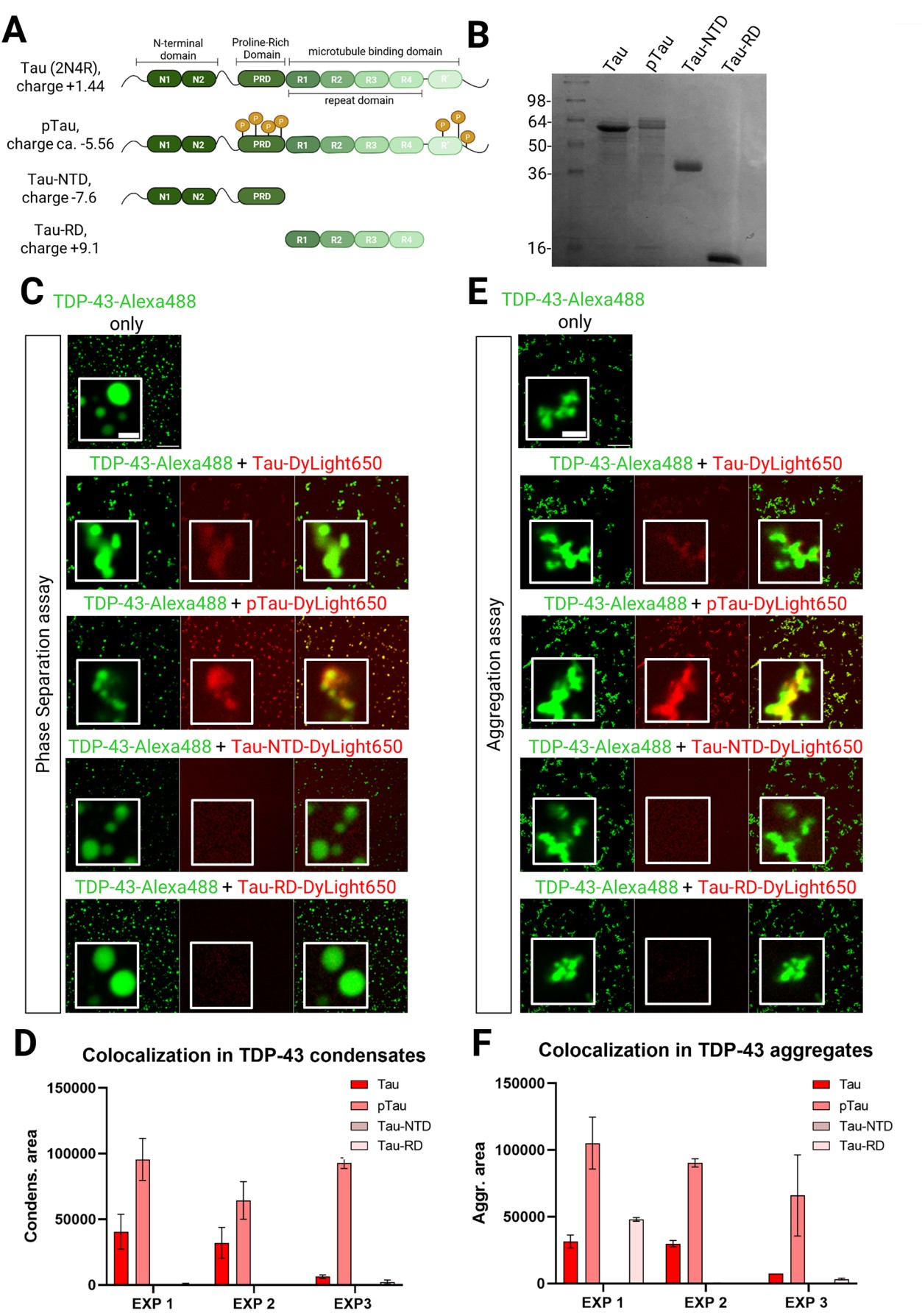

**A** Tau (2N4R), charge +1.44 — N-terminal domain (N1, N2), Proline-Rich Domain (PRD), microtubule binding domain (R1, R2, R3, R4), repeat domain

pTau, charge ca. -5.56

Tau-NTD, charge -7.6

Tau-RD, charge +9.1

**B** Tau, pTau, Tau-NTD, Tau-RD; 98, 64, 50, 36, 16

**C** Phase Separation assay
TDP-43-Alexa488 only
TDP-43-Alexa488 + Tau-DyLight650
TDP-43-Alexa488 + pTau-DyLight650
TDP-43-Alexa488 + Tau-NTD-DyLight650
TDP-43-Alexa488 + Tau-RD-DyLight650

**E** Aggregation assay
TDP-43-Alexa488 only
TDP-43-Alexa488 + Tau-DyLight650
TDP-43-Alexa488 + pTau-DyLight650
TDP-43-Alexa488 + Tau-NTD-DyLight650
TDP-43-Alexa488 + Tau-RD-DyLight650

**D** Colocalization in TDP-43 condensates — Condens. area (Tau, pTau, Tau-NTD, Tau-RD) across EXP 1, EXP 2, EXP3

**F** Colocalization in TDP-43 aggregates — Aggr. area (Tau, pTau, Tau-NTD, Tau-RD) across EXP 1, EXP 2, EXP 3

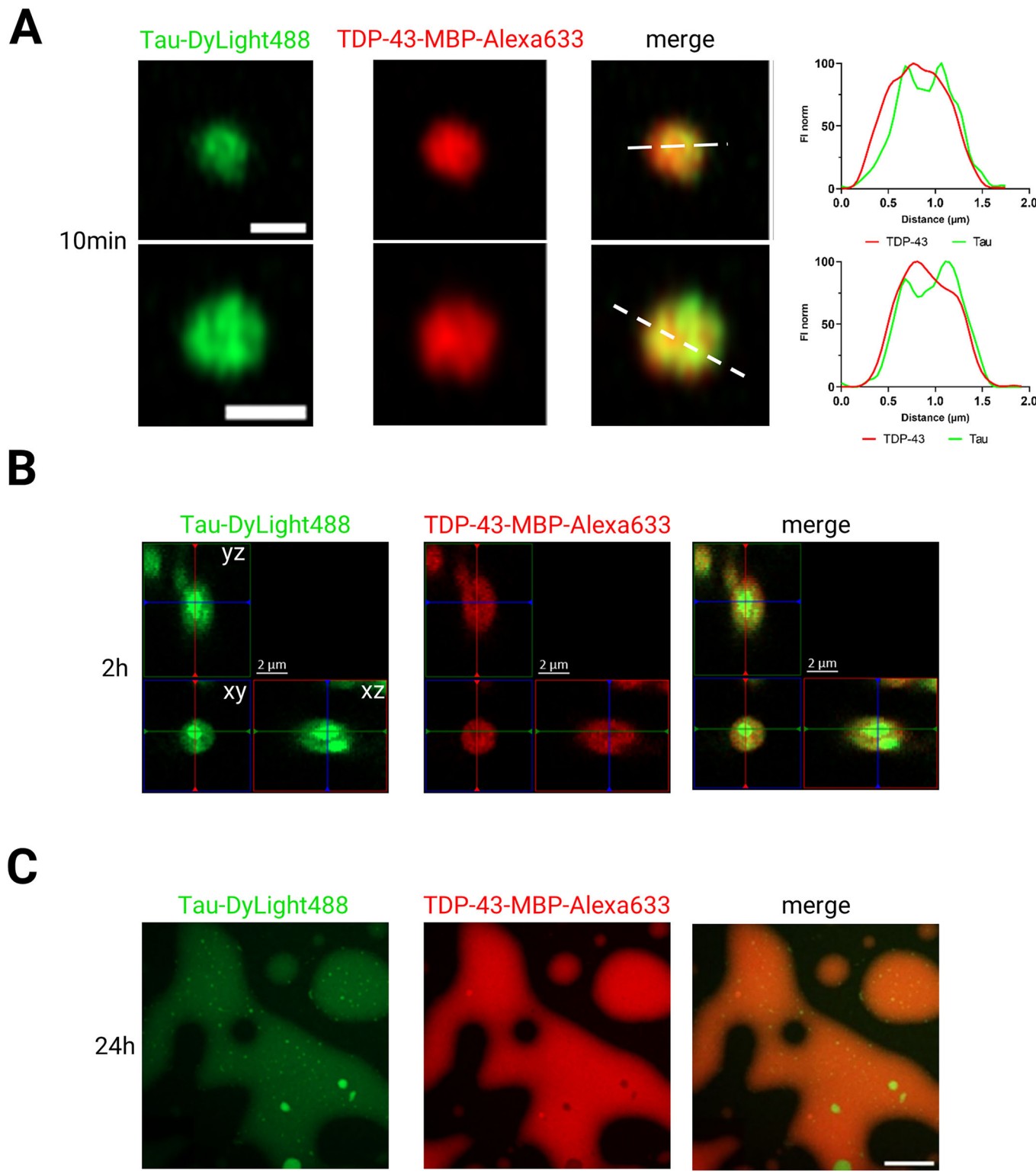

**Figure EV2.  Early intra-droplet structures appear at the initial stages of multiphasic Tau/TDP-43 co-condensate formation.**

(A–C) High resolution images using AiryScan of 1:1 DyLight488-labeled Tau with Alexa633-labeled TDP-43-MBP mixed condensates after 10 min (**A**), 2 h (**B**) or (**C**) 24 h incubation. Scale bar: 1 μm (**A**), 2 μm (**B**) or 10 μm (**C**). In (**A**), dotted lines show the origin of line profiles indicated on the right, demonstrating demixing of Tau and TDP-43 within the same condensate at early timepoints.

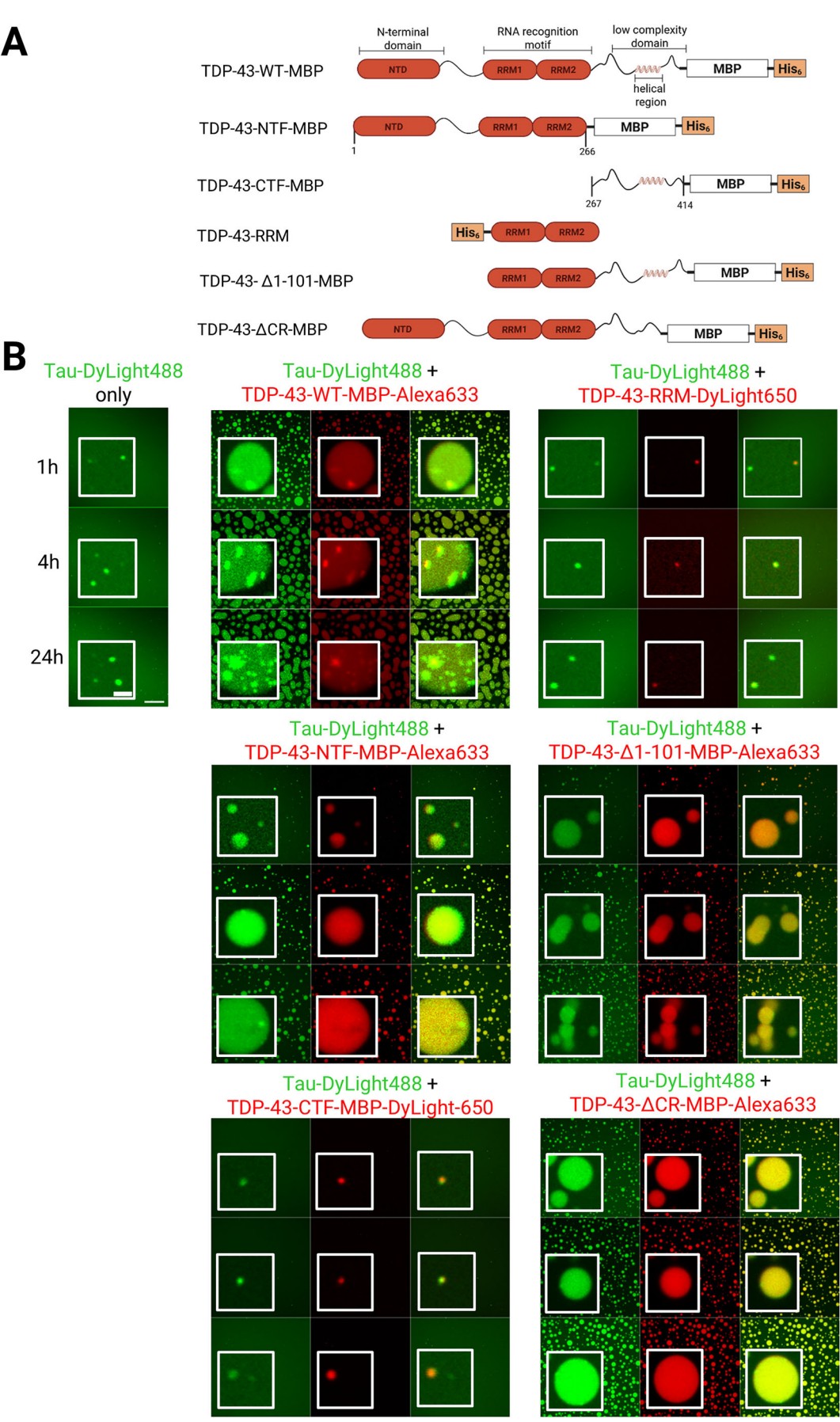

**Figure EV3. Distinct regions of TDP-43 contribute to Tau phase separation and the formation of multiphasic co-condensates.**

(A) Scheme of recombinant TDP-43 deletion mutants; created with BioRender.com. (B) Confocal microscopy images of 4 µM DyLight488-labeled Tau in equimolar presence of Alexa633-labeled TDP-43-WT-MBP, TDP-43-NTF-MBP, TDP-43-CTF-MBP, TDP-43-RRM, TDP-43-Δ1-101-MBP, or TDP-43-ΔCR-MBP at the indicated timepoints (1, 4, 24 h). Scale bar: 20 µm in overview and 2 µm in inset.

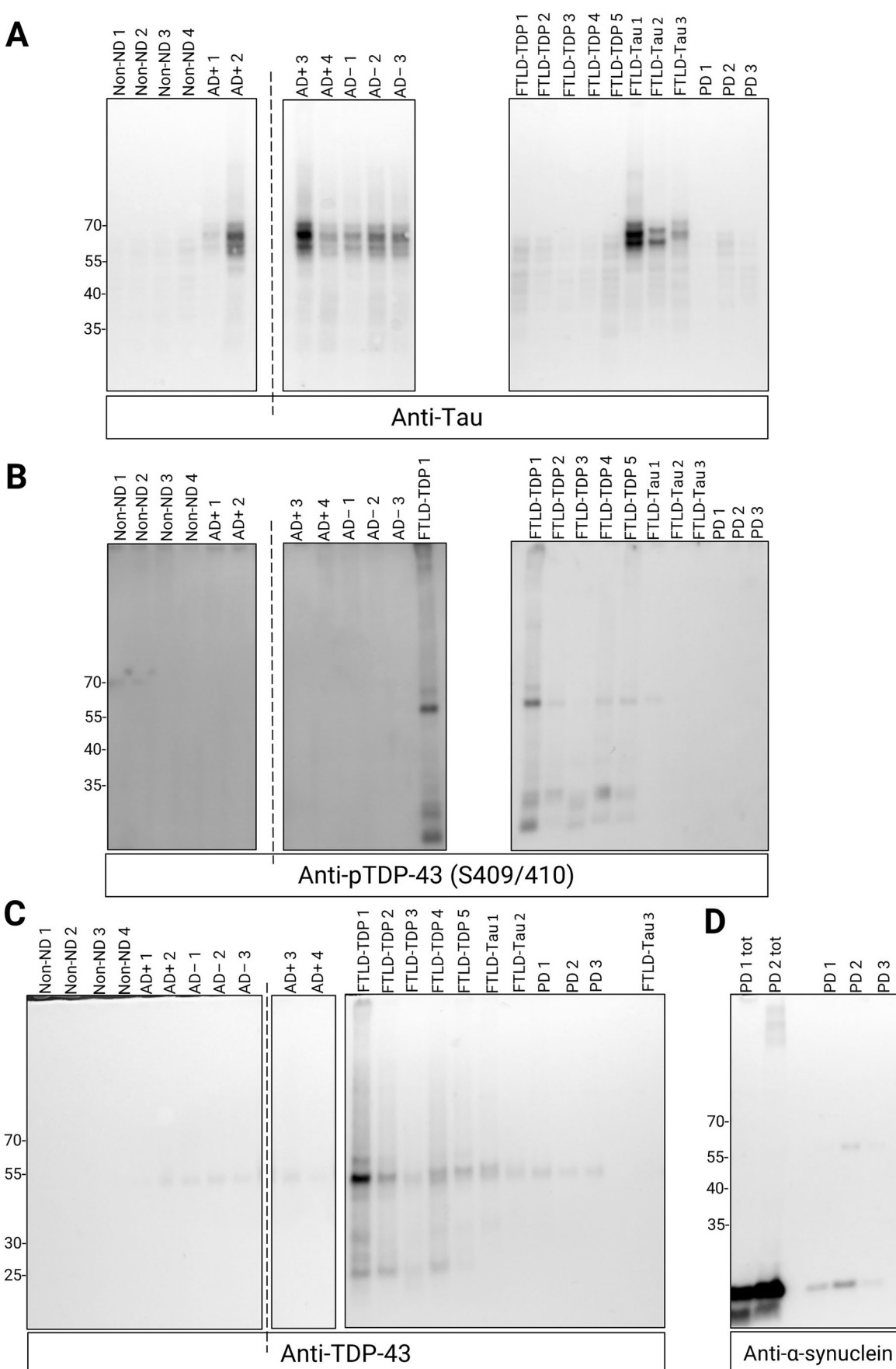

◀ **Figure EV4. Western blot characterization of SarkoSpin extracts from human patients.**

(**A**) anti-Tau antibody; (**B**) Anti-phospho-TDP-43 (S409/410) antibody; (**C**) Anti-TDP-43 antibody; (**D**) Anti-α-synuclein antibody. Western blots of SarkoSpin fractions derived from the frontal cortex of non-ND, AD +, AD–, FTLD-TDP, and FTLD-Tau patients, and from the cingulate cortex of PD patients, probed with the indicated antibodies. Dashed lines indicate divisions within the same blot.

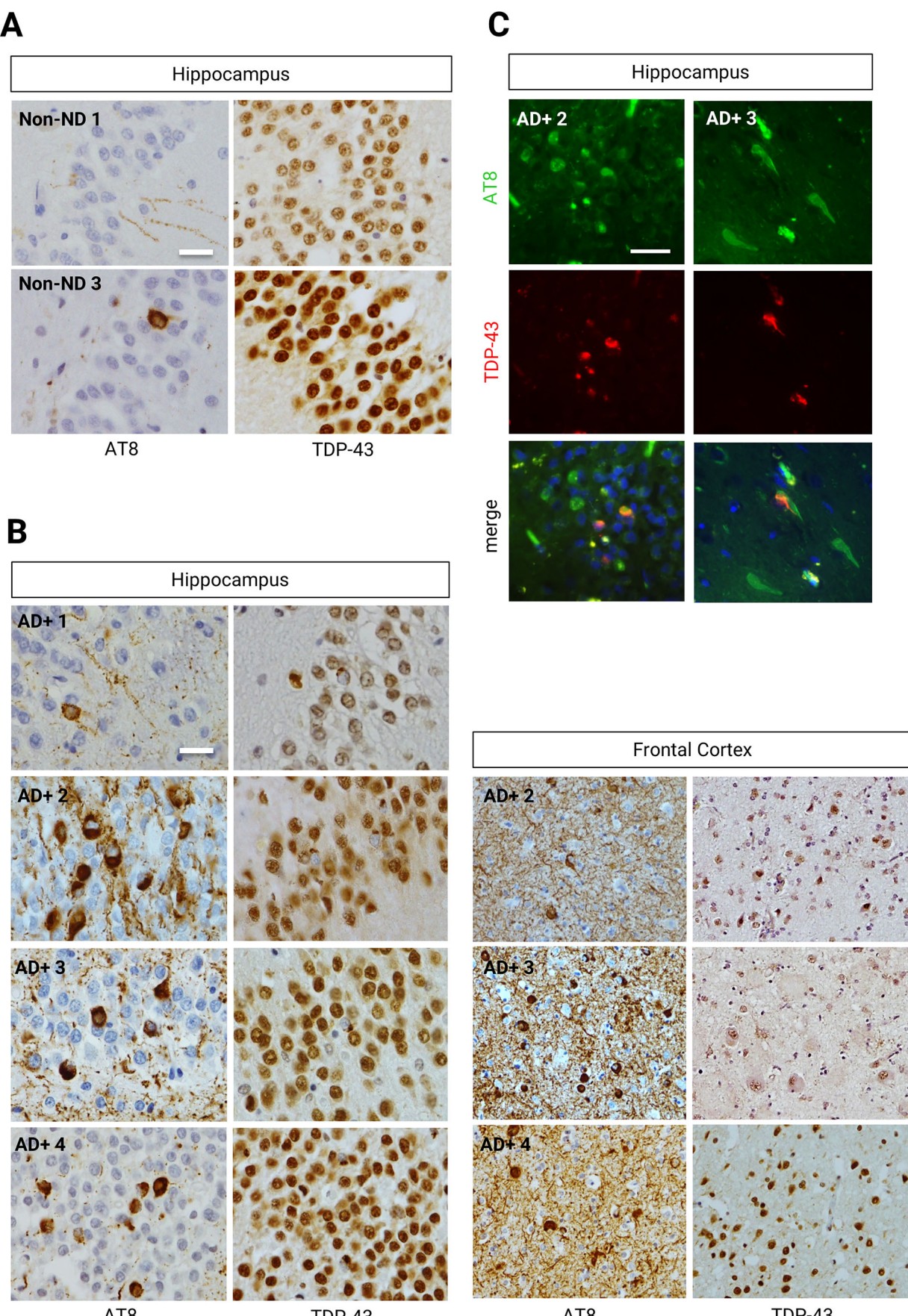

◀ **Figure EV5. Characterization of patient-derived SarkoSpin extracts by immunohistochemistry and dual immunofluorescence.**

(A, B) Immunohistochemical images of representative non-ND and AD+ brain sections stained with antibodies against phosphorylated Tau (AT8) and TDP-43 in either the hippocampus or frontal cortex. Scale bar: 50 μm. (C) Double fluorescence immunostaining of two representative AD+ cases showing phosphorylated Tau (AT8, green) and TDP-43 (red) in the hippocampus. Scale bar: 30 μm.

