## [Peer Review File · The EMBO Journal]

Direct interaction between TDP-43 and Tau promotes their co-condensation, while suppressing Tau fibril formation and seeding

Francesca Simonetti, Weijia Zhong, Saskia Hutten, Federico Uliana, Martina Schifferer, Ali Rezaei, Lisa Ramirez, Janine Hochmair, Rithika Sankar, Anusha Gopalan, Fridolin Kielisch, Henrick Riemenschneider, Viktoria Ruf, Carla Schmidt, Mikael Simons, Markus Zweckstetter, Susanne Wegmann, Tammaryn Lashley, Magdalini Polymenidou, Dieter Edbauer, and Dorothee Dormann

Corresponding author(s): Dorothee Dormann (ddormann@uni-mainz.de) , Dieter Edbauer (Dieter.Edbauer@dzne.de)

Review Timeline:

Submission Date:	26th Jul 24
Editorial Decision:	20th Sep 24
Revision Received:	30th Jun 25
Editorial Decision:	12th Aug 25
Revision Received:	28th Aug 25
Accepted:	5th Sep 25

Editor: Ioannis Papaioannou

Transaction Report:

Dear Dorothee,

Thank you again for submitting your manuscript EMBOJ-2024-118442 to The EMBO Journal for our consideration. Please accept our apologies for the rather protracted process on this occasion, which was due to difficulties in securing enough reviewers with suitable expertise to comment on all aspects of your work at the time your manuscript was submitted to our journal. Thank you for your understanding and your patience.

Your manuscript has now been seen by three experts in the field, and we have received the full set of their comments, which I have already shared with you (they are included again below). The referees show interest in your study and emphasize that the results are of significant potential impact on understanding the role of interactions between TDP-43 and Tau in neurodegeneration. However, they also identify several limitations and shortcomings that overall reduce the robustness and solidity of the manuscript as it stands. They also point out that further insight into the mechanism should be feasible and would significantly strengthen the study. All three referees have provided very detailed and well-informed reports with constructive suggestions for improvement of the study and the manuscript.

In light of this input and the potential significance of your findings, we would be open to considering a significantly strengthened version of your manuscript should you be able and willing to embark on a major revision taking on board the helpful comments and suggestions of the referees. We realize that such a revision would entail a significant amount of additional work, and we would thus be open to extending the revision time beyond our standard 3-month period (December 19, 2024) should this be necessary to allow enough time for the acquisition of the new data and revision of the manuscript. Your resubmission should be accompanied by a detailed point-by-point response addressing all referees' comments and concerns. I should add that it is the EMBO Journal policy to allow only a single round of major revision, and acceptance of your manuscript will therefore depend on the completeness of your responses in this revised version.

I would also like to encourage you to prepare and share with me a point-by-point response to the referee comments and a provisional revision plan if there are any particular referee comments you do not agree with or you cannot address in a revision. This information would be very helpful for us to discuss further the feasibility of such a revision and its potential fit for publication in The EMBO Journal. Please also let me know if you have any other questions or comments you would like to discuss further with me.

Thank you for the opportunity to consider your work for publication in The EMBO Journal. I look forward to your revision.

Best regards,

Ioannis

Instructions for preparing your revised manuscript

1. When you are ready to submit the revision, please upload:

- A Word file of the manuscript text (including legends of main Figures, EV Figures and Tables). Please make sure that changes are highlighted (or "tracked") to be clearly visible.

- Individual production-quality figure files (one file per figure). When assembling your figures, please refer to our figure preparation guidelines in order to ensure proper formatting and readability in print as well as on screen:

If the data shown in a figure are obtained from n {less than or equal to} 2, please use scatter plots showing the individual data points.

i. the name of the statistical test used to generate error bars and P values

ii. the number (n) of independent experiments (please specify technical or biological replicates) underlying each data point

(discussion of statistical methodology can be reported in the Materials and Methods section, but figure legends should contain a

basic description of n, P, and the test applied)

iii. the nature of the bars and error bars (s.d., s.e.m.).

- A point-by-point response to the referees' comments, with a detailed description of the changes made (as a word file). All referees' concerns must be fully addressed and their suggestions taken on board. When preparing your letter of response to the referees' comments, please bear in mind that this will form part of the Review Process File and will therefore be available online to the community. Please note that you have the possibility to opt out of the transparent process at any stage prior to publication by letting the editorial office know (contact@embojournal.org); if you do opt out, the Review Process File link will point to the following statement: "No Peer Review File is available with this article, as the authors have chosen not to make the review process public in this case.". For more details on our Transparent Editorial Process, please visit our website:

<https://www.embopress.org/page/journal/14602075/authorguide#transparentprocess>

- Expanded View (EV) files (replacing Supplementary Information) that are collapsible/expandable online. A maximum of 5 EV Figures can be typeset. EV Figures should be cited as "Figure EV1, Figure EV2" etc. in the text, and their respective legends should be included in the manuscript file after the legends of regular figures. See detailed instructions regarding Expanded View files here:

- For the figures that you do NOT wish to display as Expanded View figures, they should be bundled together with their legends in a single PDF file called "Appendix", which should start with a short Table of Contents (including page numbers). Appendix figures should be referred to in the main text as: "Appendix Figure S1, Appendix Figure S2" etc. Please see detailed instructions here: <https://www.embopress.org/page/journal/14602075/authorguide#expandedview>

- A complete author checklist, which you can download from our author guidelines

(<https://www.embopress.org/page/journal/14602075/authorguide>). Please note that the checklist will also be part of the Review Process File.

2. Please note that no statistics should be calculated and shown in Figures if $n=2$. Please also note that each p value should be reported as an exact value.

3. Before submitting your revision, primary datasets (and computer code, where appropriate) produced in this study need to be deposited in appropriate public databases (see <https://www.embopress.org/page/journal/14602075/authorguide#dataavailability>). Their accession numbers, databases, and the specific URLs (links) should be listed in a formal "Data availability" section (placed after Methods).

*** The Data Availability Section is restricted to new primary data that are part of this study. In case you have no data that require deposition in a public database, please state so instead of referring to the database: "Our study includes no data deposited in public repositories." under the heading "Data availability". ***

*** All links should resolve to a page where the data can be accessed. ***

*** Please remember to provide in the Data availability section of your revised manuscript reviewer passwords if the datasets are not yet public. ***

*** Please use detailed data citations for already available datasets that were re-analyzed in your study - for more information on the format, see point #9 below. ***

4. Please check that the title and the abstract of the manuscript are brief, yet explicit, even to non-specialists. The length of the title should not exceed 100 characters, and the abstract should be a single paragraph not exceeding 175 words.

5. All materials and methods need to be described in the manuscript using our "Structured Methods" format, which is now required for all research articles. According to this format, the Methods section includes a single "Reagents and Tools Table" - listing key reagents, experimental models, software and relevant equipment including their sources and relevant identifiers- followed by a "Methods and Protocols" section describing the methods. Please download and fill our Reagents and Tools Table template (.docx), which you can find in our author guide:

<https://www.embopress.org/page/journal/14602075/authorguide#structuredmethods>. When submitting your revised manuscript, please do not include the Reagents and Tools Table in the Methods section of the manuscript but upload it as a separate file choosing the file type "Reagent Table".

6. Please also note our reference format: <https://www.embopress.org/page/journal/14602075/authorguide#referencesformat>.

7. At EMBO Press we ask authors to provide source data for the main manuscript figures. Our source data coordinator will contact you to discuss which figure panels we would need source data for and will also provide you with helpful tips on how to

upload and organize the files.

8. Please remember: digital image enhancement is acceptable practice, as long as it accurately represents the original data and conforms to community standards. If a figure has been subjected to significant electronic manipulation, this must be noted in the figure legend or in the "Materials and Methods" section. The editors reserve the right to request original versions of figures and the original images that were used to assemble the figure.

9. Our journal encourages inclusion of data citations in the reference list to directly cite datasets that were obtained from public databases. Data citations in the article text are distinct from normal bibliographical citations and should directly link to the database records from which the data can be accessed. In the main text, data citations are formatted as follows: "Data ref: Smith et al, 2001" or "Data ref: NCBI Sequence Read Archive PRJNA342805, 2017". In the Reference list, data citations must be labeled with "[DATASET]". A data reference must provide the database name, accession number/identifiers, and a resolvable link to the landing page from which the data can be accessed at the end of the reference. Further instructions are available at: <https://www.embopress.org/page/journal/14602075/authorguide#referencesformat>.

10. We request authors to consider both actual and perceived competing interests. Please review our policy (<https://www.embopress.org/page/journal/14602075/authorguide#conflictsofinterest>) and update your competing interests statement if necessary. Please name this section 'Disclosure and competing interests statement' and place it after the Acknowledgements section.

11. Please note that all corresponding authors are required to provide an ORCID ID upon submission of a revised manuscript (<https://orcid.org/>). Please find instructions on how to link your ORCID ID to your account in our manuscript tracking system in our Author guidelines (<https://www.embopress.org/page/journal/14602075/authorguide#authorshipguidelines>).

12. We use CRediT to specify the contributions of each author in the journal submission system. CRediT replaces the author contribution section, which should be removed from the manuscript. Please use the free text box to provide more detailed descriptions. See also guide to authors: <https://www.embopress.org/page/journal/14602075/authorguide#authorshipguidelines>.

14. We would also welcome the submission of cover suggestions or motifs to be used by our Graphics Illustrator in designing a cover.

15. Please use the link below to submit your revision:
<https://emboj.msubmit.net/cgi-bin/main.plex>

Referee #1:

Simonetti et al. present evidence for interactions between TDP-43 and Tau during protein condensation, aggregation and aggregate seeding activity. These findings are relevant to understanding the etiology of multiple neurodegenerative disorders, in which these two proteins play key roles. Growing evidence show co-occurrence of pathology due to TDP-43 and Tau inclusions. Based on their findings, Simonetti et al. propose that TDP-43-Tau interactions alter the properties of their cognate condensates as well as aggregation in vitro and aggregate seeding in cell-based reporters. Hence, these findings are of great potential impact to understanding basic mechanisms of neurodegeneration. The work, however, lacks important controls and key assays to support the conclusions. In addition, the findings on the effect of TDP-43 on Tau aggregation and seeding activity are somewhat inconsistent. These important concerns reduce the impact of the studies.

Major concerns:

- The experiments describing changes in condensate properties upon mixing Tau and TDP-43 should include FRAP assays to determine whether changes in dynamic properties cause the differences in condensate morphology reported in various figures: Fig 1B, C, EV1B,D.
- The assays throughout the manuscript should be consistently conducted in the absence of the C-terminal MBP tag for recombinant TDP-43, full length and fragments. Some experiments, e.g., condensate formation in Fig 1, were performed after cleaving off the MBP tag. On the other hand, others, such as Figs 2, 4, and 5 used full-length or fragment TDP-43-MBP. One of the reasons for this concern is the previously established effect of C-terminally tagged TDP-43 using MBP on the condensation properties of the protein, suggesting that MBP alters the behavior of the disordered C-tail. Thus, MBP may alter TDP-43

aggregation and condensation properties. In addition, Fig 3 shows strong colocalization of MBP alone with Tau. The authors should discuss this observation and the impact of this behavior on the other results that involve MBP-tagged proteins.

- Condensation assays in Fig 3 and EV should be conducted in the absence of PEG and heparin as the presence of these coacervate promoters could diminish the significance of the observations.
- The authors should better define the results in Fig 3C as it is difficult to make conclusions on the behavior of TDP-43-MBP in the context of the Tau assemblies, especially at 4 and 24h time points.
- Based on the findings in Fig 4, the authors suggest that full length TDP-43 suppresses Tau fibrilization, whereas the CTD-MBP construct does not significantly alter fibril length. This is in contrast with findings in Fig 5B, where CTD-MBP appears to reduce Tau aggregate seeding in the reporter cell line. The authors should discuss these seemingly contradictory observations.
- The experiments in Fig 6, studying seeding activity of patient-derived extracts for Tau and TDP-43 should include immunodepletion experiments to remove any presence of Tau from FTLD-TDP and TDP-43 from AD-derived samples. This control will help determine whether seeding activity is caused by cross-seeding or undetectable amounts of TDP-43/Tau in the patient-derived samples.
- The aggregates in Figs 5 and 6 should be quantified in terms of area instead of number of aggregates per cell.

Referee #2:

This manuscript probes how direct interaction between TDP-43 and tau alters their phase separation and aggregation in vitro and in cells. Tau and TDP-43 enhance each other's phase separation into solid type aggregates. While tau enhances aggregation of TDP-43, TDP-43 prevents formation of regular fibrils of tau. This effect is demonstrated to arise from the N-terminal domains of TDP-43, not the C-terminal domain. Excitingly, brain material from FTLD with TDP-43 aggregates induces tau aggregation in cell model detecting tau seeds, while brain material from AD (with tau aggregates) induces TDP-43 aggregation in cells detecting TDP-43 seeds. The pairing of the biochemical and in cell data is excellent and makes a strong case. The mechanism could be addressed further, with additional experiments, to test the robustness of the findings and bring a means to rationalize the exciting findings.

Major

Throughout the biochemical sections, MBP and alpha synuclein are used as controls. This is a good idea, but does not provide a handle on why. The authors should perform experiments with sequence scrambled tau, mutant forms of tau and TDP-43, etc to answer the question what is it about tau? Is it just the net charge (how different is this than the other proteins?) as tau and TDP-43 are both known to be altered by nucleic acid interactions. The lack of detail here decreases the insight - the experiments demonstrate that tau does these things to TDP-43. But, although it is within reach, the experiments do little to explain why. The authors provide some efforts in one section by dividing TDP-43 into two pieces - however they term the N terminal domain something that encompasses a dimerizing/oligomerizing domain and then RNA binding domains. Which of the domains is important or is it both the first domain and the RNA binding domains. Many mutants altering the behavior of these domains (assembly, RNA binding, folding) including disease mutations have been described in the literature.

Pg 3,4 Can the authors show data that the labeling does not affect the results. If unlabeled tau (or TDP-43) is used, do the authors find that TDP-43 is still altered?

The description on page 5 that condensation "gradually formed small condensates over time" is surprising in that phase separation is typically instantaneous unless something else is happening (aggregation, oxidation, etc). The authors should explain what is happening and more clearly show the progression.

Pg 5 The description of "intra-tau" condensates is vague and not descriptive of what is observed - intra to what? And, although the authors say it "remains to be determined" how these intratau condensates are formed, the authors should try to observe this by microscopy as a function of time.

Pg 5 "prevents the formation of tau fibrils". Are these shorter aggregates ThT positive? How do these change other metrics of fibril formation / amyloid formation? This should be tested. The observation should also be more precisely explained, detailing that the progression to mature/long fibrils is inhibited. Whether these fibrils contain the other protein should be checked by immunogold EM or by TIRF using fluorescent proteins already used in the manuscript. These findings may also help answer the potential apparent contradiction that fibril formation is halted but these are co-implicated in aggregation disease.

Presenting biological replicates of Figure 5 C and 6 C,F would be helpful - at least a statement regarding the replicability with different batches of cells and different purifications of proteins.

The data on the cell seeding assays are really exciting. I think they are excellent. However, I think more controls should be shown/performed to understand if this is specific. First, some negative controls showing that Parkinson's or other type of neurodegeneration should be shown to show if these results are specific. Perhaps those exist in the literature but it would be good to show that here. Secondly and complementarily, the suggestion is that seeds of one protein type are present in the

diseases of the other - but the rest of the manuscript is saying that the one protein induces the aggregation or changes the aggregation of the other. So it is not that clear if these seeds are really present. Western blots and experiments in the cells using immunodepleted samples should be added to confirm if these things are present and what they do. If the authors are concerned about co-aggregates in immunodepletion, the authors could show co-ip Westerns then.

Minor points that are easily addressed in text:

Pg2 "more neurotoxic than solid aggregates". Solid aggregates should be replaced with something more specific, such as amyloid fibrils.

Pg2 although it is a new term not favored by all, the term "LATE" should be used in my mind to describe the subset of AD with TDP-43.

Pg 4 "same aggregation assay ... cleaving off the MBP". It would be more clear to explain when TEV is added (and add to the schematic).

Is 150 mM NaCl really physiological for intracellular proteins. May be better to say this is to mimic more physiological conditions rather than say it is physiological.

Fig 3 It is somewhat confusing to start Fig 3 with turbidity as it is not clear if there are changes in liquid vs solid qualitatively before showing what is quantitatively happening.

Fig 3C controls - the authors should comment about MBP recruitment into the condensates, as usually it is assumed it is not recruited into condensates.

Pg 5 The end of the section "Under molecular crowding conditions" could use a summary to keep the reader on track as to what is concluded here.

More details on the synuclein preparation by heat inactivation times and lysis conditions should be added on pg 11. Explanation of the differences in preparation and definition of "afterwards labeled for the experiments" should be added as this is unclear.

Where is TDP-43 labeled by maleimide dyes and does that affect its stability?

Pg 15 - the term "101.41 scaled resolution" is that supposed to field of view or resolution?

Referee #3:

Simonetti et al.

Extrapolating from mixed proteinopathies, this paper assesses interference effects for seeding between Tau and TDP43. For the most part the authors use in vitro or cell-free assays, including studies of phase separation on glass sides, but there are some experiments in HEK293 reporter cells. Endpoints for phase separation phenomena (production of condensates) include the number or particles, roundness, and area per particle, while in the reagent permutations that favor fibril formation, aggregation endpoints are studied by agarose gel electrophoresis and EM. Alpha-syn comprises an internal control with weaker or absent stimulatory effects, depending on the assay. Domain effects are appraised by using N- or C-terminal domain recombinant proteins in the aggregation assays. In the final assays, aggregation effects are followed subsequent to initiation with brain samples from different FTLD and AD patients.

The main observation in the earlier figures is a stimulatory effect of tau on assembly of recTDP43 to generate larger, irregular condensates or increased aggregation. Then, the reverse effect of TDP43 upon tau is assessed with Fig.3, apart from turbidity, displaying qualitative changes in condensates and Fibril formation, Fig. 4. Here an inhibitory effect is scored and mapped to the N-terminal domain. Switching to a FRET biosensor cell assay based on the tau RD linked to two fluorophores and primed by incubated recombinant proteins (Fig. 5), the greatest number of fluorescent foci per cell is obtained with tau plus a TDP43 N-terminal domain. This is confusing, at least in an initial reading, but is attributed to the heightened presence of seed competent oligomers in the samples. From the perspective of experimental variables between the assays, the tau substrate in the cells differs in length from that used in Figs 1-4 not only in length but also in allelic type, by having a P301S FTLD-tau mutation. Fig. 6 uses the same biosensor cells primed with human brain homogenates processed to enrich for sarkosyl insoluble species (at least probably - below). Notably FTLD-TDP can give a strong signal and given the selectivity of the cells, this might be attributed to tau aggregates in these patients. In a reversed assay looking at TDP43 aggregation, AD can give a strong signal. But there are caveats about case selection that suggest that the data in Fig. 6 need to be treated with caution (below).

The authors deduce that there are counterbalancing effects, such as increased phase separation can diminish fibril formation.

Main points.

The paper is not an easy read, and this is nothing to do with language usage (which is fluent). Rather, it is because at various points the paper deals with phase separated condensates, fibrils, sarkosyl-insoluble extracts and putative oligomer preps, which may possess intrinsic opposing properties or may have apparently contradictory properties due to limitations in the assays or from interconversion between biophysical forms. Fig. 7 may have been intended to tie the interpretation of all the results together, but to this reviewer it was not helpful. For example, would the upper panel benefit from a comparison with a pure tauopathy? The lower panel emphasizes AD seeding TDP43 aggregates, but AD more famously has seeding by Abeta and tau species, which are themselves abundant and which are not shown in the diagram. So, to this reader, the paper often comes across as an imperfect collage of raw results. The assays in Fig.6 may have been intended as a capstone experiment for the lab data, to bring the picture together (more below), but the brain samples used are not profiled for their constituent molecules by IHC in any display items. This does beg the question of what the co-deposits look like in situ for these patients, noting that co-aggregates are underscored by the authors in the concluding sentence of the Introduction.

The authors refer to "Sarkosin" extracts without explaining at the outset what they enrich for. Presumably sarkosyl-insoluble materials, which are mainly? - mature fibrils? But then again oligomers are repeatedly mentioned as having superior seeding activity. What is the status of oligomers in these extracts? In other words, to what extent is this Sarkosin procedure biasing the assay outputs and hence the conclusions. What happens when brain extracts are processed by a different procedure before a biosensor assay?

Each disease type in Fig.6 is represented by one bio sample, even though it is known that well characterized disease entities (defined, for example, by a shared germline mutation or extensive pathology profiling) still have a range of activities in seeding and biochemical assays. It has been published that seeding activity by AD samples declines with chronological age of the donors and it is known that the misfolded tau isoforms in FTLD-MAPT cases with the same mutation differ in endpoints such as denaturation curves, protease-resistant fragments, etc. In the presented data the human FTLD-TDP and the FTLD-Tau samples used in upper and lower section of Fig. 6 are not even the same, implying intra disease variation in performance parameters is indeed manifest within the sample set. If the range of variation for samples of the same disease type - in biophysical isoforms, in normalized seeding activity - is equal or greater than the distinctions between samples of three disease types currently shown, then the data as currently presented in Fig. 6 is at least misleading, and possibly incorrect. The authors need to at least assay all their samples within each disease entity to get a better picture.

Minor points

The authors go back and forwards between using TDP 43 as an MBP fusion protein or cleaved with TEV protease. In some but not all experiments, MBP on its own is used as a control. But the MBP moiety seems to have an effect.

For consistency, Fig. 3 could have a diagram of the experimental design as per the 5 other experimental data figures.

Tau 441 versus tau650 could be spelled out better; 650 sounds like a construct but it is a fluorescent label. At first glance LD650-a-syn (p4) looks like a typo meant to say DL, but it is a different fluorophore. It is confusing to the reader - perhaps use a nomenclature such as subscripts to denote the fluorophore versus the construct lengths (e.g., tau 441), which are more important variable.

"Dormann Dormann" on page 1 - Typo?

Point-by-point reply to the reviewer's comments

We would like to thank the reviewers for their valuable comments. We are happy that reviewer #1 sees great potential impact to understand basic mechanisms of neurodegeneration and that reviewer #2 finds our pairing of biochemical and in cell data excellent and our findings exciting.

We have carefully addressed all specific points raised by the reviewers and are confident that we have addressed all of them, and even gone beyond, as outlined in detail in our point-by-point response below. Altogether, we have added many additional controls and added a substantial amount of new data. Specifically, we have:

- Characterized the dynamics of TDP-43/Tau co-condensates by FRAP showing differential effects on TDP-43 and Tau mobility compared to condensates of individual proteins (new Fig. 1E,F and Fig. 4C,D)
- Performed cross-linking mass spectrometry (XL-MS), indicating a direct interaction of the Tau proline-rich region with the N-terminal domain of TDP-43 (new Fig. 2 and Appendix Fig. S2)
- Performed SDD-AGE in the absence of the MBP tag (new Appendix Fig. S3), overall confirming the findings of tagged TDP-43
- Used an alternative way to promote Tau condensation (+RNA, no crowding agent) and captured early timepoints in the formation of the multiphasic Tau/TDP-43 co-condensates by time-lapse imaging (new Fig. EV2 and Appendix Fig. S4). This confirms that the promotion of Tau condensation does not depend on a molecular crowding agent, and that the small internal Tau condensates within multiphasic co-condensates likely form by early de-mixing of Tau and TDP-43.
- Used Tau variants/fragments and hyperphosphorylated Tau and various TDP-43 deletion mutants, showing that clustering of TDP-43 condensates requires the combination of N- and C-terminal domains of Tau and does not depend on the positive net charge of Tau alone (new Fig. EV1 and EV3).
- Performed immunogold-labelling EM using Tau and TDP-43-specific antibodies, demonstrating that the small oligomeric structures formed contain either Tau or a mixture of Tau and TDP-43 (new Fig. 5E,F)
- Demonstrated that the oligomeric Tau/TDP-43 assemblies observed by EM are Thioflavin T-positive, confirming their amyloid-like character (new Fig. 5 and Appendix Fig. S6).
- Included brain extracts from Parkinson's disease patients as an additional control for the Tau/TDP-43 seeding experiments with patient extracts (revised Fig. 7), confirming the specificity of Tau and TDP-43 seeding.
- Provided Western blots and IHC images for Tau and TDP-43 for all human brain samples used in revised Fig. 7 to demonstrate the aggregate pathology present in each case (new Fig. EV4, EV5 and Appendix Fig. S7/8. Moreover, double immunofluorescence reveals partial co-aggregation of TDP-43 and Tau (new Fig. EV5C).

These extensive new data are detailed in the point-by-point responses below. We are confident that these revised and new data will fully address the reviewers' concerns.

Referee #1:

Simonetti et al. present evidence for interactions between TDP-43 and Tau during protein condensation, aggregation and aggregate seeding activity. These findings are relevant to understanding the etiology of multiple neurodegenerative disorders, in which these two proteins play key roles. Growing evidence show co-occurrence of pathology due to TDP-43 and Tau inclusions. Based on their findings, Simonetti et al. propose that TDP-43-Tau interactions alter the properties of their cognate condensates as well as aggregation in vitro and aggregate seeding in cell-based reporters. Hence, these findings are of great potential impact to understanding basic mechanisms of neurodegeneration.

We thank the reviewer for this positive assessment of the potential impact of our work!

The work, however, lacks important controls and key assays to support the conclusions. In addition, the findings on the effect of TDP-43 on Tau aggregation and seeding activity are somewhat inconsistent. These important concerns reduce the impact of the studies.

Based on the insightful reviewer comments we have included further controls and provide additional data on both in vitro aggregation and seeding that fully address these concerns.

Major concerns:

- The experiments describing changes in condensate properties upon mixing Tau and TDP-43 should include FRAP assays to determine whether changes in dynamic properties cause the differences in condensate morphology reported in various figures: Fig 1B, C; Fig. EV1B, D.

We have performed the suggested FRAP experiments of TDP-43 condensates in presence of Tau or control proteins (new Fig. 1E, F), and of Tau condensates in presence of TDP-43 or control proteins (new Fig. 4C, D) to characterize changes in the dynamic properties of the formed condensates. Interestingly, these experiments show that Tau selectively reduces the mobility of TDP-43 within co-condensates (Fig. 1E, F), whereas TDP-43 increases the mobility of Tau (Fig. 4C-E). These findings provide important mechanistic insight into the altered morphology and behavior of co-condensates and further support the conclusion that Tau and TDP-43 mutually modulate each other's phase behavior in a distinct and directional manner.

- The assays throughout the manuscript should be consistently conducted in the absence of the C-terminal MBP tag for recombinant TDP-43, full length and fragments. Some experiments, e.g., condensate formation in Fig 1, were performed after cleaving off the MBP tag. On the other hand, others, such as Figs 2, 4, and 5 used full-length or fragment TDP-43-MBP. One of the reasons for this concern is the previously established effect of C-terminally tagged TDP-43 using MBP on the condensation properties of the protein, suggesting that MBP alters the behavior of the disordered C-tail. Thus, MBP may alter TDP-43 aggregation and condensation properties.

We appreciate the reviewer's important point regarding the potential influence of the MBP tag on TDP-43 condensation and aggregation. The MBP fusion is indeed critical to maintain TDP-43 soluble during expression and purification, and we cleave the C-terminal MBP-tag using TEV protease to initiate controlled TDP-43 condensation or aggregation (as for example in Fig. 1 and 3d-f). Upon cleavage of the solubility tag, TDP-43 immediately starts to form condensates above $\sim 1 \mu\text{M}$ concentration. Thus, we cannot study the impact of the soluble, monomeric TDP-43 on Tau without the MBP tag.

To analyse the impact of soluble, monomeric TDP-43 on Tau condensate/fibril formation (Fig. 4 and 5), the MBP tag must be retained. Cleaving the MBP tag in these assays would lead to immediate TDP-43 condensation or aggregation, thereby confounding interpretation of its role in modulating Tau. Indeed, prior attempts to perform these experiments with TEV cleavage led to uncontrolled TDP-43 aggregation, precluding their use under such conditions.

To directly address the reviewer's concern about aggregation behavior, we have repeated the SDD-AGE assay under tag-free conditions. As expected, cleaving the MBP-tag accelerated

oligomerization/aggregation. The new results (shown in new Appendix Fig. S3) are completely in line with the SDD-AGE results obtained under non-cleaved conditions (Fig. 3a-c).

However, slower aggregation of tag TDP-43 allows us to better visualize the individual bands (Fig. 3a-c and Gruijs da Silva, EMBO J 2021). We now clarify this rationale in the revised manuscript to ensure transparency and reproducibility.

- In addition, Fig 3 shows strong colocalization of MBP alone with Tau. The authors should discuss this observation and the impact of this behavior on the other results that involve MBP-tagged proteins.

Indeed, the image shown in old Fig. 3 showed mild colocalization of Tau with MBP, which was further emphasized by the close-up showing one especially bright red droplet. To address this, we have replaced the image with a more representative image which now shows no MBP co-localization with Tau droplet (new Fig. 4). Importantly, the key phenotypes (enhanced Tau condensation and formation of small, internal Tau droplets) are only seen in the presence of TDP-43-MBP, and not MBP or α -synuclein, suggesting that the effect is primarily driven by the TDP-43 protein itself.

- Condensation assays in Fig 3 and EV should be conducted in the absence of PEG and heparin as the presence of these coacervate promoters could diminish the significance of the observations.

We appreciate this important point and agree that it is essential to determine whether the observed effects persist under more physiologically relevant conditions, i.e., without artificial crowding agents or strong polyanions like heparin. It is reported that Tau condensation requires the use of 5-10% PEG or negatively charged molecules, such as heparin or RNA under low salt conditions (Hochmair et al., EMBO J 2022, Kanaan et al., Nat Comm 2020). We now have used U20-RNA in low salt buffer as alternative way to promote Tau condensation (shown in new Appendix Fig. S4). Under these conditions, we detected amorphous, aggregate-like Tau structures that were strongly promoted by the presence of TDP-43-MBP, but not by MBP or α -synuclein (Appendix Fig. S4B). These results further support the idea that the influence of TDP-43 on Tau condensation is specific and also occurs independently of crowding agents. This further supports our conclusion that TDP-43 directly modulates Tau condensation through protein-protein interaction (new Fig. 2 and Appendix Fig. S2), not via nonspecific effects of experimental additives.

- The authors should better define the results in Fig 3C as it is difficult to make conclusions on the behavior of TDP-43-MBP in the context of the Tau assemblies, especially at 4 and 24h time points.

We agree that image quality was not ideal in Fig. 3C particularly at later time points. To address this, we have repeated the experiment to obtain higher quality images (on a new Zeiss Axio Observer.Z1/7 confocal microscope) (now shown in revised Fig. 4B). To further clarify the molecular basis of TDP-43's influence on Tau condensation, we also expanded the analysis by including a panel of TDP-43 mutants (that lack certain domains or are unable to phase separate) and have compared them to TDP-43 wild-type, to gain further molecular insights into what drives the promotion of Tau phase separation and multiphasic condensates (new Fig. EV3). The results reveal that only full-length TDP-43 robustly induces the characteristic multiphasic phenotype, suggesting that multivalent interactions across multiple domains are necessary for this effect.

These data not only resolve the image quality issue but also provide deeper mechanistic insight into how specific regions of TDP-43 contribute to Tau phase separation and compartmentalization within co-condensates.

- Based on the findings in Fig 4, the authors suggest that full length TDP-43 suppresses Tau fibrilization, whereas the CTD-MBP construct does not significantly alter fibril length. This is in contrast with findings in Fig 5B, where CTD-MBP appears to reduce Tau aggregate seeding in the reporter cell line. The authors should discuss these seemingly contradictory observations.

We thank the reviewer for highlighting this important point. To address the apparent discrepancy between the fibrillization and seeding data with CTF-MBP, we have repeated both the EM and Tau biosensor cell experiments using newly prepared, independently validated protein batches and multiple biological replicates. These updated and expanded results are now presented in the revised Fig. 5.

The revised data consistently show that full-length TDP-43 and its N-terminal fragment (NTF) robustly suppress Tau fibril formation and these samples also completely abolish seeding in the Tau biosensor cell line. In contrast, the C-terminal fragment (CTF) does not affect Tau fibrillization, and Tau fibrils formed in the presence of CTF retain full seeding competence, comparable to Tau alone. These new results resolve the previously conflicting observations and clearly demonstrate that only full-length TDP-43 and its N-terminal fragment (aa. 1-273) modulate Tau aggregation and seeding.

- The experiments in Fig 6, studying seeding activity of patient-derived extracts for Tau and TDP-43 should include immunodepletion experiments to remove any presence of Tau from FTLD-TDP and TDP-43 from AD-derived samples. This control will help determine whether seeding activity is caused by cross-seeding or undetectable amounts of TDP-43/Tau in the patient-derived samples.

We appreciate the reviewer's suggestion and fully agree that ruling out potential cross-seeding effects is critical for correctly interpreting seeding assay results.

After careful re-analysis of our SarkoSpin extraction and seeding procedure, we have changed the way how we normalize input amounts of the SarkoSpin material. Previously, protein concentration was measured using the BCA assay after pellet resuspension. However, we found that this approach is highly subjective to technical variabilities, such as the amount of residual sarkosyl, resuspension and pipetting errors. In the revised manuscript, total protein concentration is assessed after tissue homogenization (prior to pelleting) and normalized against the Non-ND control.

In addition to this technical refinement, we increased the number of patient samples (3-4 per group), stratified AD patients into a Tau/TDP-43-positive (AD+) and Tau-only (AD-) group, and added Parkinson's disease (PD) patients as additional specificity control. In addition, we performed multiple independent replicate experiments to account for biological variability. These improvements have resulted in a **revised dataset** (now presented in **revised Fig. 7**) that leads to an updated conclusion: FTLD-TDP brain extracts now consistently show only background-level seeding in Tau biosensor cells, eliminating the concern of cross-seeding from TDP-43 to Tau (revised Figure 7B/C). Likewise, TDP-43 negative AD cases do not seed TDP-43. Since we now do no longer observe potential "cross-seeding". Therefore, immunodepletion experiments are no longer required to clarify the source of the observed seeding activity. We have updated the manuscript to reflect this new interpretation and clarified the methodological changes made to improve assay consistency.

- The aggregates in Figs 5 and 6 should be quantified in terms of area instead of number of aggregates per cell.

We now have quantified both the number and area of aggregates in the Tau and TDP-43 seeding assays with patient extracts and report both quantifications (new Fig. 7). Both metrics show consistent trends and support the conclusions described in the main text.

Referee #2:

This manuscript probes how direct interaction between TDP-43 and tau alters their phase separation and aggregation in vitro and in cells. Tau and TDP-43 enhance each other's phase separation into solid type aggregates. While tau enhances aggregation of TDP-43, TDP-43 prevents formation of regular fibrils of tau. This effect is demonstrated to arise from the N-terminal domains of TDP-43, not the C-terminal domain. Excitingly, brain material from FTLTD with TDP-43 aggregates induces tau aggregation in cell model detecting tau seeds, while brain material from AD (with tau aggregates) induces TDP-43 aggregation in cells detecting TDP-43 seeds. The pairing of the biochemical and in cell data is excellent and makes a strong case. The mechanism could be addressed further, with additional experiments, to test the robustness of the findings and bring a means to rationalize the exciting findings.

We thank the referee for the many positive comments regarding our work and will provide additional experiments to make them more robust and better rationalize our findings.

Major

- Throughout the biochemical sections, MBP and alpha synuclein are used as controls. This is a good idea, but does not provide a handle on why. The authors should perform experiments with sequence scrambled tau, mutant forms of tau and TDP-43, etc to answer the question what is it about tau? Is it just the net charge (how different is this than the other proteins?) as tau and TDP-43 are both known to be altered by nucleic acid interactions. The lack of detail here decreases the insight - the experiments demonstrate that tau does these things to TDP-43. But, although it is within reach, the experiments do little to explain why. The authors provide some efforts in one section by dividing TDP-43 into two pieces - however they term the N terminal domain something that encompasses a dimerizing/oligomerizing domain and then RNA binding domains. Which of the domains is important or is it both the first domain and the RNA binding domains. Many mutants altering the behavior of these domains (assembly, RNA binding, folding) including disease mutations have been described in the literature.

We thank the reviewer for this insightful and constructive critique. We fully agree that using additional Tau and TDP-43 variants will provide valuable mechanistic insights. In response, we have significantly expanded our domain-mapping approach and now provide a more detailed and systematic analysis using both Tau and TDP-43 variants (see new Figs. EV1 and EV3).

In the revised manuscript on p.4, we have now included information on the net charge of Tau (+1.44 at pH 7.4) vs. TDP-43 and the control proteins, which all have a negative net charge at pH 7.4 (TDP-43: -5.27; MBP-Tev-His: -13.29; synuclein: -9.73). Among the tested proteins only Tau has a mildly positive net charge, raising the question of whether charge underlies its effect on TDP-43

To gain further mechanistic insights we have performed experiments with the following Tau variants containing different Tau functional domains / having different net charges (new Fig. EV1):

- Tau N-terminal domain (aa 1-256; includes the proline-rich domain (PRD) that drives Tau phase separation (Zhang et al., J Cell Biol 2020); net charge at pH 7.4 = -7.6)
- Tau-repeat domain (RD) (aa 244-372; R1-R4; microtubule binding region and core part of Tau fibrillar aggregates; net charge at pH 7.4 = +9.1) (Wille et al., J Cell Biol 1992).
- In vitro phosphorylated Tau (net charge at pH 7.4 ca. -5.56 (Chakraborty et al., 2024)).

These experiments revealed that only full-length (including pTau), but not isolated PRD or RD fragments, modulate TDP-43 condensation and aggregation, indicating that multivalent interactions across the entire Tau molecule are required. Notably, the effect does not correlate simply with net charge, as both negatively charged pTau and positively charged RD failed to replicate the full-length Tau effect on their own.

To gain further insights into which domains/features of TDP-43 are relevant for the observed effects on Tau phase separation / fibrilization, we have performed experiments using an extended panel of TDP-43 deletion constructs and domain mutants (new Fig. EV3):

- TDP-43 N-terminal fragment (NTF, aa. 1-266)
- C-terminal fragment (CTF, aa. 267 - 414)
- TDP-43-tandem RRM domains (aa. 102-270)
- TDP-43 lacking the N-terminal dimerizing/oligomerizing domain + NLS region (Δ aa. 1-101)
- TDP-43 lacking the conserved α -helical region in the CTD (aa. 321-340), which is necessary for TDP-43 phase separation (Conicella et al., PNAS 2020)

These experiments demonstrate that only full-length TDP-43 robustly promotes multiphasic Tau condensates. While some partial activity is seen with the NTF, no single domain is sufficient, supporting a model where multivalent, cooperative interactions across multiple domains are required to alter phase separation of Tau.

Finally, we have performed cross-linking mass spectrometry (XL-MS) of TDP-43/Tau mixtures under TDP-43 condensation conditions, indicating a direct interaction of the Tau proline-rich domain with the N-terminal domain of TDP-43 (new Fig. 2).

Taken together, these new experiments provide mechanistic insight into why Tau selectively modulates TDP-43 behavior, identifying domain-specific and multivalent interactions, rather than simple effects of net charge or non-specific crowding. We have clarified this interpretation in the revised manuscript.

- Pg 3,4 Can the authors show data that the labeling does not affect the results. If unlabeled tau (or TDP-43) is used, do the authors find that TDP-43 is still altered?

This is indeed a critical point. To ensure that fluorescent labelling does not influence our findings, we had already previously used unlabelled proteins in key assays and have further extended use of this important control in the revised manuscript. For the phase separation assays, Fig. 1c vs. 1b already shows that unlabelled Tau has similar effects (clustering of TDP-43 condensates) as labelled Tau. For the aggregation assays in Fig. 3E, we also used completely unlabelled Tau protein. We highlighted this in the revised text.

For the Tau condensation assay (now shown in revised Fig. 4), we have additionally performed a control experiment with unlabelled TDP-43 (new Appendix Fig. S5A), confirming that unlabelled TDP-43 also promotes formation of multiphasic Tau condensates to a similar degree as fluorescently labelled TDP-43 does. These controls are clearly highlighted in the revised manuscript to avoid any ambiguity regarding potential labelling artefacts.

- The description on page 5 that condensation "gradually formed small condensates over time" is surprising in that phase separation is typically instantaneous unless something else is happening (aggregation, oxidation, etc). The authors should explain what is happening and more clearly show the progression.

We appreciate the reviewer's comment and the opportunity to clarify this point. While phase separation/protein condensation can occur rapidly under optimal conditions, it is still a progressive process, in which the number/size of condensates increases over time (Oswald ripening). Depending on how well the assay conditions (i.e., buffer composition, crowding, protein concentration etc.) promote condensation of the respective protein(s), this process can take seconds or hours. For Tau, condensation is largely driven by electrostatic interactions (Boyko et al., 2019). Accordingly, Tau condensates form almost instantly at low salt conditions with or even without crowding agents, whereas ion concentrations of ~150 mM (used here in the assay) reduce Tau's condensation ability and require the presence of molecular crowding agents (see e.g. Hochmair et al., EMBO J 2022, Ambadipudi et al. Nat Comm 2017, Kanaan et al. Nat Comm 2020). When adding binding partners that promote Tau condensation, like RNA (Hochmair et al., EMBO J 2022) or here TDP-43, Tau condensation becomes more efficient even at 150 mM ion concentration, which can be observed as faster condensate formation / larger Tau condensates.

To better visualize this process, we have now included a time-course series capturing early (10 min), intermediate (2 h), and late (24 h) time points of Tau/TDP-43 co-condensation. These data are shown in revised Fig. EV2 and clearly illustrate the progressive evolution of multiphasic co-condensates, including internal droplet formation.

- Pg 5 The description of "intra-tau" condensates is vague and not descriptive of what is observed - intra to what? And, although the authors say it "remains to be determined" how these intratau condensates are formed, the authors should try to observe this by microscopy as a function of time.

We have rephrased our observations on the formation of multiphasic Tau-TDP-43 co-condensates more precisely and now don't use the ambiguous term "intra-Tau" condensates anymore. We now give a much more detailed description of the process, by capturing very early timepoints in the formation of these multiphasic Tau/TDP-43 co-condensates by time-lapse imaging. Strikingly, co-condensed, multiphasic structures already formed after 10 min (new Fig. EV2). We also provide high resolution images of the 2h and 24h timepoints to provide a better description of these multiphasic structures. These images suggest that intra-condensate Tau-assemblies are enclosed within the TDP-43 condensate at 2 h and persisted until 24 h.

- Pg 5 "prevents the formation of tau fibrils". Are these shorter aggregates ThT positive? How do these change other metrics of fibril formation / amyloid formation? This should be tested. The observation should also be more precisely explained, detailing that the progression to mature/long fibrils is inhibited. Whether these fibrils contain the other protein should be checked by immunogold EM or by TIRF using fluorescent proteins already used in the manuscript. These findings may also help answer the potential apparent contradiction that fibril formation is halted but these are co-implicated in aggregation disease.

We thank the reviewer for these excellent suggestions, which have led to several informative new experiments providing a more nuanced understanding of the assemblies. We have performed Thioflavin-T (ThT) incorporation assays under Tau fibrillization conditions in the presence of different TDP-43 variants (new Fig. 5A, B). These experiments show that ThT incorporation is significantly reduced for Tau/TDP-43-MBP mixtures. Residual ThT incorporation follows a slower kinetics than seen for Tau alone or Tau in presence of the control proteins. This suggests that the Tau/TDP-43 assemblies have partial β -sheet character, consistent with early-stage or aberrant fibrillar structures.

The requested time course experiment (showing that the progression to mature/long Tau fibrils is inhibited already on day 1 and 2) is shown in Appendix Fig. S6.

Finally, we have performed immunogold-labelling EM using antibodies specific for Tau and TDP-43, which demonstrates that the small oligomeric structures formed by mixtures of Tau and TDP-43-MBP are either Tau-positive or Tau/TDP-43 double-positive (new Fig. 5 E,F). Together, these new results support a refined conclusion: full-length TDP-43 suppresses the progression of Tau into mature fibrils, instead stabilizing smaller, ThT-positive, oligomeric assemblies, some of which contain both proteins. These findings align with the broader hypothesis that non-fibrillar Tau/TDP-43 complexes may represent alternative, possibly toxic species relevant to disease pathology despite the absence of classical amyloid fibrils.

- Presenting biological replicates of Figure 5 C and 6 C, F would be helpful - at least a statement regarding the replicability with different batches of cells and different purifications of proteins.

We fully agree with the reviewer that demonstrating reproducibility is essential. In response, we have replicated critical experiments and improved documentation for the relevant figures:

We have repeated the seeding experiment in the Tau biosensor cells with recombinant fibril preparations (previous Fig. 5C, now new Fig. 6C-E) with n=5 times. Each replicate was performed with different batches of cells on different days and with different batches of purified proteins (as now clearly stated in the figure legend and the methods section of the manuscript (p. 15)).

For seeding experiments with human brain material (previously Fig 6, now improved Fig. 7), we have also added 3-4 more patient samples per group and state that independent experiments were performed with different batches of cells on different days ($n \geq 3$, as stated in the revised legend). To further support the overall reproducibility of the phenotype, despite high patient-to-patient variability, we have included additional representative images for all samples shown in Fig. 7C and F to the supplemental material (new Appendix Fig. S8).

Taken together, these additions support the robustness of our observations across independent replicates and biological sources.

- The data on the cell seeding assays are really exciting. I think they are excellent. However, I think more controls should be shown/performed to understand if this is specific. First, some negative controls showing that Parkinson's or other type of neurodegeneration should be shown to show if these results are specific. Perhaps those exist in the literature but it would be good to show that here. Secondly and complementarily, the suggestion is that seeds of one protein type are present in the diseases of the other - but the rest of the manuscript is saying that the one protein induces the aggregation or changes the aggregation of the other. So it is not that clear if these seeds are really present. Western blots and experiments in the cells using immunodepleted samples should be added to confirm if these things are present and what they do. If the authors are concerned about co-aggregates in immunodepletion, the authors could show co-ip Westerns then.

We thank the reviewer for the positive comments and the helpful suggestions. We have included extracts from Parkinson's disease (PD) patients as an additional negative control and control for specificity. This shows that PD extracts do not seed in the Tau biosensor cells (revised Fig. 7A-C) or the TDP-43 seeding assay (revised Fig. 7D-F), underlining the specificity of these assays.

Moreover, we have also carefully analysed the SarkoSpin-extracted brain material by Western blots with antibodies specific to Tau, phosphoS409/S410-TDP-43, TDP-43 and α -synuclein (new Figure EV4), to demonstrate which proteins are detectable in the different patient samples.

We have also stratified AD patients into a Tau/TDP-43-positive (AD+) and Tau-only (AD-) group, which gave very interesting results and shows that brains from AD patient with Tau/TDP-43 co-pathology exhibited lower Tau seeding capacity than AD patients with Tau-only pathology (new Fig. 7A-C), despite similar levels of Tau present in both groups (new Fig. EV4). This result is in line with our results from the Tau seeding assay with recombinant Tau/TDP-43 samples, which showed that Tau seeding is strongly reduced by presence of TDP-43 (new Fig. 6).

Based on our revised protocol and expanded patient cohort, we now observe clear separation of seeding activity by disease group, and no longer detect potential cross-seeding signals (e.g., TDP-43 seeds in AD or Tau seeds in FTLTDP). Therefore, immunodepletion is no longer essential to clarify the presence of specific seeds.

Minor points that are easily addressed in text:

All these minor points have been addressed by text/figure revisions.

- Pg2 "more neurotoxic than solid aggregates". Solid aggregates should be replaced with something more specific, such as amyloid fibrils.
We now say "fibrous, amyloid-like Tau aggregates".
- Pg2 although it is a new term not favoured by all, the term "LATE" should be used in my mind to describe the subset of AD with TDP-43.
LATE is a TDP-43 proteinopathy that often occurs independently of AD pathology but can also occur alongside it. In the revised manuscript, we explicitly mention this when introducing the term "AD+" to indicate cases of AD with documented TDP-43 co-pathology consistent with limbic predominant age-related TDP-43 encephalopathy neuropathological changes (LATE-NC). We believe this notation is more accessible to non-specialist readers, while still conveying the necessary distinction.

- Pg 4 "same aggregation assay ... cleaving off the MBP". It would be more clear to explain when TEV is added (and add to the schematic).
As indicated in the cartoon in Fig. 3D, TEV cleavage is added at the beginning of the reaction, before shaking for 30 min. We also made sure to clearly explain this in the text (p. 6).
- Is 150 mM NaCl really physiological for intracellular proteins. May be better to say this is to mimic more physiological conditions rather than say it is physiological.
We rephrased and now say "near physiological", since it's similar to the physiological 140 mM KCl of the cytoplasm.
- Fig 3 It is somewhat confusing to start Fig 3 with turbidity as it is not clear if there are changes in liquid vs solid qualitatively before showing what is quantitatively happening.
We agree the turbidity assay was confusing and didn't add much, so we have removed it from the manuscript.
- Fig 3C controls - the authors should comment about MBP recruitment into the condensates, as usually it is assumed it is not recruited into condensates.
As explained above under the comments to reviewer 1, the image shown in old Fig. 3 showed mild colocalization of Tau with MBP, which was further emphasized by the close-up showing one especially bright red droplet. To address this, we have replaced the image with a more representative image which now shows no MBP co-localization with Tau droplet (new Fig. 4). Importantly, the key phenotypes (enhanced Tau condensation and formation of small, internal Tau droplets) are only seen in the presence of TDP-43-MBP, and not MBP or α -synuclein, suggesting that the effect is primarily driven by the TDP-43 protein itself.
- Pg 5 The end of the section "Under molecular crowding conditions" could use a summary to keep the reader on track as to what is concluded here.
We have added a summary sentence to each section/paragraph.
- More details on the synuclein preparation by heat inactivation times and lysis conditions should be added on pg 11. Explanation of the differences in preparation and definition of "afterwards labeled for the experiments" should be added as this is unclear.
These details have been added to the methods section (p. 15).
- Where is TDP-43 labeled by maleimide dyes and does that affect its stability?
TDP-43 is labeled using maleimide-conjugated fluorescent dyes, which covalently bind to cysteine residues in the protein. We have not noted adverse effects of dye conjugation on protein stability or phase separation. We included various experiments where we compared unlabelled and labelled protein, to exclude that effects are coming from dye conjugation (Fig. 1, Appendix Figure S1, Fig. 3, Appendix Fig. S5).
- Pg 15 - the term "101.41 scaled resolution" is that supposed to field of view or resolution?
The description has been updated to image pixel size to reflect a more standard way of reporting microscopy parameters.

Referee #3:

Extrapolating from mixed proteinopathies, this paper assesses interference effects for seeding between Tau and TDP43. For the most part the authors use in vitro or cell-free assays, including studies of phase separation on glass sides, but there are some experiments in HEK293 reporter cells. Endpoints for phase separation phenomena (production of condensates) include the number or particles, roundness, and area per particle, while in the reagent permutations that favor fibril formation, aggregation endpoints are studied by agarose gel electrophoresis and EM. Alpha-syn comprises an internal control with weaker or absent stimulatory effects, depending on the assay. Domain effects are appraised by using N- or C-terminal domain recombinant proteins in the aggregation assays. In the final assays, aggregation effects are followed subsequent to initiation with brain samples from different FTLD and AD patients.

The main observation in the earlier figures is a stimulatory effect of tau on assembly of recTDP43 to generate larger, irregular condensates or increased aggregation. Then, the reverse effect of TDP43 upon tau is assessed with Fig.3, apart from turbidity, displaying qualitative changes in condensates and Fibril formation, Fig. 4. Here an inhibitory effect is scored and mapped to the N-terminal domain. Switching to a FRET biosensor cell assay based on the tau RD linked to two fluorophores and primed by incubated recombinant proteins (Fig. 5), the greatest number of fluorescent foci per cell is obtained with tau plus a TDP43 N-terminal domain. This is confusing, at least in an initial reading, but is attributed to the heightened presence of seed competent oligomers in the samples. From the perspective of experimental variables between the assays, the tau substrate in the cells differs in length from that used in Figs 1-4 not only in length but also in allelic type, by having a P301S FTLD-tau mutation. Fig. 6 uses the same biosensor cells primed with human brain homogenates processed to enrich for sarkosyl insoluble species (at least probably - below). Notably FTLD-TDP can give a strong signal and given the selectivity of the cells, this might be attributed to tau aggregates in these patients. In a reversed assay looking at TDP43 aggregation, AD can give a strong signal. But there are caveats about case selection that suggest that the data in Fig. 6 need to be treated with caution (below).

The authors deduce that there are counterbalancing effects, such as increased phase separation can diminish fibril formation.

We thank the reviewer for this comprehensive and insightful summary, and for identifying areas that benefit from clarification. In the point-by-point response, we address the apparent discrepancies and outline the experimental rationale behind the system-specific findings.

Main points.

- The paper is not an easy read, and this is nothing to do with language usage (which is fluent). Rather, it is because at various points the paper deals with phase separated condensates, fibrils, sarkosyl-insoluble extracts and putative oligomer preps, which may possess intrinsic opposing properties or may have apparently contradictory properties due to limitations in the assays or from interconversion between biophysical forms.

We thank the reviewer for this important observation. We fully recognize that working across multiple biophysical states of Tau and TDP-43, including phase-separated condensates, fibrils, oligomers, and Sarkosyl-insoluble material, is inherently complex. We have put in a lot of effort to make the text of the revised manuscript more accessible. We now also provide a much more informative schematic graphical abstract (instead of previous summary Figure 7, see next point), which now summarizes the main experimental approaches and main results obtained. We have more openly discussed the limitations and remaining questions in the revised manuscript, e.g.: (i) the molecular nature of the patient brain-derived TDP-43 seeds remains to be determined (discussion, p. 12) ; (ii) Studies in more sensitive neuronal

seeding platforms (Rummens et al., 2025; Scialò et al., 2025) and *in vivo* models will be required to test the impact of the formed assemblies on the neurodegenerative phenotype (see discussion on p. 11); (iii) the inherent high complexity and variability of the SarkoSpin extraction method, as noted in previous studies (De Rossi et al., 2021; Laferrière et al., 2019; Scialò et al., 2025) (see discussion, p. 12).

- Fig. 7 may have been intended to tie the interpretation of all the results together, but to this reviewer it was not helpful. For example, would the upper panel benefit from a comparison with a pure tauopathy? The lower panel emphasizes AD seeding TDP43 aggregates, but AD more famously has seeding by Abeta and tau species, which are themselves abundant and which are not shown in the diagram. So, to this reader, the paper often comes across as an imperfect collage of raw results. The assays in Fig.6 may have been intended as a capstone experiment for the lab data, to bring the picture together (more below), but the brain samples used are not profiled for their constituent molecules by IHC in any display items. This does beg the question of what the co-deposits look like *in situ* for these patients, noting that co-aggregates are underscored by the authors in the concluding sentence of the Introduction.

We thank the reviewer for these thoughtful and constructive observations. We have carefully revised Fig. 7 and turned it into a much more telling graphical abstract, comparing pure Tau-only pathology (AD-) with mixed Tau/TDP-43 pathology (AD+), as suggested by the reviewer. We did not include A β explicitly in the diagram because it was not studied in this work and rather focused on the Tau/TDP-43 interaction.

In response to the concern about insufficient characterization of patient samples we now provide IHC images for Tau and TDP-43 for all disease groups used in the seeding assays in the revised Fig. 7 (see IHCs in new Fig. EV5 and Appendix Fig. S7). These example images now show the protein pathology in the brain regions used in our seeding experiments and complement the comprehensive summary of all used cases given in Table 1. We also show what the co-deposits look like *in situ* by showing double-fluorescent immunostaining for Tau/TDP-43 in hippocampus of two different AD+ cases (new EV5C). We are confident that these revisions significantly enhance the clarity, integration, and potential clinical relevance of the manuscript.

- The authors refer to "Sarkospin" extracts without explaining at the outset what they enrich for. Presumably sarkosyl-insoluble materials, which are mainly? - mature fibrils? But then again oligomers are repeatedly mentioned as having superior seeding activity. What is the status of oligomers in these extracts? In other words, to what extent is this Sarkospin procedure biasing the assay outputs and hence the conclusions. What happens when brain extracts are processed by a different procedure before a biosensor assay?

We thank the reviewer for highlighting this important point regarding the biochemical nature of the SarkoSpin extracts and their influence on seeding readouts. The SarkoSpin extract procedure has been thoroughly detailed and characterized in a previous publication (Laferrière et al., Nature Neuroscience, 2019), including a proteomic description of what proteins are extracted by this procedure. This analysis has shown that TDP-43 is significantly enriched in these extracts, while traces of other insoluble proteins are also detectable. This study also showed that the SarkoSpin pellets contain fibrils of TDP-43, as was evident by transmission electron microscopy and immune-gold labeling of the extracts. Sonication of the SarkoSpin pellets prior to incubation with the cells is used to generate smaller fragments, as is the standard procedure for other patient-derived protein fibrils. To our knowledge, no available protocol can currently consistently isolate or detect pathological TDP-43 oligomers within the preparation, especially if these are soluble oligomers. Compared to other isolation procedures, the SarkoSpin procedure consistently isolates high amounts of TDP-43 seeds, with minimal amounts of co-precipitating normal TDP-43 (Laferrière et al, 2019). As such, it is widely used and currently best suited for comparative seeding studies.

We now explain that “the molecular nature of the patient brain-derived TDP-43 seeds remains to be determined” (discussion, p. 12) and that SarkoSpin extracts are known to contain fibrils, but is it not known if they contain oligomers or other types of TDP-43 assemblies (Laferrière et al, 2019).

Each disease type in Fig.6 is represented by one bio sample, even though it is known that well characterized disease entities (defined, for example, by a shared germline mutation or extensive pathology profiling) still have a range of activities in seeding and biochemical assays. It has been published that seeding activity by AD samples declines with chronological age of the donors and it is known that the misfolded tau isoforms in FTLD-MAPT cases with the same mutation differ in endpoints such as denaturation curves, protease-resistant fragments, etc. In the presented data the human FTLD-TDP and the FTLD-Tau samples used in upper and lower section of Fig. 6 are not even the same, implying intra disease variation in performance parameters is indeed manifest within the sample set. If the range of variation for samples of the same disease type - in biophysical isoforms, in normalized seeding activity - is equal or greater than the distinctions between samples of three disease types currently shown, then the data as currently presented in Fig. 6 is at least misleading, and possibly incorrect. The authors need to at least assay all their samples within each disease entity to get a better picture.

We thank the reviewer for raising this critical point regarding inter-patient variability, which is indeed a known and important factor in studies involving human brain tissue. In response, we have included additional cases in each group (see extended Table 1). While we show representative images from a single patient per disease entity in Figure 7 (revised version of previous Fig. 6), we have used bio sample from 3-4 patients from each disease entity and show the corresponding images of all these cases in Appendix Fig. S8 to be fully transparent about the within-group patient-to-patient variability in our assays. We also added Western blot of all brain extracts for Tau, phosphoS409/S410-TDP-43, TDP-43 and α -synuclein (new Figure EV4) to characterize the relevant insoluble proteins in each sample.

For the revision experiments we have also refined our experimental procedure to reduce variability. After careful re-analysis of our SarkoSpin extraction and seeding procedure, we have changed the way how we normalize input amounts of the SarkoSpin material. Previously, protein concentration was measured using the BCA assay after pellet resuspension. However, we found that this approach is highly subjective to technical variabilities, such as the amount of residual sarkosyl, resuspension and pipetting errors. In the revised manuscript, total protein concentration is assessed after tissue homogenization (prior to pelleting) and normalized against the Non-ND control.

In addition to this technical refinement, we increased the number of patient samples (3-4 per group), stratified AD patients into a Tau/TDP-43-positive (AD+) and Tau-only (AD-) group, and added Parkinson's disease (PD) patients as additional specificity control. In addition, we performed multiple independent replicate experiments to account for biological variability. These improvements have resulted in a **revised dataset** (now presented in **revised Fig. 7**) that leads to an updated conclusion: FTLD-TDP brain extracts now consistently show only background-level seeding in Tau biosensor cells, eliminating the concern of cross-seeding from TDP-43 to Tau (revised Figure 7B/C). Likewise, TDP-43 negative AD cases do not seed TDP-43. We have updated the manuscript to reflect this new interpretation and clarified the methodological changes made to improve assay consistency.

Moreover, we have explicitly commented on this variability between different donors (discussion, p. 12) and specify the number of biological samples and independent experiments (see legend of Fig. 7).

Together, this now provides a much more comprehensive characterization of the patient material used in our seeding experiments, and we discuss the caveats that come with the patient-to-patient variability. Together, we hope that these data will address the concerns raised by this reviewer.

Minor points

- The authors go back and forwards between using TDP 43 as an MBP fusion protein or cleaved with TEV protease. In some but not all experiments, MBP on its own is used as a control. But the MBP moiety seems to have an effect.

We appreciate the reviewer's important point regarding the potential influence of the MBP tag on TDP-43 condensation and aggregation, which was also raised by reviewer 1. The MBP fusion is indeed critical to maintain TDP-43 soluble during expression and purification, and we cleave the C-terminal MBP-tag

using TEV protease to initiate controlled TDP-43 condensation or aggregation (as for example in Fig. 1 and 3d-f). Upon cleavage of the solubility tag, TDP-43 immediately starts to form condensates above ~ 1 μ M concentration. Thus, we cannot study the impact of the soluble, monomeric TDP-43 on Tau without the MBP tag.

To analyse the impact of soluble, monomeric TDP-43 on Tau condensate/fibril formation (Fig. 4 and 5), the MBP tag must be retained. Cleaving the MBP tag in these assays would lead to immediate TDP-43 condensation or aggregation, thereby confounding interpretation of its role in modulating Tau. Indeed, prior attempts to perform these experiments with TEV cleavage led to uncontrolled TDP-43 aggregation, precluding their use under such conditions.

To directly address the reviewer's concern about aggregation behavior, we have repeated the SDD-AGE assay under tag-free conditions. As expected, cleaving the MBP-tag accelerated oligomerization/aggregation. The new results (shown in new Appendix Fig. S3) are completely in line with the SDD-AGE results obtained under non-cleaved conditions (Fig. 3a-c).

However, slower aggregation of tag TDP-43 allows us to better visualize the individual bands (Fig. 3a-c and Gruijs da Silva, EMBO J 2021). We now clarify this rationale in the revised manuscript to ensure transparency and reproducibility.

- For consistency, Fig. 3 could have a diagram of the experimental design as per the 5 other experimental data figures.

We agree and have added such a diagram to the revised Fig. 3 (now Fig. 4A).

- Tau 441 versus tau650 could be spelled out better; 650 sounds like a construct but it is a fluorescent label. At first glance LD650-a-syn (p4) looks like a typo meant to say DL, but it is a different fluorophore. It is confusing to the reader - perhaps use a nomenclature such as subscripts to denote the fluorophore versus the construct lengths (e.g., tau 441), which are more important variable.

We apologize that our nomenclature was confusing. We thank the reviewer for pointing this out and have now revised our nomenclature, so that it becomes clear what the numbers mean (fluorophore vs. construct length) and have now consistently spelled out dye names, incl. in all figures. We hope this significantly improves readability and removes any ambiguity between dye labeling and protein identity.

- "Dormann Dormann" on page 1 - Typo?

Indeed, this was a typo, thanks for pointing it out!

Dear Dorothee, dear Eddie,

Thank you again for submitting your revised manuscript (EMBOJ-2024-118442R) to The EMBO Journal for our consideration, and for your patience during peer review. Your manuscript has been sent back to the three original referees who had previously assessed the first version of your manuscript, and we have now received their comments, which you can find below.

I am very pleased to say that all three referees acknowledge that the manuscript has been significantly strengthened by the addition of new data and analyses, as well as by appropriate textual revisions that, together, sufficiently address the majority of the initially raised concerns. All three referees are supportive of the publication of this work in The EMBO Journal once a list of rather minor remaining concerns have been successfully addressed in a final version of your manuscript. We have reviewed the remaining comments in our team and find them reasonable and constructive; as you will see, most of them can be fully addressed by further textual clarification/discussion. Please submit along with your revised manuscript a detailed point-by-point response detailing any changes to the manuscript.

From the editorial side, there are also a few changes and corrections we need you to make in the final version of the manuscript before we can proceed with its acceptance for publication:

- Regarding the funding information entered in our online manuscript handling system, please add all funders to the "More Funders" list rather than to the Comments box.
- Please reduce the number of keywords (after the Abstract of your revised manuscript) to a maximum of 5 (7 keywords are currently listed); keywords should preferably be broad terms that will improve online search engine discoverability of your article.
- Please rename heading "Materials and Methods" to "Methods".
- Thank you for providing the referees access to your deposited datasets. Please make sure that all datasets will be publicly available at the time of publication. The reviewer access username/password can now be removed from the Data availability statement of your manuscript. Instead, please make sure that the database, identifier, and permanent and specific URL are provided for each dataset.
- The author contributions statement should be removed from the manuscript file. Instead, we use CRediT to specify the contributions of each author in the journal submission system. Please feel free to use the free text box to provide more detailed descriptions during submission. See also our guide to authors for more information: <https://www.embopress.org/page/journal/14602075/authorguide#authorshipguidelines>.
- We noticed that callouts for Fig. 7A are missing.
- "Table 1" and its callouts should be renamed to "Table EV1"; its legend must be included above the table in the same Excel file.
- The Movie files and their corresponding callouts should be renamed to "Movie EV1" and "Movie EV2"; their legends should be removed from the main manuscript file and instead zipped together with each Movie file.
- The heading on the title page of the Appendix PDF file should be "Appendix for:", followed by the manuscript's title and a Table of Contents including page numbers for all listed items.
- Please note that EMBO press papers are accompanied online by:
 - A) a short (2 sentences) summary of the findings and their significance,
 - B) 2-5 short bullet points highlighting the key results, and
 - C) a synopsis image in .jpg or .png format that is exactly 550 pixels wide and 300-600 pixels high (the height is variable). Please note that all text needs to be legible at the final size.Please upload this information along with your revised manuscript (the text for A and B should be provided in a separate Word file).
- During our routine data checks, our data editors have raised the following queries regarding figures, data, and legends. Please make sure that all requests below are completely addressed in the final version of your manuscript (please highlight all changes in the revised manuscript):
 1. Your Figure panel 1D contains graphs showing the mean and SD of n=2 biological replicates. Please note that, as per our journal's policy, no statistics can be shown or discussed when n=2; instead, the individual data points should be shown.
 2. Please provide the exact p-values in the legends of Figures 1D, F; 4D, 5D, 6B, 7C.
 3. Please note that the scale bar needs to be defined for Figures EV5 A, C.
 4. Please note that the scale bar and its definition are missing for Figure EV5 B.

Please also note that as part of the EMBO publications' Transparent Editorial Process, The EMBO Journal publishes online a Peer Review File along with each accepted manuscript. This File will be published in conjunction with your paper and will include the referee reports, your point-by-point response and all pertinent correspondence relating to the manuscript. You can opt out of this by letting the editorial office know (contact@embojournal.org). If you do opt out, the Peer Review File link will point to the following statement: "No Peer Review File is available with this article, as the authors have chosen not to make the review process public in this case."

We look forward to seeing a final version of your manuscript as soon as possible. Please let us know if you have any questions and use this link to submit your revision: <https://emboj.msubmit.net/cgi-bin/main.plex>.

Best regards,

Ioannis

Referee #1:

For the most part, the authors have addressed the reviewers' concerns.

The challenges associated with removing the MBP tag, or any solubilizing tag, from the TDP-43 construct are well taken, given the difficulty of maintaining the protein in a soluble form. However, the use of an alternative tag at the N-terminus could be considered to minimize potential unintended effects of tagging the C-tail and of MBP. Since experiments without the MBP tag were not performed, the limitations of maintaining the MBP tag for the phase separation and fibril formation experiments, already noted in the response letter, should be clearly stated and discussed in the manuscript.

Referee #2:

The authors have extensively addressed the comments of all reviewers. The topic is challenging and so it is not an easy task to make everything clear but I think the authors have done a good job. A few additional comments remain that are all minor.

Previous reviewer comment: "In addition, Fig 3 shows strong colocalization of MBP alone with Tau. The authors should discuss this observation and the impact of this behavior on the other results that involve MBP-tagged proteins."

Author response: "Indeed, the image shown in old Fig. 3 showed mild colocalization of Tau with MBP, which was further emphasized by the close-up showing one especially bright red droplet. To address this, we have replaced the image with a more representative image which now shows no MBP co-localization with Tau droplet (new Fig. 4). Importantly, the key phenotypes (enhanced Tau condensation and formation of small, internal Tau droplets) are only seen in the presence of TDP-43-MBP, and not MBP or -synuclein, suggesting that the effect is primarily driven by the TDP-43 protein itself."

New comment: the authors should explain why MBP shows mild colocalization sometimes and no colocalization other times. Some type of quantification would help - even fluorescence intensity in images of MBP alone vs the fusion. I would say this is most concerning to me because seems like it should not disappear and hard to say what is "more representative" without seeing through the eyes of the author who may have done this dozens of times but we reviewers are not given proof of that.

Author response: "These new results resolve the previously conflicting observations and clearly demonstrate that only full-length TDP-43 and its N-terminal fragment (aa. 1-273) modulate Tau aggregation and seeding."

New comment: The authors should carefully explain why the NTF has this effect. Could the authors test the mechanism?

Author response: "While phase separation/protein condensation can occur rapidly under optimal conditions, it is still a progressive process, in which the number/size of condensates increases over time (Oswald ripening)."

New comment: yes, condensates change vs time - in this case it is may be from fusion of droplets into larger ones, not just Oswald ripening. But in any case I agree. Please change text to say "gradually formed larger condensates over time" to make it

clear to the reader that the condensate size is changing over time possibly from condensates surpassing the diffraction limit over time as they fuse or ripen, not that condensation takes time in the way that aggregation takes time (has a lag phase) - even ripening is smaller droplets losing out to larger ones, but the saturation concentration and hence the dilute phase concentration does not change vs time in the same way that it does in aggregation experiments.

Referee #3:

Tau/TDP43 paper.

Over the past year the authors have undertaken extensive changes to the paper including additional analyses and experimental techniques and, not least, have had to reassess their earlier interpretations. The current conclusion encompasses an inhibitory effect of TDP43 upon tau in co-aggregates that may diminish tau pathogenicity, yet give way to a state where accumulating TDP43 seeds produce their own detrimental impact upon brain functioning. Molecular profiling and stratification of the human brain material (shown across several display items) has added to the package and, given the common occurrence of co-pathologies, the area is of great interest. The paper is recommended for publication.

Minor point

Fig S4 title would be better to say "TDP43-MBP, but not MBP..."

Very minor point not requiring a response.

The authors mention a new graphic abstract that captures the summary presented in the old Fig.7. This could be important to unlocking key messages in the paper, but it was not provided.

Point-by-point response to the editor's and reviewers' comments

We are very happy that all three referees are supportive of our publication in *The EMBO Journal* and thank them for taking the time to reevaluate our manuscript and making further suggestions for improvement. As suggested, we have now also addressed any remaining comments by further textual revision / clarification, as outlined here and marked up in the manuscript by yellow highlights.

Editorial comments:

- Regarding the funding information entered in our online manuscript handling system, please add all funders to the "More Funders" list rather than to the Comments box.

This has been done.

- Please reduce the number of keywords (after the Abstract of your revised manuscript) to a maximum of 5 (7 keywords are currently listed); keywords should preferably be broad terms that will improve online search engine discoverability of your article.

We have now limited the number of keywords to 5.

- Please rename heading "Materials and Methods" to "Methods".

This has been done.

- Thank you for providing the referees access to your deposited datasets. Please make sure that all datasets will be publicly available at the time of publication. The reviewer access username/password can now be removed from the Data availability statement of your manuscript. Instead, please make sure that the database, identifier, and permanent and specific URL are provided for each dataset.

This has been done and we made sure that all datasets are now publicly available through the provided links and identifiers.

- The author contributions statement should be removed from the manuscript file. Instead, we use CRediT to specify the contributions of each author in the journal submission system. Please feel free to use the free text box to provide more detailed descriptions during submission. See also our guide to authors for more information: <https://www.embopress.org/page/journal/14602075/authorguide#authorshipguidelines>.

We have now removed the author contributions statement from the manuscript file and entered the respective contribution(s) of each author in the journal submission system.

- We noticed that callouts for Fig. 7A are missing.

Callout of Fig. 7A has been added to the manuscript text (p. 9).

- "Table 1" and its callouts should be renamed to "Table EV1"; its legend must be included above the table in the same Excel file.

We have renamed the table and its callouts and included the legend into the Excel file.

- The Movie files and their corresponding callouts should be renamed to "Movie EV1" and "Movie EV2"; their legends should be removed from the main manuscript file and instead zipped together with each Movie file.

This has been done.

- The heading on the title page of the Appendix PDF file should be "Appendix for:", followed by the manuscript's title and a Table of Contents including page numbers for all listed items.

This has been done.

- Please note that EMBO press papers are accompanied online by:

A) a short (2 sentences) summary of the findings and their significance,

B) 2-5 short bullet points highlighting the key results, and

C) a synopsis image in .jpg or .png format that is exactly 550 pixels wide and 300-600 pixels high (the height is variable). Please note that all text needs to be legible at the final size. Please upload this information along with your revised manuscript (the text for A and B should be provided in a separate Word file).

We provide the text for A) and B) in a separate Word file and submit a synopsis image with the requested width.

- During our routine data checks, our data editors have raised the following queries regarding figures, data, and legends. Please make sure that all requests below are completely addressed in the final version of your manuscript (please highlight all changes in the revised manuscript):

1. Your Figure panel 1D contains graphs showing the mean and SD of $n=2$ biological replicates. Please note that, as per our journal's policy, no statistics can be shown or discussed when $n=2$; instead, the individual data points should be shown.

We have removed the statistics for this dataset and the individual data points are now shown in Fig. 1D.

2. Please provide the exact p-values in the legends of Figures 1D, F; 4D, 5D, 6B, 7C.

For Fig. 1D: We have removed the statistics (see point 1 above).

For Fig. 1F, 5D, 6B and 7C: The software we used (GraphPad Prism) does not provide exact p-values when they are very small, hence we unfortunately cannot provide the exact value. We clarified with the handling editor (Ioannis Papaioannou) that this is fine.

4D: This has been done.

3. Please note that the scale bar needs to be defined for Figures EV5 A, C.

This has been done.

4. Please note that the scale bar and its definition are missing for Figure EV5 B.

This has been done.

Please also note that as part of the EMBO publications' Transparent Editorial Process, The EMBO Journal publishes online a Peer Review File along with each accepted manuscript. This File will be published in conjunction with your paper and will include the referee reports, your point-by-point response and all pertinent correspondence relating to the manuscript. You can opt out of this by letting the editorial office know (contact@embojournal.org). If you do opt out, the Peer Review File link will point to the following statement: "No Peer Review File is available with this article, as the authors have chosen not to make the review process public in this case."

We are happy that EMBO Journal publishes the Peer Review File along with our accepted manuscript.

Referee #1:

For the most part, the authors have addressed the reviewers' concerns. The challenges associated with removing the MBP tag, or any solubilizing tag, from the TDP-43 construct are well taken, given the difficulty of maintaining the protein in a soluble form. However, the use of an alternative tag at the N-terminus could be considered to minimize potential unintended effects of tagging the C-tail and of MBP. Since experiments without the MBP tag were not performed, the limitations of maintaining the MBP tag for the phase separation and fibril formation experiments, already noted in the response letter, should be clearly stated and discussed in the manuscript.

We agree with the reviewer that it would be good to state and discuss this limitation, hence we have now added the following paragraph to our Discussion: "It should be noted that to analyze the impact of soluble TDP-43 on Tau condensate/fibril formation (Fig. 4 and 5), we had to retain the MBP tag, as removal of this solubility tag leads to immediate TDP-43 condensation above $\sim 1 \mu\text{M}$ concentration. Thus, we were unable to study the impact of soluble, monomeric TDP-43 on Tau without the MBP tag. Given this limitation, validation

studies with alternative tags or a solubility tag at the N-terminus of TDP-43 should be conducted in the future”.

Referee #2:

The authors have extensively addressed the comments of all reviewers. The topic is challenging and so it is not an easy task to make everything clear but I think the authors have done a good job. A few additional comments remain that are all minor.

Previous reviewer comment: "In addition, Fig 3 shows strong colocalization of MBP alone with Tau. The authors should discuss this observation and the impact of this behavior on the other results that involve MBP-tagged proteins."

Author response: "Indeed, the image shown in old Fig. 3 showed mild colocalization of Tau with MBP, which was further emphasized by the close-up showing one especially bright red droplet. To address this, we have replaced the image with a more representative image which now shows no MBP co-localization with Tau droplet (new Fig. 4). Importantly, the key phenotypes (enhanced Tau condensation and formation of small, internal Tau droplets) are only seen in the presence of TDP-43-MBP, and not MBP or α -synuclein, suggesting that the effect is primarily driven by the TDP-43 protein itself."

New comment: the authors should explain why MBP shows mild colocalization sometimes and no colocalization other times. Some type of quantification would help - even fluorescence intensity in images of MBP alone vs the fusion. I would say this is most concerning to me because seems like it should not disappear and hard to say what is "more representative" without seeing through the eyes of the author who may have done this dozens of times but we reviewers are not given proof of that.

We apologize if our statement that we have chosen "a more representative image" has raised confusion. We confirm that we have chosen a highly representative image from an experiment that has been replicated 6 times and always yielded similar results shown. Since the difference in signal intensity between TDP-43-MBP and MBP in Tau condensates is vastly different, i.e. most pixels in the TDP-43-MBP channel are saturated and most pixels in the MBP-only channel are close to zero, we cannot reliably quantify it.

To assure the reader that we have chosen a highly representative image and the results were always consistent, we have now added the following statement to the legend of Fig. 4B: "Images shown are representative images from one experiment out of 6 experimental replicates, all of which yielded similar results."

Author response: "These new results resolve the previously conflicting observations and clearly demonstrate that only full-length TDP-43 and its N-terminal fragment (aa. 1-273) modulate Tau aggregation and seeding."

New comment: The authors should carefully explain why the NTF has this effect. Could the authors test the mechanism?

We cannot provide a definitive answer as to why the TDP-43 NTF has this effect. We can only speculate that the high negative net charge of the TDP-43 NTF ($z = -8.3$ at pH 7.4) could promote its association with the positively charged Tau protein ($z = +1.44$ at pH 7.4), while TDP-43 CTF also carries a positive charge and may be repelled ($z = +2.2$ at pH 7.4), but this a very simplified view. Our cross-linking mass spectrometry experiment under TDP-43 condensation conditions also demonstrate a direct interaction of the TDP-43 N-terminus with the Tau proline-rich domain (Fig. 2). These experiments, however, do not exclude an involvement of the C-terminal low complexity region of TDP-43, hence further interaction studies, e.g. with additional truncation/mutant constructs or different methods, such as nuclear magnetic resonance (NMR) spectroscopy, are needed in the future to provide additional molecular insights into the TDP-43-Tau interaction. To point this out, we have added the following paragraph to the discussion on p. 11:

"Moreover, we found that the TDP-43 NTD interacts with Tau's PRD region (aa. 174 and 225) in the condensed state (Fig. 2), possibly promoted by the high negative charge of the TDP-43 NTF ($z = -8.3$ at pH 7.4). To obtain a complete molecular picture, further interaction studies,

e.g. experiments with additional truncation/mutant proteins or high resolution structural methods, such as nuclear magnetic resonance (NMR) spectroscopy, should be performed in the future.”

Author response: "While phase separation/protein condensation can occur rapidly under optimal conditions, it is still a progressive process, in which the number/size of condensates increases over time (Oswald ripening)."

New comment: yes, condensates change vs time - in this case it may be from fusion of droplets into larger ones, not just Ostwald ripening. But in any case I agree. Please change text to say "gradually formed larger condensates over time" to make it clear to the reader that the condensate size is changing over time possibly from condensates surpassing the diffraction limit over time as they fuse or ripen, not that condensation takes time in the way that aggregation takes time (has a lag phase) - even ripening is smaller droplets losing out to larger ones, but the saturation concentration and hence the dilute phase concentration does not change vs time in the same way that it does in aggregation experiments.

As suggested by the reviewer, we have now changed the statement in the results text to "Tau alone gradually formed larger condensates over time" (p. 5).

Referee #3:

Tau/TDP43 paper.

Over the past year the authors have undertaken extensive changes to the paper including additional analyses and experimental techniques and, not least, have had to reassess their earlier interpretations. The current conclusion encompasses an inhibitory effect of TDP43 upon tau in co-aggregates that may diminish tau pathogenicity, yet give way to a state where accumulating TDP43 seeds produce their own detrimental impact upon brain functioning. Molecular profiling and stratification of the human brain material (shown across several display items) has added to the package and, given the common occurrence of co-pathologies, the area is of great interest. The paper is recommended for publication.

Minor point:

Fig S4 title would be better to say "TDP43-MBP, but not MBP..."

We assume that the reviewer meant Fig. 4 and have accordingly changed the title of Fig. 4 to "TDP-43-MBP, but not MBP or α -synuclein, causes the formation of large, amorphous Tau condensates in vitro". We agree with the reviewer that this is a good addition to this figure title.

Very minor point not requiring a response.

The authors mention a new graphic abstract that captures the summary presented in the old Fig.7. This could be important to unlocking key messages in the paper, but it was not provided. We apologize for the confusion. We meant to refer to the synopsis image that we provided along with our submission, we apologize if it was not clear that we were referring to this file when we mentioned the "new graphical abstract".

Dear Dorothee,

Congratulations on an excellent work! I am very pleased to inform you that your manuscript has been accepted for publication in The EMBO Journal. Thank you for comprehensively addressing the initially raised referee concerns and all editorial requests for changes and corrections.

Please send me (by e-mail) the corrected Figure 1 and the revised legend for its panel D when you get a chance, so that we can move forward with the production of your article.

Your manuscript will then be processed for publication by EMBO Press. It will be copy edited and you will receive page proofs prior to publication. Please note that you will be contacted by Springer Nature Author Services to complete licensing and payment information.

If you have any questions, please do not hesitate to contact the Editorial Office. Thank you for your contribution to The EMBO Journal. Working with you has been a pleasure!

Best wishes,

Ioannis
